# Complexity Lower Bounds of Adaptive Gradient Algorithms for Non-convex Stochastic Optimization under Relaxed Smoothness

**Michael Crawshaw**
Department of Computer Science
George Mason University
Fairfax, VA 22030, USA
`mcrawsha@gmu.edu`

**Mingrui Liu**
Department of Computer Science
George Mason University
Fairfax, VA 22030, USA
`mingruil@gmu.edu`

## Abstract

Recent results in non-convex stochastic optimization demonstrate the convergence of popular adaptive algorithms (e.g., AdaGrad) under the $(L_0, L_1)$-smoothness condition, but the rate of convergence is a higher-order polynomial in terms of problem parameters like the smoothness constants. The complexity guaranteed by such algorithms to find an $\epsilon$-stationary point may be significantly larger than the optimal complexity of $\Theta\left(\Delta L \sigma^2 \epsilon^{-4}\right)$ achieved by SGD in the $L$-smooth setting, where $\Delta$ is the initial optimality gap, $\sigma^2$ is the variance of stochastic gradient. However, it is currently not known whether these higher-order dependencies can be tightened. To answer this question, we investigate complexity lower bounds for several adaptive optimization algorithms in the $(L_0, L_1)$-smooth setting, with a focus on the dependence in terms of problem parameters $\Delta, L_0, L_1$. We provide complexity bounds for three variations of AdaGrad, which show at least a quadratic dependence on problem parameters $\Delta, L_0, L_1$. Notably, we show that the decorrelated variant of AdaGrad-Norm requires at least $\Omega\left(\Delta^2 L_1^2 \sigma^2 \epsilon^{-4}\right)$ stochastic gradient queries to find an $\epsilon$-stationary point. We also provide a lower bound for SGD with a broad class of adaptive stepsizes. Our results show that, for certain adaptive algorithms, the $(L_0, L_1)$-smooth setting is fundamentally more difficult than the standard smooth setting, in terms of the initial optimality gap and the smoothness constants.

## 1 Introduction

The best performing optimization algorithms for modern deep learning are gradient-based optimizers with adaptive step sizes. For today's large-scale deep learning tasks, such as training Large Language Models (LLMs), classical non-adaptive optimizers like SGD perform significantly worse than their adaptive counterparts, such as Adam (Kingma & Ba, 2014) and AdamW (Loshchilov & Hutter, 2018). However, it remains open to theoretically characterize the efficiency of adaptive gradient algorithms for non-convex optimization.

An increasingly popular framework for describing optimization in deep learning is $(L_0, L_1)$-smoothness, also known as relaxed smoothness (Zhang et al., 2020b). The conventional smoothness condition asserts that the norm of the objective's Hessian is upper bounded by a constant, while the weaker relaxed smoothness enforces only that the Hessian norm is upper bounded by an affine function of the gradient norm (see Assumption 1). Empirical evidence suggests that this condition may characterize neural network training (for certain architectures) more accurately than conventional smoothness (Zhang et al., 2020b; Crawshaw et al., 2022).

Several recent works analyze the efficiency of adaptive algorithms for non-convex optimization, particularly AdaGrad-Norm (Li & Orabona, 2019; Ward et al., 2020; Wang et al., 2023; Attia & Koren, 2023; Faw et al., 2023) and AdaGrad (Wang et al., 2023). Indeed, adaptive algorithms are suited for relaxed smoothness, since the local curvature of a relaxed smooth objective can be determined from gradient information, and adaptive algorithms adjust their step size based on gradients. Existing

Table 1: Iteration complexity to find an $\epsilon$-stationary point in non-convex stochastic optimization. $\Delta$ is the initial optimality gap, $L$ and $(L_0, L_1)$ are the smoothness constants for the smooth and relaxed smooth cases, respectively. $\sigma$ and $(\sigma_1, \sigma_2)$ are constants bounding the stochastic gradient noise, depending on the stochastic assumption. See Assumptions 1 and 2 for the full definitions. $\gamma$ is the stabilization constant of AdaGrad. $*$ denotes a high-probability guarantee with failure probability $\delta$. $\gamma_1, \gamma_2, \gamma_3$ are defined and discussed in Section 6.

| | Complexity | Stochasticity |
|---|:---:|:---:|
| *L*-smooth | | |
| SGD (Ghadimi & Lan, 2013) | $\Theta\left(\frac{\Delta L \sigma^2}{\epsilon^4} + \frac{\Delta L}{\epsilon^2}\right)$ | (Bounded-Var) |
| Decorrelated AdaGrad-Norm (Li & Orabona, 2019) | $\mathcal{O}\left(\left(\frac{\Delta L \sigma^2}{\epsilon^4} + \frac{\Delta L}{\epsilon^2}\right)\left(1 + \frac{\Delta L + \sigma^2}{\gamma^2}\right) + \frac{\sqrt{\Delta L (\gamma^2 + \sigma^2)}}{\epsilon^2}\right)$ | (Subgaussian) |
| AdaGrad-Norm (Wang et al., 2023) | $\tilde{\mathcal{O}}\left(\left(\Delta L + \frac{\Delta^2 L^2 \sigma_2^4 + \sigma_1^4}{\gamma^2}\right)\left(\frac{\sigma_1^2}{\delta^4 \epsilon^4} + \frac{\sigma_2^2}{\delta^2 \epsilon^2}\right)\right)^*$ | (Affine-Var) |
| AdaGrad-Norm (Attia & Koren, 2023) | $\tilde{\mathcal{O}}\left(\frac{\Delta L \sigma_1^2(\sigma_2^2+1)+\sigma_1^4}{\epsilon^4} + \frac{\Delta L(1+\sigma_2^2)+\gamma\sqrt{\Delta L(1+\sigma_2^2)+\sigma_1^2(1+\sigma_2^2)}}{\epsilon^2}\right)^*$ | (Affine-Noise) |
| AdaGrad-Norm (Yang et al., 2024) | $\mathcal{O}\left(\frac{\Delta L \sigma^2 + \sigma^4}{\epsilon^4} + \frac{\Delta L + \gamma\sqrt{\Delta L} + \sigma^2 + \gamma\sigma}{\epsilon^2}\right)$ | (Bounded-Var) |
| $(L_0, L_1)$-smooth | | |
| SGD (Li et al., 2024) | $\mathcal{O}\left(\frac{(\Delta+\sigma)^4 L_1^2}{\delta^4 \epsilon^4} + \frac{(\Delta+\sigma)^3 L_0}{\delta^3 \epsilon^4} + \frac{(\Delta+\sigma)^2 L_1^2}{\delta^2 \epsilon^2} + \frac{(\Delta+\sigma)L_0}{\delta \epsilon^2}\right)^*$ | (Bounded-Var) |
| Gradient Clipping Zhang et al. (2020b;a) | $\mathcal{O}\left(\frac{\Delta L_0 \sigma^2}{\epsilon^4}\right)$ | (Bounded-Noise) |
| Gradient Clipping Koloskova et al. (2023) | $\mathcal{O}\left(\frac{\Delta L_1 \sigma^4}{\epsilon^5} + \frac{\Delta}{\epsilon^2}\left(\frac{\sigma^2}{\epsilon^2} + 1\right)\left(L_0 + \sqrt{L_0 L_1 \epsilon} + L_1 \epsilon\right)\right)$ | (Bounded-Var) |
| AdaGrad-Norm Wang et al. (2023) | $\tilde{\mathcal{O}}\left(\left(\Delta^2 L_1^2(1+\sigma_2^4) + \Delta L_0 + \frac{(\Delta^4 L_1^4 + \Delta^2 L_0^2)\sigma_2^4 + \sigma_1^4}{\gamma^2}\right)\left(\frac{\sigma_1^2}{\delta^4 \epsilon^4} + \frac{\sigma_2^2}{\delta^2 \epsilon^2}\right)\right)^*$ | (Affine-Var) |
| Decorrelated AdaGrad-Norm (Theorem 1) | $\tilde{\Omega}\left(\frac{\Delta^2 L_1^2 \sigma^2}{\epsilon^4} + \frac{\Delta L_0 \sigma^2}{\epsilon^4} + \frac{\Delta^2 L_1^2}{\epsilon^2}\right)$ | (Bounded-Noise) |
| Decorrelated AdaGrad (Theorem 2) | $\tilde{\Omega}\left(\frac{\Delta^2 L_0^2 \sigma^2}{\gamma^2 \epsilon^4} + \frac{\Delta^2 L_1^2 \sigma^2}{\gamma^2 \epsilon^2}\right)$ | (Bounded-Noise) |
| AdaGrad (Theorem 3) | $\tilde{\Omega}\left(\frac{\Delta^2 L_0^2}{\epsilon^4} + \frac{\Delta^2 L_1^2}{\epsilon^2}\right)$ | (Bounded-Noise) |
| Single-step Adaptive SGD (Theorem 4) | $\tilde{\Omega}\left(\frac{\Delta L_0 \sigma_1^2}{\epsilon^4} + \frac{(\Delta L_1)^{2-\gamma_2-\gamma_3}\sigma_1^{\gamma_2+\gamma_3-\gamma_1}}{\epsilon^{2-\gamma_1}}\right)$ | (Affine-Noise) |

works demonstrate that AdaGrad can find an $\epsilon$-stationary point with iteration complexity that scales as $\epsilon^{-4}$ in terms of $\epsilon$, which matches the complexity of SGD in the stochastic, non-convex setting. However, these guarantees also show that the complexity of AdaGrad (and some variants) is upper bounded by a higher-order polynomial (i.e., at least quadratic) in terms of problem parameters such as $\Delta$ (initial optimality gap), $\sigma^2$ (variance of stochastic gradient), and the smoothness constants. See Table 1 for a summary of these guarantees. This suggests the following question:

> ***Can AdaGrad-type algorithms converge under relaxed smoothness without a higher-order polynomial complexity in terms of problem parameters?***

In this paper, we answer this question negatively for several variants of the AdaGrad algorithm by providing complexity lower bounds that scale quadratically in terms of the problem parameters $\Delta, L_0, L_1$. Our results are summarized in Table 1. This shows that, in the non-convex, stochastic, relaxed smooth settings, the variants of AdaGrad considered here cannot recover the $\Delta L_0 \sigma^2 \epsilon^{-4}$ complexity from the $L_0$-smooth case; in this sense, these algorithms suffer a fundamental difficulty in the relaxed smooth setting. In comparison, SGD with gradient clipping does achieve the classical complexity of $\Delta L_0 \sigma^2 \epsilon^{-4}$ under the same setting as investigated in our lower bounds (Zhang et al., 2020a), which shows the surprising consequence that SGD with gradient clipping outperforms AdaGrad in this setting. Additionally, we give a lower bound for adaptive SGD with a broad class of adaptive step sizes, in a setting where stochastic gradient noise scales linearly with the gradient norm.

We emphasize that the complexity's dependence on problem parameters can be important for distinguishing the relative performance of optimization algorithms. A classic example is the case of smooth, strongly convex functions, where both gradient descent and Nesterov's Accelerated Gradient

(NAG) exhibit linear convergence, but the iteration complexity of NAG is faster than GD by a factor of $\sqrt{\kappa}$, where $\kappa$ is the condition number of the objective function (Nesterov, 2013).

Our contributions can be summarized as follows:

1. In Theorem 1, we provide a complexity lower bound of $\Omega\left(\Delta^2 L_1^2 \sigma^2 \epsilon^{-4}\right)$ for Decorrelated AdaGrad-Norm (which uses decorrelated step sizes and a shared learning rate for all coordinates) under $(L_0, L_1)$-smoothness and almost surely bounded gradient noise. The proof uses a novel construction of a difficult objective for which Decorrelated AdaGrad-Norm may diverge (depending on the choice of hyperparameter), combined with a high-dimensional objective (adapted from Drori & Shamir (2020)). This lower bound matches the upper bound of AdaGrad-Norm in two out of three dominating terms, and only differs in terms of $\sigma$. See Section 4 for a comparison between these upper and lower bounds.

2. In Theorem 2, we lower bound the complexity of Decorrelated AdaGrad by $\Omega\left(\Delta^2 L_0^2 \sigma^2 \gamma^{-2} \epsilon^{-4}\right)$, where $\gamma$ is a hyperparameter. The proof uses a novel high-dimensional objective for which the algorithm diverges when $\eta \geq \tilde{\Omega}(\gamma/(L_1\sigma))$. Theorem 3 extends this result for the original AdaGrad algorithm, achieving a lower bound of $\Omega\left(\Delta^2 L_0^2 \epsilon^{-4}\right)$. While our lower bound for AdaGrad is weaker than for its decorrelated counterpart, this complexity is still larger than the optimal smooth rate in regimes when $\Delta$ or the smoothness constants are large compared to $\sigma$.

3. In Theorem 4, we consider the setting of $(L_0, L_1)$-smoothness and gradient noise bounded by an affine function of the gradient norm. For SGD with a broad class of adaptive step sizes, we show a lower bound that is nearly quadratic in the problem parameters $\Delta, L_1$. This is proven by showing that one of the following must hold: (1) adaptive SGD can be forced into a biased random walk with a constant probability of divergence, or (2) the adaptive step size is $\mathcal{O}\left(1/(\Delta L_1^2)\right)$ when optimizing a function with gradient magnitude equal to $\epsilon$, which leads to slow convergence.

The remainder of the paper is structured as follows. We discuss related work in Section 2, then give the formal problem statement in Section 3. We then present our complexity lower bounds for Decorrelated AdaGrad-Norm (Section 4), Decorrelated AdaGrad and the original AdaGrad (Section 5), and adaptive SGD (Section 6). We conclude with Section 7.

## 2 RELATED WORK

**Relaxed Smoothness.** Relaxed smoothness was introduced by Zhang et al. (2020b), who showed that GD with normalization converges faster than GD under this condition. This inspired a line of work focusing on efficient algorithms under this condition. Zhang et al. (2020a) showed an improved analysis of gradient clipping, and Jin et al. (2021) considered a non-convex distributionally robust optimization satisfying this condition. Several recent works (Liu et al., 2022; Crawshaw et al., 2023a;b) designed communication-efficient federated learning algorithms under relaxed smoothness. Li et al. (2024) analyzed gradient-based methods without gradient clipping under generalized smoothness. Crawshaw et al. (2022) studied a coordinate-wise version of relaxed smoothness, empirically showed that transformers satisfy this condition, and designed a generalized signSGD algorithm with convergence guarantees. Chen et al. (2023) proposed a new notion of $\alpha$-symmetric generalized smoothness and analyzed a class of normalized GD algorithms. More recently, a few works have investigated momentum and variance reduction techniques within the framework of individual relaxed smooth conditions (Liu et al., 2023) or on average relaxed smooth conditions (Reisizadeh et al., 2023).

**Adaptive Gradient Methods.** Adaptive gradient optimization algorithms automatically adjust the step size for each coordinate based on gradient information, and have become very important in machine learning. Examples include Adagrad (Duchi et al., 2011; McMahan & Streeter, 2010), Adam (Kingma & Ba, 2014), RMSProp (Tieleman & Hinton, 2012), and other variants (Loshchilov & Hutter, 2018; Shazeer & Stern, 2018). Most theoretical analyses of adaptive optimization methods are based on the assumptions of smoothness or convexity (Reddi et al., 2018; Chen et al., 2018; Guo et al., 2021). Recently, some works established convergence results for AdaGrad-Norm (Faw et al., 2023; Wang et al., 2023) and Adam (Li et al., 2023) under the relaxed smoothness condition, and all of the convergence rates in these works exhibit a higher order polynomial dependence on $L_1$.

**Lower Bounds.** Lower bounds for first-order convex optimization are well studied (Nemirovskii et al., 1983; Nesterov, 2013; Woodworth & Srebro, 2017; 2016). The lower bounds of nonconvex smooth optimization were studied in the deterministic setting (Cartis et al., 2010; Carmon et al., 2020; 2021), finite-sum setting (Fang et al., 2018) and stochastic setting (Drori & Shamir, 2020; Arjevani et al., 2023). For relaxed smooth problems, Zhang et al. (2020b) and Crawshaw et al. (2022) derived a lower bound for GD and showed that its complexity depends on the maximum magnitude of the gradient in a sublevel set. Faw et al. (2023) considered the lower bound for normalized SGD, clipped SGD, and signSGD with momentum in the affine noise setting, and showed that these algorithms cannot converge under certain parameter regimes. Crawshaw et al. (2023b) showed a lower bound for minibatch SGD with gradient clipping in the affine noise setting.

## 3 PROBLEM STATEMENT

### 3.1 OPTIMIZATION OBJECTIVES

We consider the problem of finding an approximate stationary point of a nonconvex, relaxed smooth function with access to a stochastic gradient. Let $\Delta, L_0, L_1, \sigma_1, \sigma_2, \sigma > 0$. We will denote the objective function as $f : \mathbb{R}^d \to \mathbb{R}$, the stochastic gradient as $g : \mathbb{R}^d \times \Xi \to \mathbb{R}^d$, and the noise distribution as $\mathcal{D}$, which is a distribution over $\Xi$. We then consider the set of problem instances $(f, g, \mathcal{D})$ satisfying the following conditions:

**Assumption 1.** *(1)* $f$ *is bounded from below and* $f(\mathbf{0}) - \inf_{\boldsymbol{x}} f(\boldsymbol{x}) \leq \Delta$. *(2)* $f$ *is continuously differentiable and* $(L_0, L_1)$*-smooth: For every* $\boldsymbol{x}, \boldsymbol{y} \in \mathbb{R}^d$ *with* $\|\boldsymbol{x} - \boldsymbol{y}\| \leq 1/L_1$:

$$\|\nabla f(\boldsymbol{x}) - \nabla f(\boldsymbol{y})\| \leq (L_0 + L_1 \|\nabla f(\boldsymbol{x})\|)\|\boldsymbol{x} - \boldsymbol{y}\|.$$

*(3)* $\mathbb{E}_{\xi \sim \mathcal{D}}[g(\boldsymbol{x}, \xi)] = \nabla f(\boldsymbol{x})$ *for all* $\boldsymbol{x} \in \mathbb{R}^d$.

**Assumption 2.** *For all* $\boldsymbol{x} \in \mathbb{R}^d$:

(Bounded-Noise) $\|g(\boldsymbol{x}, \xi) - \nabla f(\boldsymbol{x})\| \leq \sigma$ *almost surely over* $\xi \sim \mathcal{D}$.

(Affine-Noise) $\|g(\boldsymbol{x}, \xi) - \nabla f(\boldsymbol{x})\| \leq \sigma_1 + \sigma_2 \|\nabla f(\boldsymbol{x})\|$ *almost surely over* $\xi \sim \mathcal{D}$.

(Bounded-Var) $\mathbb{E}_{\xi \sim \mathcal{D}}\left[\|g(\boldsymbol{x}, \xi) - \nabla f(\boldsymbol{x})\|^2\right] \leq \sigma^2$.

(Affine-Var) $\mathbb{E}_{\xi \sim \mathcal{D}}\left[\|g(\boldsymbol{x}, \xi) - \nabla f(\boldsymbol{x})\|^2\right] \leq \sigma_1^2 + \sigma_2^2\|\nabla f(\boldsymbol{x})\|^2$.

(Subgaussian) $\mathbb{E}_{\xi \sim \mathcal{D}}\left[\exp(\|g(\boldsymbol{x}, \xi) - \nabla f(\boldsymbol{x})\|^2/\sigma^2)\right] \leq 1$.

We will denote by $\mathcal{F}_{\mathrm{as}}(\Delta, L_0, L_1, \sigma)$ the set of problems $(f, g, \mathcal{D})$ satisfying Assumption 1 and (Bounded-Noise), and by $\mathcal{F}_{\mathrm{aff}}(\Delta, L_0, L_1, \sigma_1, \sigma_2)$ those satisfying Assumption 1 and (Affine-Noise).

In this paper, we present new results under (Bounded-Noise) and (Affine-Noise), though we state the other assumptions for discussion with related work. It is important to note that (Bounded-Noise) is strictly stronger than (Bounded-Var). Therefore, the lower bounds that we prove for $\mathcal{F}_{\mathrm{as}}$ also hold for the class of problems satisfying Assumption 1 and (Bounded-Var). This is because any difficult problem instance in the former class is also in the latter. An analogous statement for (Affine-Noise) and (Affine-Var) holds by the same reasoning. Our primary focus for stochasticity in this work is (Bounded-Noise), since this is the standard assumption used by early work on relaxed smoothness (Zhang et al., 2020b;a).

### 3.2 OPTIMIZATION ALGORITHMS

We will consider four optimization algorithms — Decorrelated AdaGrad-Norm, Decorrelated Ada-Grad, AdaGrad, and single-step adaptive SGD — and their behavior for problems in $\mathcal{F}_{\mathrm{as}}$ and $\mathcal{F}_{\mathrm{aff}}$.

**Decorrelated AdaGrad-Norm** We first consider a variant of AdaGrad that we refer to as Decorrelated AdaGrad-Norm:

$$\boldsymbol{x}_{t+1} = \boldsymbol{x}_t - \frac{\eta}{\sqrt{\gamma^2 + \sum_{i=0}^{t-1} \|\boldsymbol{g}_i\|^2}} \boldsymbol{g}_t, \tag{1}$$

where $\eta > 0$ is a step size coefficient, $\boldsymbol{g}_t = g(\boldsymbol{x}_t, \xi_t)$, and $\xi_t \sim \mathcal{D}$ is independent over $t$. Notice that the denominator contains the sum of squared gradient norms, as opposed to the coordinate-wise operations used in the original AdaGrad. This type of denominator is used in AdaGrad-Norm, whose convergence was studied under various conditions in Ward et al. (2020); Faw et al. (2022); Attia & Koren (2023); Wang et al. (2023); Yang et al. (2024). Further, the sum of squared gradients in the denominator ranges from $i = 0$ to $i = t - 1$, meaning that it does not contain the most recent stochastic gradient $g_t$. This type of decorrelated step size was considered in Li & Orabona (2019), which provided convergence guarantees in the smooth setting (see Table 1).

**AdaGrad and Decorrelated AdaGrad** Next, we consider two coordinate-wise variants of AdaGrad, including the original AdaGrad and a variation with a decorrelated step size. The original AdaGrad (Duchi et al., 2011) is defined as follows:

$$\boldsymbol{x}_{t+1} = \boldsymbol{x}_t - \frac{\eta}{\sqrt{\gamma^2 + \sum_{i=0}^{t} \boldsymbol{g}_i^2}} \boldsymbol{g}_t, \tag{2}$$

where the squaring $\boldsymbol{g}_i^2$ is performed element-wise. Decorrelated AdaGrad (Li & Orabona, 2019) is similarly defined as

$$\boldsymbol{x}_{t+1} = \boldsymbol{x}_t - \frac{\eta}{\sqrt{\gamma^2 + \sum_{i=0}^{t-1} \boldsymbol{g}_i^2}} \boldsymbol{g}_t, \tag{3}$$

the only difference from AdaGrad being that the sum in the denominator does not contain the gradient from the current step, so the step size at step $t$ is independent of the stochastic gradient noise at step $t$.

**Single-Step Adaptive SGD** Last, we consider a class of algorithms that implement stochastic gradient descent with an adaptive step size, but whose step size function only depends on the current gradient. For $\alpha : \mathbb{R}^d \to \mathbb{R}$, single-step adaptive SGD is defined as:

$$\boldsymbol{x}_{t+1} = \boldsymbol{x}_t - \alpha(\boldsymbol{g}_t)\boldsymbol{g}_t, \tag{4}$$

where again $\boldsymbol{g}_t = g(\boldsymbol{x}_t, \xi_t)$ and $\xi_t \sim \mathcal{D}$ is independent over $t$. At each step $t$, the update $\boldsymbol{x}_{t+1} - \boldsymbol{x}_t$ is determined completely by the stochastic gradient sampled at step $t$, hence the name "single-step". However, the step size in the direction $\boldsymbol{g}_t$ is computed as an *arbitrary* function $\alpha$ of the stochastic gradient. This class of algorithms includes SGD with constant step size, SGD with gradient clipping, and normalized SGD; it does not include Adam or AdaGrad.

## 3.3 COMPLEXITY

Given a problem $(f, g, \mathcal{D})$ and $\epsilon > 0$, the goal of an optimization algorithm $A$ is to find an $\epsilon$-approximate stationary point of $f$, that is, a point $\boldsymbol{x} \in \mathbb{R}^d$ such that $\|\nabla f(\boldsymbol{x})\| < \epsilon$. We want to characterize the number of gradient calls required by an algorithm to find such a point. Since an algorithm can only gain information about the objective $f$ through *stochastic* gradients, it cannot necessarily guarantee to find an $\epsilon$-stationary point, but it may find one in expectation or with high probability. Denote by $\{\boldsymbol{x}_t\}$ the sequence of points at which the stochastic gradient is queried by $A$ when given $(f, g, \mathcal{D})$ as input. We then define the worst-case complexity of $A$ on problem class $\mathcal{F}$ as

$$\mathcal{T}(A, \mathcal{F}, \epsilon) = \sup_{(f,g,\mathcal{D}) \in \mathcal{F}} \min \left\{ t \geq 1 \ \middle| \ \min_{s < t} \mathbb{E}\left[\|\nabla f(\boldsymbol{x}_s)\|\right] < \epsilon \right\}.$$

To summarize, the worst-case complexity $\mathcal{T}(A, \mathcal{F}, \epsilon)$ measures the number of gradient calls required by $A$ to find an $\epsilon$-approximate stationary point in expectation, for any problem in $\mathcal{F}$. We also consider the worst-case complexity for finding an $\epsilon$-stationary point with high probability:

$$\mathcal{T}(A, \mathcal{F}, \epsilon, \delta) = \sup_{(f,g,\mathcal{D}) \in \mathcal{F}} \min \left\{ t \geq 1 \ \middle| \ \Pr\left(\min_{s < t} \|\nabla f(\boldsymbol{x}_s)\| < \epsilon\right) > 1 - \delta \right\}.$$

Following Arjevani et al. (2023), most of our results (Theorems 1, 2, 3) will provide in-expectation lower bounds, that is, lower bounds for $\mathcal{T}(A, \mathcal{F}, \epsilon)$. Our last result (Theorem 4) will provide lower bounds for $\mathcal{T}(A, \mathcal{F}, \epsilon, \delta)$ for any given $\delta$, i.e., high-probability lower bounds. Throughout the paper, $\mathcal{O}(\cdot), \Omega(\cdot)$ and $\Theta(\cdot)$ omit universal constants, and $\tilde{\mathcal{O}}(\cdot), \tilde{\Omega}(\cdot)$, and $\tilde{\Theta}(\cdot)$ omit universal constants and factors logarithmic in terms of problem parameters $\Delta, L_0, L_1, \sigma_1, \sigma_2, \sigma$, and target gradient norm $\epsilon$.

## 4 DECORRELATED ADAGRAD-NORM

Our first result gives a lower bound for Decorrelated AdaGrad-Norm, which shows that the complexity has a quadratic dependence in terms of problem parameters $\Delta, L_1$.

**Theorem 1.** *Denote $\mathcal{F} = \mathcal{F}_{as}(\Delta, L_0, L_1, \sigma)$, and let algorithm $A_{DAN}$ denote Decorrelated AdaGrad-Norm (Equation 1) with parameters $\eta > 0$ and $0 < \gamma \leq \tilde{\mathcal{O}}(\Delta L_1)$. Let $0 < \epsilon \leq \mathcal{O}\left(\min\left\{\sqrt{\Delta L_0}, \sqrt{\Delta L_1 \gamma}, \Delta L_1\right\}\right)$. If $\Delta L_1^2 \geq L_0$, then*

$$\mathcal{T}(A_{DAN}, \mathcal{F}, \epsilon) \geq \Omega\left(\frac{\Delta^2 L_1^2 \sigma^2}{\epsilon^4} + \frac{\Delta L_0 \sigma^2 \log(1 + \sigma^2/\gamma^2)}{\epsilon^4} + \frac{\Delta^2 L_1^2}{\epsilon^2}\right).$$

The proof is given in Appendix A. Before giving a sketch of the proof, we make a few observations about the result. **(1)** The lower bound contains the term $\Delta L_0 \sigma^2 \epsilon^{-4}$, which is the optimal complexity for a first-order algorithm in the $L_0$-smooth, non-convex, stochastic setting (Arjevani et al., 2023). This means that Decorrelated AdaGrad-Norm requires *at least* as many iterations to solve the current problem as any first order algorithm requires to solve the smooth counterpart. **(2)** The dominating term is quadratic in $\Delta, L_1$. Therefore, **under relaxed smoothness, Decorrelated AdaGrad-Norm cannot recover the optimal complexity of the smooth case**, as one might hope. **(3)** In the deterministic case (i.e., $\sigma = 0$), the complexity is $\Delta^2 L_1^2 \epsilon^{-2}$, which is still quadratic in the problem parameters $\Delta, L_1$ and does not match the complexity achieved by deterministic GD in the $L_0$-smooth case, i.e., $\Delta L_0 \epsilon^{-2}$. **(4)** **This lower bound for Decorrelated AdaGrad-Norm matches the upper bound of AdaGrad-Norm in two out of three dominating terms**. The dominating terms of the upper bound of AdaGrad-Norm from Wang et al. (2023) (see Table 1) are

$$\tilde{\mathcal{O}}\left(\frac{\Delta^2 L_1^2 \sigma^2}{\epsilon^4} + \frac{\Delta L_0 \sigma^2}{\epsilon^4} + \frac{\sigma^6}{\gamma^2 \epsilon^4}\right). \tag{5}$$

The first two terms of this upper bound match our lower bound up to log terms. Note that this result (Theorem 8 from Wang et al. (2023)) uses (Bounded-Var), whereas we use (Bounded-Noise). However, their upper bound still applies for the stronger (Bounded-Noise) and our lower bound still applies for the weaker (Bounded-Var). The gap between our lower bound and this upper bound is the third term (due to the noise $\sigma$), which means that either (a) the upper bound can be decreased; (b) the lower bound can be increased; (c) Decorrelated AdaGrad-Norm differs from AdaGrad-Norm in its dependence on the noise $\sigma$; or (d) the gap is caused by the difference in noise assumptions.

Lastly, the condition $\Delta L_1^2 \geq L_0$ for Theorem 1 ensures that $(L_0, L_1)$-smoothness does not degenerate to $L$-smoothness. Indeed, Lemma 3.5 of Li et al. (2024) implies that $\|\nabla f(\boldsymbol{x})\| \leq \mathcal{O}(\Delta L_1)$ for every $\boldsymbol{x}$ with $f(\boldsymbol{x}) \leq f(\boldsymbol{0})$, so $\|\nabla^2 f(\boldsymbol{x})\| \leq L_0 + L_1 \|\nabla f(\boldsymbol{x})\| \leq \mathcal{O}\left(L_0 + \Delta L_1^2\right)$. Therefore, if the condition $\Delta L_1^2 \geq L_0$ fails, then any objective $f$ which is $(L_0, L_1)$-smooth is also $\Theta(L_0)$-smooth in a sublevel set containing the initial point. We also require an upper bound on the stabilization constant: $\gamma \leq \tilde{\mathcal{O}}(\Delta L_1)$, which covers all practical regimes in which $\gamma$ is chosen as a small constant. In Appendix E, we show that this condition can be removed in the deterministic setting while recovering the complexity lower-bound $\tilde{\Omega}(\Delta^2 L_1^2 \epsilon^{-2})$.

### 4.1 PROOF OUTLINE

The proof of Theorem 1 follows two cases, depending on the choice of the parameter $\eta$. If $\eta \geq 1/L_1$, then the algorithm can diverge on a fast growing function. On the other hand, if $\eta \leq 1/L_1$, then the algorithm converges slowly on a function with small gradient. This proof structure is similar to previous lower bounds under relaxed smoothness (Zhang et al., 2020b; Crawshaw et al., 2022), but our result requires significantly different constructions due to the structure of AdaGrad updates, and since previous bounds only achieve $\epsilon^{-2}$ dependence, whereas we show $\epsilon^{-4}$ dependence.

**Divergence when $\eta \geq 1/L_1$** We want the update size $\|\boldsymbol{x}_{t+1} - \boldsymbol{x}_t\|$ to be lower bounded by a constant, but the step size decreases over $t$ due to the sum of squared gradients in the denominator. Intuitively, this means that the gradient magnitude $\|\boldsymbol{g}_t\|$ should increase with $t$ to offset the decreasing step size. However, faster growth of $\|\boldsymbol{g}_t\|$ causes faster decrease in the effective step size. We can balance these two effects and force the trajectory to diverge with a properly constructed objective function and a sequence of gradients satisfying $\|\boldsymbol{g}_t\| = \Theta\left((t \log t)^t\right)$, which is executed in Lemma 1.

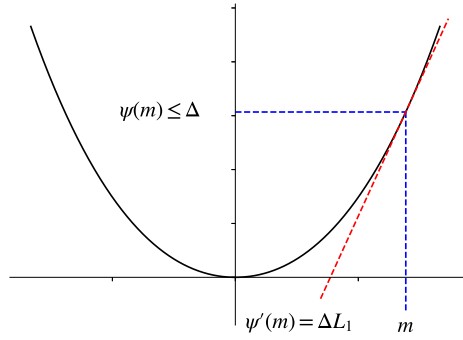
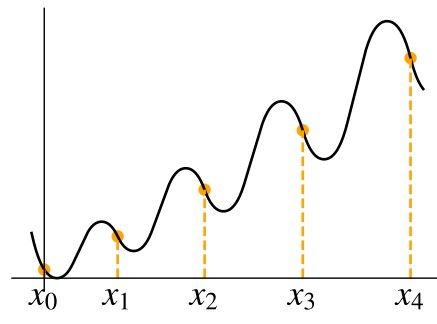

(a) Graph of $\psi(x) = \frac{L_0}{L_1^2} \left( \exp(L_1|x|) - L_1|x| - 1 \right)$.

(b) Trajectory from Lemma 1.

Figure 1: Objectives from Lemma 1. $m := (\psi')^{-1}(\Delta L_1) = \frac{1}{L_1} \log \left( 1 + \frac{\Delta L_1^2}{L_0} \right)$.

**Lemma 1.** *Suppose that $\Delta L_1^2 \geq L_0$, $\eta \geq \frac{1}{L_1}$, and $\gamma \leq \tilde{\mathcal{O}}(\eta \Delta L_1^2)$. Then there exists a problem instance $(f, g, \mathcal{D}) \in \mathcal{F}_{as}(\Delta, L_0, L_1, 0)$ such that $\|\nabla f(\boldsymbol{x}_t)\| \geq \Delta L_1$ for all $t \geq 0$.*

The trajectory of the algorithm analyzed in Lemma 1 is informally pictured in Figure 1b. The objective function is a piecewise combination of copies of the function $\psi(x) = \frac{L_0}{L_1^2} \left( \exp(L_1|x|) - L_1|x| - 1 \right)$, which is shown in Figure 1a. $\psi$ is constructed to satisfy $|\psi''(x)| = L_0 + L_1|\psi'(x)|$ for all $x$, so it grows as quickly as possible under $(L_0, L_1)$-smoothness. As shown in Figure 1b, at each step the algorithm receives a gradient $g_t$ and "jumps" over a valley to reach a new point with gradient $g_{t+1}$. In order to achieve this jump, we need the sequence of gradients to satisfy

$$\frac{\eta |g_t|}{\sqrt{\gamma^2 + \sum_{i=0}^{t-1} g_i^2}} \geq \frac{4}{L_1} \log \left( 1 + \frac{L_1 |g_{t+1}|}{L_0} \right).$$

This recurrent inequality is tricky since the magnitude of each $|g_t|$ is constrained not just by the history $\{|g_i|\}_{i<t}$, but also by the future $g_{t+1}$. We show that this requirement is satisfied if $g_t = \Theta \left( \left( (t+1) \log(1 + \Delta L_1^2 L_0^{-1}(t+1)) \right)^t \Delta L_1 \right)$, and that this sequence of gradients can be realized by an objective function in $\mathcal{F}_{as}(\Delta, L_0, L_1, \sigma)$.

**Slow convergence when $\eta \leq 1/L_1$** In this case, previous lower bounds (Zhang et al., 2020b; Crawshaw et al., 2022) consider a one-dimensional function with deterministic gradients to show a complexity of $\Omega \left( \Delta^2 L_1^2 \epsilon^{-2} \right)$. To achieve $\epsilon^{-4}$ dependence, we consider a high-dimensional function with stochastic gradients (adapted from Drori & Shamir (2020)), for which the first partial derivative $\nabla_1 f$ has magnitude $\epsilon$, and the stochastic gradient noise affects coordinates with index greater than 1. Since the same learning rate is shared by all coordinates, the noise in later coordinates will decrease the learning rate for the first coordinate. Combining with $\eta \leq 1/L_1$ leads to the desired complexity.

**Lemma 2.** *Let $T = \Theta \left( \Delta^2 L_1^2 \sigma^2 \epsilon^{-4} + \Delta L_0 \sigma^2 \epsilon^{-4} + \Delta^2 L_1^2 \epsilon^{-2} \right)$, and suppose $d \geq T$ and $\epsilon \leq \mathcal{O} \left( \min \left\{ \sqrt{\Delta L_0}, \sqrt{\Delta L_1 \gamma} \right\} \right)$. If $\eta \leq \frac{1}{L_1}$, then there exists some $(f, g, \mathcal{D}) \in \mathcal{F}_{as}(\Delta, L_0, L_1, \sigma)$ such that $\|\nabla f(\boldsymbol{x}_t)\| = \epsilon$ for all $0 \leq t \leq T - 1$.*

See Appendix A for details on the objective function, stochastic gradient oracle, and analysis of the trajectory. The theorem is then proved by combining Lemmas 1 and 2: No matter the choice of the parameter $\eta$, the algorithm will not find an $\epsilon$-stationary point within the first $T$ steps.

## 5 ADAGRAD AND DECORRELATED ADAGRAD

Here we present lower bounds for Decorrelated AdaGrad (Theorem 2) and the original AdaGrad (Theorem 3). Both lower bounds are quadratic in $\Delta, L_0$, but our result for Decorrelated AdaGrad has a stronger dependence on $\sigma$ than that of AdaGrad. This discrepancy is further discussed below.

**Theorem 2.** *Denote $\mathcal{F} = \mathcal{F}_{as}(\Delta, L_0, L_1, \sigma)$, and let $A_{DA}$ denote Decorrelated AdaGrad (Equation 2) with parameters $\eta, \gamma > 0$. Suppose $0 < \epsilon < \tilde{\mathcal{O}}\left(\min\left\{\Delta L_1, \sqrt{\Delta L_1 \sigma}\right\}\right)$. Then*

$$\mathcal{T}(A_{DA}, \mathcal{F}, \epsilon) \geq \Omega\left(\frac{\Delta^2 L_0^2 \sigma^2}{\gamma^2 \epsilon^4} + \frac{\Delta^2 L_1^2 \sigma^2}{\gamma^2 \epsilon^2 \log^2\left(1 + \frac{\Delta L_1^2}{L_0}\right)}\right).$$

**Theorem 3.** *Denote $\mathcal{F} = \mathcal{F}_{as}(\Delta, L_0, L_1, \sigma)$, and let $A_{ada}$ denote AdaGrad (Equation 3) with parameters $\eta, \gamma > 0$. Suppose $0 < \epsilon < \tilde{\mathcal{O}}\left(\min\left\{\Delta L_1, \sqrt{\Delta L_1 \sigma}\right\}\right)$. If $\gamma \leq \sigma$, then*

$$\mathcal{T}(A_{ada}, \mathcal{F}, \epsilon) \geq \Omega\left(\frac{\Delta^2 L_0^2}{\epsilon^4} + \frac{\Delta^2 L_1^2}{\epsilon^2 \log^2\left(1 + \frac{\Delta L_1^2}{L_0}\right)}\right).$$

The proofs of both theorems above are given in Appendix B. The results above exhibit several important properties. **(1)** As in Theorem 1, the dominating term $\Delta^2 L_0^2 \sigma^2 \gamma^{-2} \epsilon^{-4}$ of the lower bound in Theorem 2 is greater than the optimal complexity $\Delta L_0 \sigma^2 \epsilon^{-4}$ of the smooth case (up to the choice of $\gamma$, which is usually a small constant). In fact, the dominating term is quadratic in $\Delta$ and $L_0$, so **the complexity of Decorrelated AdaGrad in this setting is fundamentally larger than the optimal complexity of the smooth counterpart**. Unlike Theorem 1, $L_1$ does not appear in the dominating term of this bound. **(2)** Compared to Decorrelated AdaGrad, our bound for AdaGrad loses a factor of $\sigma^2/\gamma^2$. This arises in our construction from the fact that, at step $t$, any noise present in $g_t$ will also appear in the denominator of the update, so that the update size of AdaGrad is not as sensitive to noise as the decorrelated counterpart. Still, in regimes where $\Delta, L_0$ are large compared to $\sigma$, our complexity lower bound of $\tilde{\Omega}(\Delta^2 L_0^2 \epsilon^{-4})$ is larger than the optimal complexity of the smooth case. This shows that **AdaGrad cannot recover the optimal complexity of the smooth case in all relaxed smooth regimes**. **(3)** The lower bound of Theorem 2 diverges to $\infty$ when the $\gamma$ goes to 0, which confirms the conventional wisdom that a non-zero stabilization constant is necessary in practice.

## 5.1 PROOF OUTLINE

The structure of the proof is similar to Theorem 1 (outlined in Section 4.1), but we can achieve divergence for Decorrelated AdaGrad under the weaker condition $\eta \geq \Theta\left(\min\left\{\gamma/(L_1\sigma), \gamma\epsilon/(L_0\sigma)\right\}\right)$ using a novel high-dimensional construction that takes advantage of the coordinate-wise learning rates of Decorrelated AdaGrad by injecting noise into one coordinate per timestep. When $\eta$ is smaller than this threshold, convergence is slow for a one-dimensional, linear function with slope $\epsilon$.

**Divergence when** $\eta \geq \Theta\left(\min\left\{\gamma/(L_1\sigma), \gamma\epsilon/(L_0\sigma)\right\}\right)$ For any $d \geq 1$, we consider the objective function $f(\boldsymbol{x}) = \sum_{i=1}^{d} \psi(\langle\boldsymbol{x}, \mathbf{e}_i\rangle)$, where $\psi$ is as defined in Section 4.1. Letting $m = (\psi')^{-1}(\epsilon) = \frac{1}{L_1}\log\left(1 + \frac{L_1\epsilon}{L_0}\right)$, consider the initialization $\boldsymbol{x}_0 = m\mathbf{e}_1$, which by construction satisfies $\|\nabla f(\boldsymbol{x}_0)\| = \epsilon$. For the initialization, all of the coordinates besides the first one are already at their optimal values, and the partial gradient for these coordinates is zero; the stochastic gradient injects noise into the second coordinate, so that $\boldsymbol{g}_0 = \nabla f(\boldsymbol{x}_0) \pm \sigma\mathbf{e}_2 = \epsilon\mathbf{e}_1 \pm \sigma\mathbf{e}_2$. Based on the magnitude of $\eta$, this guarantees $|\langle\boldsymbol{x}_1, \mathbf{e}_2\rangle| \geq m$, and consequently $|\nabla f(\boldsymbol{x}_1)| \geq |\psi'(\langle\boldsymbol{x}_1, \mathbf{e}_2\rangle)| \geq \psi'(m) = \epsilon$. Intuitively, the size of $\eta$ causes the second coordinate to "jump" from the minimum at 0 to another point whose partial derivative is larger than $\epsilon$. This process continues with $t$: at each step $t$, the stochastic gradient noise affects the coordinate indexed $(t+2)$, so that this coordinate of the iterate jumps from 0 to a point with magnitude at least $m$. This guarantees that the algorithm does not reach a stationary point for $d$ steps, and $d$ can be arbitrarily large. An important detail of this process is that the coordinate-wise learning rates ensure that the length of each "jump" (i.e., the per-coordinate update size) does not decrease with $t$. This argument is made formal in the following lemma.

**Lemma 3.** *Let $0 < \epsilon < \mathcal{O}(\Delta L_1)$. If $\eta \geq \Omega\left(\frac{\gamma}{L_1\sigma}\log\left(1 + \frac{L_1\epsilon}{L_0}\right)\right)$, then for any $T \geq 1$, there exists some $f \in \mathcal{F}$ such that Decorrelated AdaGrad satisfies $\|\nabla f(\boldsymbol{x}_t)\| \geq \epsilon$ for all $0 \leq t \leq T - 1$.*

*Similarly, if $\eta \geq \Omega\left(\frac{1}{L_1}\log\left(1 + \frac{L_1\epsilon}{L_0}\right)\right)$ and $\gamma \leq \sigma$, then for any $T \geq 1$ there exists some $f \in \mathcal{F}$ such that AdaGrad satisfies $\|\nabla f(\boldsymbol{x}_t)\| \geq \epsilon$ for all $0 \leq t \leq T - 1$.*

Notice that the $\eta$ requirement for AdaGrad in Lemma 3 is stronger than that of Decorrelated AdaGrad. As previously mentioned, this happens because the size of AdaGrad's update is less sensitive to stochastic gradient noise than Decorrelated AdaGrad.

**Slow convergence when** $\eta \leq \Theta\left(\min\left\{\gamma/(L_1\sigma), \gamma\epsilon/(L_0\sigma)\right\}\right)$ In this case, the desired complexity follows by analyzing the trajectory of each algorithm on a one-dimensional, linear function with slope equal to $\epsilon$, similarly to existing lower bounds (Zhang et al., 2020b; Crawshaw et al., 2022).

**Lemma 4.** *Let* $0 < \epsilon < \mathcal{O}\left(\min\left\{\Delta L_1, \sqrt{\Delta L_1}\sigma\right\}\right)$. *If* $\eta \leq \frac{\sqrt{2}\gamma}{L_1\sigma}\log\left(1 + \frac{L_1\epsilon}{L_0}\right)$, *then there exists some* $(f, g, \mathcal{D}) \in \mathcal{F}_{as}(\Delta, L_0, L_1, \sigma)$ *such that Decorrelated AdaGrad satisfies* $\|\nabla f(\boldsymbol{x}_t)\| \geq \epsilon$ *for all* $t \leq \tilde{\mathcal{O}}\left(\Delta^2 L_0^2\sigma^2\gamma^{-2}\epsilon^{-4} + \Delta^2 L_1^2\sigma^2\gamma^{-2}\epsilon^{-2}\right)$.

*Similarly, if* $\eta \leq \mathcal{O}\left(\frac{1}{L_1}\log\left(1 + \frac{L_1\epsilon}{L_0}\right)\right)$, *then there exists some* $(f, g, \mathcal{D}) \in \mathcal{F}_{as}(\Delta, L_0, L_1, \sigma)$ *such that AdaGrad satisfies* $\|\nabla f(\boldsymbol{x}_t)\| \geq \epsilon$ *for all* $t \leq \mathcal{O}\left(\Delta^2 L_0^2\epsilon^{-4} + \Delta^2 L_1^2\epsilon^{-2}\right)$.

Theorems 2 and 3 can then be proven by combining Lemmas 3 and 4.

## 6 SINGLE-STEP ADAPTIVE SGD

In this section, we consider single-step adaptive SGD (Equation 4). Our lower bound shows that, due to relaxed smoothness and affine noise, any algorithm of this type will incur a higher-order dependence on $\Delta, L_1$. The results below are stated in terms of constants $\gamma_i$ and a function $\zeta$, which are defined in terms of $\delta$ and $\sigma_2$ (see Appendix C.1 for the definitions). In the discussion following the theorem statement, we specify the limiting behavior of these constants in terms of $\sigma_2, \delta$.

**Theorem 4.** *Denote* $G = \tilde{\Theta}(\Delta L_1)$ *and suppose* $G \geq \sigma_1$. *Let* $0 < \epsilon \leq \min\left\{\sigma_1, \frac{G}{2}, \frac{G-\sigma_1}{\sigma_2-1}, \frac{\sqrt{\Delta L_0}}{\sqrt{2}}\right\}$. *Let algorithm* $A_{single}$ *denote single-step adaptive SGD (Equation 4) with any step size function* $\alpha : \mathbb{R}^d \to \mathbb{R}$ *for a sufficiently large d, and let* $\mathcal{F} = \mathcal{F}_{\text{aff}}(\Delta, L_0, L_1, \sigma_1, \sigma_2)$. *If* $\sigma_2 \geq 3$, *then*

$$\mathcal{T}(A_{single}, \mathcal{F}, \epsilon, \delta) \geq \tilde{\Omega}\left(\frac{\Delta L_0\sigma_1^2}{\epsilon^4} + \frac{(\Delta L_1)^{2-\gamma_2-\gamma_3}\sigma_1^{\gamma_2+\gamma_3-\gamma_1}}{\epsilon^{2-\gamma_1}}\right).$$

*Otherwise, if* $1 < \sigma_2 < 3$, *then*

$$\mathcal{T}(A_{single}, \mathcal{F}, \epsilon, \delta) \geq \tilde{\Omega}\left(\frac{\Delta L_0\sigma_1^2}{\epsilon^4} + \frac{(\Delta L_1)^{2-\gamma_5-\gamma_6}}{\epsilon^{2-\gamma_4}}(\sigma_2-1)^2\left(\frac{\sigma_1}{\sigma_2-1}\right)^{\gamma_5+\gamma_6-\gamma_4}\right).$$

The proof is given in Appendix C. Below, we specify the error terms $\gamma_i$ in two regimes of $\sigma_2$.

**Large** $\sigma_2$ For $\sigma_2 > 3$, the error terms $\gamma_1, \gamma_2, \gamma_3$ satisfy: $\gamma_1, \gamma_3 = \Theta\left(\log\left(1 + \zeta(2/3, \delta)\right)\right)$ and $\gamma_2 = \Theta\left(\sigma_2^{-1}\right)$, where $\zeta(p, \delta)$ is defined in Equation 25. Lemma 15 shows that $\lim_{\delta\to 0}\zeta(p, \delta) = 0$ for all $p \in (0, 1)$, so when $\delta \to 0$, the lower bound approaches

$$\Omega\left(\frac{\Delta L_0\sigma^2}{\epsilon^4} + \frac{\Delta^2 L_1^2}{\epsilon^2\log\left(1 + \frac{\Delta L_1^2}{L_0}\right)}\left(\frac{\sigma_1}{\Delta L_1}\right)^{\Theta(1/\sigma_2)}\right).$$

In this limiting case, the complexity has a nearly quadratic dependence on $\Delta, L_1$, but only in the non-dominating term. Still, we emphasize the generality of our result, which applies for any adaptive SGD algorithm whose learning rate only depends on the current gradient, and shows that adaptivity based on the current gradient alone will incur higher-order dependencies on $\Delta, L_1$.

**Small** $\sigma_2$ Existing lower bounds that utilize similar constructions of a biased random walk under affine noise (Faw et al., 2023; Crawshaw et al., 2023b) require that $\sigma_2$ be bounded away from 1. Our Theorem 4 covers the case that $\sigma_2 \to 1$, albeit with a lower bound that approaches 0 when $\sigma_2 \to 1$. The error terms $\gamma_4, \gamma_5, \gamma_6$ depend only on $\delta, \sigma_2$ and satisfy: $\lim_{\delta\to 0}\gamma_4 = 0$, $\lim_{\sigma_2\to 1}\gamma_5 = 1$, $\lim_{\sigma_2\to 1}\gamma_6 = 0$, $\lim_{\delta\to 0}\gamma_6 = 0$. Note that $\gamma_5$ does not depend on $\delta$, and $\gamma_5 < 1$. Therefore, letting $\delta \to 0$ yields a lower bound of

$$\Omega\left(\frac{\Delta L_0\sigma^2}{\epsilon^4} + \frac{\Delta L_1\sigma_1(\sigma_2-1)}{\epsilon^2\log\left(1 + \frac{\Delta L_1^2}{L_0}\right)}\left(\frac{\Delta L_1(\sigma_2-1)}{\sigma_1}\right)^{1-\gamma_5}\right).$$

Since $1 - \gamma_5 \to 0$ as $\sigma_2 \to 1$, the second term in the lower bound goes to $0$ when $\sigma_2 \to 1$. This shows that our construction relies on $\sigma_2$ bounded away from $1$, and raises the question whether a single-step adaptive SGD algorithm can converge without a quadratic dependence on $\Delta, L_1$ in the regime $\sigma_2 < 1$.

## 6.1 Proof Outline

The full proof of Theorem 4 can be found in Appendix C, and we provide a sketch of the main ideas here. The proof of Theorem 4 has three main steps:

**Step 1** If the step size function $\alpha$ does not satisfy $0 \leq \alpha(\boldsymbol{g}) \leq \tilde{\mathcal{O}}\left(1/(L_1\|\boldsymbol{g}\|)\right)$ for every $\boldsymbol{g}$ with $\|\boldsymbol{g}\| \in [\epsilon, \sigma_1 + (\sigma_2 + 1)\Delta L_1]$, then $A_{\text{single}}$ will diverge for some exponential $f_{\text{exp}}$, proven in Lemma 9. The difficult objective is constructed similarly as in Lemma 1, pictured in Figure 1b.

**Step 2** If there exist $\boldsymbol{g}_1, \boldsymbol{g}_2 \in \mathbb{R}^d$ such that $\boldsymbol{g}_2 = c\boldsymbol{g}_1$ for some $c < 0$, and $\|\boldsymbol{g}_1\| \leq \mathcal{O}(\|\boldsymbol{g}_2\|)$, but $\alpha(\boldsymbol{g}_1)\|\boldsymbol{g}_1\| \geq \Omega(\alpha(\boldsymbol{g}_2)\|\boldsymbol{g}_2\|)$ (i.e., a "tricky pair", see Definition 1 in Appendix C.1), then there is some $(f, g, \mathcal{D}) \in \mathcal{F}$ for which $A_{\text{single}}$ will diverge. The construction is based on the idea that $\boldsymbol{g}_1$ and $\boldsymbol{g}_2$ are stochastic gradients at a given point, where $\boldsymbol{g}_2$ points towards the minimum and $\boldsymbol{g}_1$ points away from the minimum, but $\alpha(\boldsymbol{g}_1)$ is close enough to $\alpha(\boldsymbol{g}_2)$ that $A_{\text{single}}$ has nearly equal expected movement in each direction. In this case, the sequence $\{\boldsymbol{x}_t\}$ follows a biased random walk that will diverge with probability at least $\delta$. This argument is made formal in Lemma 10.

**Step 3** If neither of the above cases hold, then $\alpha(\boldsymbol{g}) \leq \tilde{\mathcal{O}}\left(1/(L_1\|\boldsymbol{g}\|)\right)$ and there do not exist any tricky pairs. The non-existence of tricky pairs means that $\alpha(\boldsymbol{g})\|\boldsymbol{g}\|$ grows sufficiently fast in terms of $\|\boldsymbol{g}\|$ when $\|\boldsymbol{g}\| \in [\epsilon, \sigma_1 + (\sigma_2 + 1)\Delta L_1]$. In order for $\alpha$ to respect $\alpha(\boldsymbol{g}) \leq \tilde{\mathcal{O}}\left(1/(L_1\|\boldsymbol{g}\|)\right)$ while also growing quickly, it must be that $\alpha(\boldsymbol{g})$ is small whenever $\|\boldsymbol{g}\|$ is small. Lemma 11 formalizes this idea to show an upper bound for $\alpha(\boldsymbol{g})$ whenever $\|\boldsymbol{g}\| = \epsilon$. The final bound follows by analyzing the trajectory of $A_{\text{single}}$ for a piecewise linear objective with gradient $\boldsymbol{g}$ satisfying $\|\boldsymbol{g}\| = \epsilon$, since the convergence rate is inversely proportional to $\alpha(\boldsymbol{g})$. (Lemma 12)

## 7 Discussion and Conclusion

It has been stated in the optimization literature (Woodworth et al., 2018; 2021) that a complexity lower bound should not be interpreted as an unquestionable limit of performance, but rather as a tool to examine the assumptions that led to the bound and explore alternatives. In this spirit, we consider the implications of our choice of problem formulation.

First, the optimization problem investigated in this paper (i.e., problem instances satisfying Assumptions 1 and 2) may not be a sufficient theoretical framework to explain the behavior of adaptive optimization algorithms in deep learning. A complete explanation of this type may require additional assumptions about the structure of the objective functions, such as enforcing a neural network architecture or a particular data distribution.

Also, the negative results presented in this paper may be bypassed by algorithms other than those we have considered. In particular, it is possible that higher-order polynomial dependence on problem parameters can be avoided by more practical algorithms such as Adam and AdamW. Presently, it is unknown whether these algorithms can recover the optimal complexity of the smooth case (as does SGD with clipping), or if they behave more like the AdaGrad variants considered in this paper.

**Limitations** The most important limitation of our work is that our strongest lower bounds (Theorem 1, 2) are obtained for decorrelated variants of AdaGrad, which are not commonly used in practice. We view these decorrelated methods as a starting point for lower bounds of adaptive algorithms under relaxed smoothness, similarly to early work Li & Orabona (2019) showing upper bounds for decorrelated versions of adaptive algorithms. Also, our result for the original AdaGrad (Theorem 3) is weaker in terms of the dependence on $\sigma$. It remains open whether this result can be improved, and whether there is a fundamental difference in the complexity of AdaGrad compared to the decorrelated variants. Further, even if the same complexity can be achieved by the original AdaGrad, the existing upper bounds do not exactly match our lower bounds. Therefore, it remains to exactly characterize the complexity of AdaGrad (and its variants) by providing matching upper and lower bounds.

ACKNOWLEDGMENTS

We would like to thank the anonymous reviewers for their helpful comments. This work is supported by the Institute for Digital Innovation fellowship, a ORIEI seed funding, an IDIA P3 fellowship from George Mason University, a Cisco Faculty Research Award, and NSF award #2436217, #2425687.

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

## A  PROOF OF THEOREM 1

**Lemma 5** (Restatement of Lemma 1). *Suppose that $\Delta L_1^2 \geq L_0$, and $\eta \geq \frac{1}{L_1}$, and*

$$\gamma \leq \frac{\eta \Delta L_1^2}{8 \log\left(1 + 48 \frac{\Delta L_1^2}{L_0}\right)}. \tag{6}$$

*Then there exists a problem instance $(f, g, \mathcal{D}) \in \mathcal{F}_{as}(\Delta, L_0, L_1, 0)$ such that $\|\nabla f(\boldsymbol{x}_t)\| \geq \Delta L_1$ for all $t \geq 0$.*

*Proof.* The difficult function $f$ will be piecewise linear and exponential, and constructed in such a way that the gradients $\nabla f(\boldsymbol{x}_t)$ increase at a rate of $\approx t^t$. The rapid growth rate of the gradients ensures, even with the update normalization, that the update size $\|\boldsymbol{x}_{t+1} - \boldsymbol{x}_t\|$ increases at every step.

Recall the function $\psi : \mathbb{R} \to \mathbb{R}$ defined as

$$\psi(x) = \frac{L_0}{L_1^2}\left(\exp(L_1|x|) - L_1|x| - 1\right),$$

with

$$\psi'(x) = \operatorname{sign}(x)\frac{L_0}{L_1}\left(\exp(L_1|x|) - 1\right).$$

It is straightforward to verify that $\psi$ bounded from below by $0$, continuously differentiable, and $(L_0, L_1)$-smooth.

The difficult function $f$ will be constructed in terms of the following:

$$g_t = \left(576(t+1)\log\left(1 + \frac{\Delta L_1^2}{L_0}(t+1)\right)\right)^t \Delta L_1$$

$$m_t = \frac{1}{L_1}\log\left(1 + \frac{L_1 g_t}{L_0}\right)$$

$$\ell_t = \frac{\eta g_t}{\sqrt{\gamma^2 + \sum_{i=0}^{t-1} g_i^2}}$$

$$d_t = \sum_{i=0}^{t-1} \ell_i.$$

For each $t \geq 0$, define $\phi_t : \mathbb{R} \to \mathbb{R}$ as:

$$\phi_t(x) = \begin{cases} \psi(x - m_t) & x \leq m_t + m_{t+1} \\ g_t(x - m_t - m_{t+1}) + \psi(m_{t+1}) & x \in (m_t + m_{t+1}, \ell_t - 2m_{t+1}) \\ -\psi(x - \ell_t + m_{t+1}) + 2\psi(m_{t+1}) + g_t(\ell_t - 3m_{t+1} - m_t) & x \geq \ell_t - 2m_{t+1} \end{cases}.$$

These functions are constructed to satisfy $\phi_t'(0) = \psi'(-m_t) = -g_t$ and $\phi_t'(\ell_t) = \psi'(-m_{t+1}) = -g_{t+1}$. To see that this definition makes sense, we should show that $\ell_t - 2m_{t+1} \geq m_t + m_{t+1}$, so that the boundary of the first piece is smaller than the boundary of the third piece. This is equivalent to: $\ell_t \geq m_t + 3m_{t+1}$. Using $\Delta L_1^2 \geq L_0$, the sequence $g_t$ is increasing, and consequently so is $m_t$. Therefore it suffices to prove

$$\ell_t \geq 4m_{t+1}$$

$$\frac{\eta g_t}{\sqrt{\gamma^2 + \sum_{i=0}^{t-1} g_i^2}} \geq \frac{4}{L_1}\log\left(1 + \frac{L_1 g_{t+1}}{L_0}\right), \tag{7}$$

We will prove Equation 7 separately for the cases $t = 0$ and $t \geq 1$.

**Case 1** $t = 0$. In this case, the desired condition is

$$\frac{\eta g_0}{\gamma} \geq \frac{4}{L_1} \log\left(1 + \frac{L_1 g_1}{L_0}\right).$$

The RHS of the above inequality can be bounded as

$$\frac{4}{L_1} \log\left(1 + \frac{L_1 g_1}{L_0}\right) = 4\log\left(1 + \frac{L_1}{L_0}\left(1152 \log\left(1 + 2\frac{\Delta L_1^2}{L_0}\right)\right)\Delta L_1\right)$$

$$= \frac{4}{L_1} \log\left(1 + 1152\frac{\Delta L_1^2}{L_0} \log\left(1 + 2\frac{\Delta L_1^2}{L_0}\right)\right)$$

$$\overset{(i)}{\leq} \frac{4}{L_1} \log\left(1 + 2304\left(\frac{\Delta L_1^2}{L_0}\right)^2\right)$$

$$\overset{(ii)}{\leq} \frac{8}{L_1} \log\left(1 + 48\frac{\Delta L_1^2}{L_0}\right)$$

$$\overset{(iii)}{\leq} \frac{\Delta L_1 \eta}{\gamma},$$

where $(i)$ uses $\log x \leq 1 + x$ for all $x > 0$, $(ii)$ uses $\log(1 + x^n) \leq \log((1 + x)^n) \leq n \log(1 + x)$ for all $x > 0$, and $(iii)$ uses the assumed condition on $\gamma$ (Equation 6). This concludes the first case.

**Case 2** $t \geq 1$. We first simplify the denominator $\sqrt{\gamma^2 + \sum_{i=0}^{t-1} g_i^2}$. First, the assumed condition on $\gamma$ (Equation 6) implies that $\gamma \leq \eta \Delta L_1^2 = \eta L_1 g_0$, so

$$\gamma^2 + \sum_{i=0}^{t-1} g_i^2 \leq \eta^2 L_1^2 g_0^2 + \sum_{i=0}^{t-1} g_i^2 \leq (1 + \eta^2 L_1^2) \sum_{i=0}^{t-1} g_i^2 \leq 2\eta^2 L_1^2 \sum_{i=1}^{t-1} g_i^2.$$

Also,

$$\sum_{i=0}^{t-2} g_i^2 \leq (t-1)g_{t-2}^2$$

$$= (t-1)\left(576(t-1)\log\left(1 + \frac{\Delta L_1^2}{L_0}(t-1)\right)\right)^{2(t-2)}\Delta^2 L_1^2$$

$$\leq (t-1)\left(576t\log\left(1 + \frac{\Delta L_1^2}{L_0}t\right)\right)^{2(t-2)}\Delta^2 L_1^2$$

$$\overset{(i)}{\leq} \left(576t\log\left(1 + \frac{\Delta L_1^2}{L_0}t\right)\right)^{2(t-1)}\Delta^2 L_1^2$$

$$= g_{t-1}^2.$$

Therefore $\gamma^2 + \sum_{i=0}^{t-1} g_i^2 \leq 4\eta^2 L_1^2 g_{t-1}^2$. So the LHS of Equation 7 can be bounded as

$$\frac{\eta g_t}{\sqrt{\gamma^2 + \sum_{i=0}^{t-1} g_i^2}} \geq \frac{g_t}{2L_1 g_{t-1}}$$

$$= \frac{\left(576(t+1)\log\left(1 + \frac{\Delta L_1^2}{L_0}(t+1)\right)\right)^t \Delta L_1}{2L_1\left(576t\log\left(1 + \frac{\Delta L_1^2}{L_0}t\right)\right)^{t-1}\Delta L_1}$$

$$= \frac{288(t+1)}{L_1} \log\left(1 + \frac{\Delta L_1^2}{L_0}(t+1)\right)\left(\frac{576(t+1)\log\left(1 + \frac{\Delta L_1^2}{L_0}(t+1)\right)}{576t\log\left(1 + \frac{\Delta L_1^2}{L_0}t\right)}\right)^{t-1}$$

$$\geq \frac{288(t+1)}{L_1} \log\left(1 + \frac{\Delta L_1^2}{L_0}(t+1)\right). \tag{8}$$

The RHS of Equation 7 can be bounded as

$$\frac{4}{L_1}\log\left(1+\frac{L_1 g_{t+1}}{L_0}\right) = \frac{4}{L_1}\log\left(1+\frac{\Delta L_1^2}{L_0}\left(576(t+2)\log\left(1+\frac{\Delta L_1^2}{L_0}(t+2)\right)\right)^{t+1}\right)$$

$$\overset{(i)}{\leq} \frac{4(t+1)}{L_1}\log\left(1+576\left(\frac{\Delta L_1^2}{L_0}\right)^{1/(t+1)}(t+2)\log\left(1+\frac{\Delta L_1^2}{L_0}(t+2)\right)\right)$$

$$\overset{(ii)}{\leq} \frac{4(t+1)}{L_1}\log\left(1+576\left(\frac{\Delta L_1^2}{L_0}\right)^{1+1/(t+1)}(t+2)^2\right)$$

$$\overset{(iii)}{\leq} \frac{4(t+1)}{L_1}\log\left(1+576\left(\frac{\Delta L_1^2}{L_0}\right)^2(t+2)^2\right)$$

$$\overset{(iv)}{\leq} \frac{8(t+1)}{L_1}\log\left(1+24\frac{\Delta L_1^2}{L_0}(t+2)\right)$$

$$\overset{(v)}{\leq} \frac{8(t+1)}{L_1}\log\left(1+36\frac{\Delta L_1^2}{L_0}(t+1)\right),$$

where $(i)$ uses $\log(1+x^n) \leq \log((1+x)^n) = n\log(1+x)$ for $x > 0$, $(ii)$ uses $\log(1+x) \leq x$ for all $x > 0$, $(iii)$ uses $\Delta L_1^2 \geq L_0$, $(iv)$ again uses $\log(1+x^n) \leq n\log(1+x)$, and $(v)$ uses $t + 2 \leq \frac{3}{2}(t+1)$ since $t \geq 1$. Further,

$$\frac{4}{L_1}\log\left(1+\frac{L_1 g_{t+1}}{L_0}\right) \overset{(i)}{\leq} \frac{8(t+1)}{L_1}\left(\log\left(1+\frac{\Delta L_1^2}{L_0}(t+1)\right)+\log(37)\right)$$

$$= \frac{8(t+1)}{L_1}\log\left(1+\frac{\Delta L_1^2}{L_0}(t+1)\right)+8\log(37)(t+1)$$

$$\overset{(ii)}{\leq} \frac{8(1+\log(37))(t+1)}{L_1}\log\left(1+\frac{\Delta L_1^2}{L_0}(t+1)\right)$$

$$\leq \frac{288(t+1)}{L_1}\log\left(1+\frac{\Delta L_1^2}{L_0}(t+1)\right), \tag{9}$$

where $(i)$ uses $\log(1+ab) \leq \log((1+a)(1+b)) \leq \log(1+a)+\log(1+b)$ for all $a,b > 0$, and $(ii)$ uses $\Delta L_1^2 \geq L_0$ and $t \geq 1$. Combining Equation 8 and Equation 9 proves Equation 7.

This proves that the definition of $\phi_t$ makes sense for all $t$. We can finally define the difficult objective $f$ as follows:

$$f(x) = \phi_{j(x)}(x - d_{j(x)}) + \sum_{i=0}^{j(x)-1}\phi_i(\ell_i),$$

where

$$j(x) = \begin{cases}\max\{t \geq 0 \mid d_t \leq x\} & x \geq 0 \\ 0 & x < 0\end{cases}.$$

With this definition, $f$ is essentially a piece-wise function, where each piece is an interval $[d_t, d_{t+1}]$ whose function value is a translation of $\phi_t$. $f$ is informally pictured in Figure 1b of the main text. Notice that $f$ is continuous and differentiable within each piece. At the boundary of each piece,

$$\lim_{x \to d_{t+1}^-} f(x) = \lim_{x \to d_{t+1}^-} \phi_t(x - d_t) + \sum_{i=0}^{t-1}\phi_i(\ell_i)$$

$$= \phi_t(d_{t+1} - d_t) + \sum_{i=0}^{t-1}\phi_i(\ell_i) = \phi_t(\ell_t) + \sum_{i=0}^{t-1}\phi_i(\ell_i) = \sum_{i=0}^{t}\phi_i(\ell_i),$$

$$\lim_{x \to d_{t+1}^+} f(x) = \lim_{x \to d_{t+1}^+} \phi_{t+1}(x - d_{t+1}) + \sum_{i=0}^{t}\phi_i(\ell_i) = \phi_{t+1}(0) + \sum_{i=0}^{t}\phi_i(\ell_i) = \sum_{i=0}^{t}\phi_i(\ell_i).$$

Also

$$\lim_{x \to d_{t+1}^-} f'(x) = \lim_{x \to d_{t+1}^-} \phi_t'(x - d_t) = \phi_t'(d_{t+1} - d_t) = \phi_t'(\ell_t) = -g_{t+1}$$

$$\lim_{x \to d_{t+1}^+} f'(x) = \lim_{x \to d_{t+1}^+} \phi_{t+1}'(x - d_{t+1}) = \phi_{t+1}'(0) = -g_{t+1}.$$

Therefore $f$ is differentiable everywhere. Also,

$$\inf_x f(x) = \inf_{t \geq 0} \left\{ \inf_{x \in [0, \ell_t]} \phi_t(x) + \sum_{i=0}^{t-1} \phi_i(\ell_i) \right\} \geq \inf_{t \geq 0} \inf_{x \in [0, \ell_t]} \phi_t(x) = 0.$$

The initial point $x_0 = 0$ satisfies

$$f(x_0) = \phi_0(0) = \psi(-m_0) = \frac{g_0}{L_1} - \frac{L_0}{L_1^2} \log\left(1 + \frac{L_1 g_0}{L_0}\right) \leq \frac{g_0}{L_1} = \Delta.$$

Therefore $f(x_0) - \inf_x f(x) \leq \Delta$. Since each $\phi_t$ is $(L_0, L_1)$-smooth, so is $f$.

We will use a stochastic gradient $g, \mathcal{D}$ for this function which is always equal to the true gradient, so that the noise conditions are trivially satisfied. Therefore $(f, g, \mathcal{D}) \in \mathcal{F}_{as}(\Delta, L_0, L_1, 0)$.

Now, consider the trajectory when starting from the initial point $x_0 = 0$. We claim that $x_t = d_t$ for all $t \geq 0$, which we will prove by induction. The base case $t = 0$ holds by construction. So suppose that $x_i = d_i$ for all $0 \leq i \leq t$. Then $f'(x_i) = f'(d_i) = -g_i$ for all $i$. So

$$x_{t+1} = x_t - \frac{\eta f'(x_t)}{\sqrt{\gamma^2 + \sum_{i=0}^{t-1} (f'(x_i))^2}} = d_t + \frac{\eta g_t}{\sqrt{\gamma^2 + \sum_{i=0}^{t-1} g_i^2}} = d_t + \ell_t = d_{t+1}.$$

This completes the induction.

Therefore, for all $t \geq 0$, we have $|f'(x_t)| = g_t \geq g_0 = \Delta L_1$. $\qquad \square$

The following lemma uses a difficult objective which is adapted from Theorem 2 of Drori & Shamir (2020).

**Lemma 6** (Restatement of Lemma 2). *Let*

$$T = 1 + \frac{\Delta^2 L_1^2 \sigma^2}{144\epsilon^4} + \frac{\Delta L_0 \sigma^2 \log(1 + \sigma^2/\gamma^2)}{24\epsilon^4} + \frac{\Delta^2 L_1^2}{144\epsilon^2}.$$

*and suppose $d \geq T$ and $\epsilon \leq \min\left\{ \frac{\sqrt{2}}{3}\sqrt{\Delta L_0}, \frac{1}{\sqrt{3}}\sqrt{\Delta L_1 \gamma} \right\}$. If $\eta \leq \frac{1}{L_1}$, then there exists some $(f, g, \mathcal{D}) \in \mathcal{F}_{as}(\Delta, L_0, L_1, \sigma)$ such that $\|\nabla f(\boldsymbol{x}_t)\| = \epsilon$ for all $0 \leq t \leq T - 1$.*

*Proof.* Let $d \geq T$, and define $f : \mathbb{R}^d \to \mathbb{R}$ as:

$$f(\boldsymbol{x}) = \epsilon\langle \boldsymbol{x}, \mathbf{e}_1 \rangle + \sum_{i=2}^{T} h_i(\langle \boldsymbol{x}_t, \mathbf{e}_i \rangle),$$

where

$$h_i(x) = \begin{cases} \frac{L_0}{2}x^2 & |x| < \frac{a_i}{2} \\ -\frac{L_0}{2}(x - a_i)^2 + \frac{L_0}{4}a_i^2 & |x| \in \left[\frac{a_i}{2}, a_i\right] \\ \frac{L_0}{4}a_i^2 & |x| > a_i \end{cases}$$

$$a_i = \alpha_i \sigma$$

$$\alpha_i = \frac{\eta}{\sqrt{\gamma^2 + (i-2)(\epsilon^2 + \sigma^2)}}.$$

To see that $f$ is $(L_0, L_1)$-smooth, notice that each $h_i$ is $L_0$-smooth. Therefore, for any $\boldsymbol{x}, \boldsymbol{y} \in \mathbb{R}^d$,

$$\|\nabla f(\boldsymbol{x}) - \nabla f(\boldsymbol{y})\|^2 = (\nabla_1 f(\boldsymbol{x}) - \nabla_1 f(\boldsymbol{y}))^2 + \sum_{i=2}^{d} (\nabla_i f(\boldsymbol{x}) - \nabla_i f(\boldsymbol{y}))^2$$

$$= \sum_{i=2}^{d} (h_i'(x_i) - h_i'(y_i))^2$$

$$\leq L^2 \sum_{i=2}^{d} (x_i - y_i)^2$$

$$\leq L^2 \|\boldsymbol{x} - \boldsymbol{y}\|^2.$$

Therefore $f$ is $L_0$-smooth, and consequently is also $(L_0, L_1)$-smooth. We will also define the following stochastic gradient for $f$:

$$F(\boldsymbol{x}, \xi) = \nabla f(\boldsymbol{x}) + (2\xi - 1)\sigma \mathbf{e}_{j(\boldsymbol{x})},$$

where

$$j(\boldsymbol{x}) = \begin{cases} T & \langle \boldsymbol{x}, \mathbf{e}_i \rangle \neq 0 \text{ for all } i \text{ with } 2 \leq i \leq d \\ \min \{2 \leq i \leq d \mid \langle \boldsymbol{x}, \mathbf{e}_i \rangle = 0\} & \text{otherwise} \end{cases}$$

This oracle is defined so that the stochastic gradient noise at step $t$ only affects coordinate $t + 2$ (this will be shown later). Let $\mathcal{D}$ be the distribution of $\xi$, defined as $P(\xi = 0) = P(\xi = 1) = \frac{1}{2}$. With this definition, the stochastic gradient $F$ satisfies

$$\mathbb{E}[F(\boldsymbol{x}, \xi)] = \nabla f(\boldsymbol{x})$$
$$\|F(\boldsymbol{x}, \xi) - \nabla f(\boldsymbol{x})\| \leq \sigma \quad \text{(almost surely)}.$$

Therefore, all of the conditions for $(f, F, \mathcal{D}) \in \mathcal{F}_{\mathrm{as}}(\Delta, L_0, L_1, \sigma)$ are satisfied other than the condition that $f$ is bounded from below and $f(\boldsymbol{x}_0) - \inf_{\boldsymbol{x}} f(\boldsymbol{x}) \leq \Delta$. This condition will be addressed at the end of this lemma's proof.

Now consider the trajectory when optimizing $f$ from the starting point $\boldsymbol{x}_0 = \mathbf{0}$. We claim that, for each $0 \leq t \leq T - 1$, the iterate $\boldsymbol{x}_t$ satisfies the following conditions:

$$\langle \boldsymbol{x}_t, \mathbf{e}_1 \rangle = -\epsilon \sum_{i=2}^{t+1} \alpha_i \tag{10}$$

$$|\langle \boldsymbol{x}_t, \mathbf{e}_j \rangle| = a_j \quad \text{for all } 2 \leq j \leq t + 1 \tag{11}$$

$$\langle \boldsymbol{x}_t, \mathbf{e}_j \rangle = 0 \quad \text{for all } j > t + 1. \tag{12}$$

We will prove this claim by induction. The base case $t = 0$ holds since $\boldsymbol{x}_0 = \mathbf{0}$. So suppose that for some $0 \leq t \leq T - 2$ the claim holds for all $0 \leq i \leq t$. Then for each such $i$,

$$\nabla_1 f(\boldsymbol{x}_i) = \epsilon$$

$$\nabla_j f(\boldsymbol{x}_i) = h_j'(\langle \boldsymbol{x}_i, \mathbf{e}_j \rangle) \overset{(i)}{=} h_j'(a_j) \overset{(iii)}{=} 0 \quad \text{for all } 2 \leq j \leq i + 1$$

$$\nabla_j f(\boldsymbol{x}_i) = h_j'(\langle \boldsymbol{x}_i, \mathbf{e}_j \rangle) \overset{(ii)}{=} h_j'(0) \overset{(iv)}{=} 0 \quad \text{for all } j > i + 1,$$

where $(i)$ uses Equation 11 from the inductive hypothesis together with the fact that $h_j'(a_j) = h_j'(-a_j) = 0$, $(ii)$ uses Equation 12 from the inductive hypothesis, and both $(iii)$ and $(iv)$ use the definition of $h_j$. Therefore $\nabla f(\boldsymbol{x}_i) = \epsilon \mathbf{e}_1$. Also, $j(\boldsymbol{x}) = i + 2$. From the definition of the stochastic gradient oracle,

$$F(\boldsymbol{x}_i, \xi_i) = \nabla f(\boldsymbol{x}_i) + (2\xi_i - 1)\sigma \mathbf{e}_{j(\boldsymbol{x}_i)}$$
$$= \epsilon \mathbf{e}_1 + (2\xi_t - 1)\sigma \mathbf{e}_{i+2}.$$

Therefore, the update from $\boldsymbol{x}_t$ to $\boldsymbol{x}_{t+1}$ only affects coordinates with index 1 and index $t + 2$. Further, the above implies $\|F(\boldsymbol{x}_i, \xi_i)\|^2 = \epsilon^2 + \sigma^2$. Therefore, the effective learning rate of the algorithm at step $t$ is

$$\eta_t = \frac{\eta}{\sqrt{\gamma^2 + \sum_{i=0}^{t-1} \|F(\boldsymbol{x}_i, \xi_i)\|^2}} = \frac{\eta}{\sqrt{\gamma^2 + t(\epsilon^2 + \sigma^2)}} = \alpha_{t+2}.$$

We can then verify the inductive hypothesis for step $t+1$ by considering the coordinates of $\boldsymbol{x}_{t+1}$:

$$
\begin{aligned}
\langle \boldsymbol{x}_{t+1}, \mathbf{e}_1 \rangle &= \langle \boldsymbol{x}_t - \eta_t F(\boldsymbol{x}_t, \xi_t), \mathbf{e}_1 \rangle \\
&= \langle \boldsymbol{x}_t, \mathbf{e}_1 \rangle - \eta_t \langle F(\boldsymbol{x}_t, \xi_t), \mathbf{e}_1 \rangle \\
&= \langle \boldsymbol{x}_t, \mathbf{e}_1 \rangle - \epsilon \alpha_{t+2} \\
&\stackrel{(i)}{=} -\epsilon \left( \sum_{i=2}^{t+1} \alpha_t \right) - \epsilon \alpha_{t+2} \\
&= -\epsilon \sum_{i=1}^{t+2} \alpha_t,
\end{aligned}
$$

where $(i)$ uses Equation 10 from the inductive hypothesis, and this completes the inductive step for Equation 10. For Equation 11, we separately consider $j \leq t+1$ and $j = t+2$. For $j \leq t+1$,

$$
|\langle \boldsymbol{x}_{t+1}, \mathbf{e}_j \rangle| = |\langle \boldsymbol{x}_t, \mathbf{e}_j \rangle - \eta_t \langle F(\boldsymbol{x}_t, \xi_t), \mathbf{e}_j \rangle| = |\langle \boldsymbol{x}_t, \mathbf{e}_j \rangle| \stackrel{(i)}{=} a_j,
$$

where $(i)$ uses Equation 11 from the inductive hypothesis. For $j = t+2$:

$$
|\langle \boldsymbol{x}_{t+1}, \mathbf{e}_{t+2} \rangle| = |\langle \boldsymbol{x}_t, \mathbf{e}_{t+2} \rangle - \eta_t \langle F(\boldsymbol{x}_t, \xi_t), \mathbf{e}_{t+2} \rangle| \stackrel{(i)}{=} \eta_t \sigma = \alpha_{t+2} \sigma = a_{t+2},
$$

where $(i)$ uses Equation 12 from the inductive hypothesis. This completes the inductive step for Equation 11. For Equation 12, we consider $j > t+1$:

$$
\langle \boldsymbol{x}_{t+1}, \mathbf{e}_j \rangle = \langle \boldsymbol{x}_t, \mathbf{e}_j \rangle - \eta_t \langle F(\boldsymbol{x}_t, \xi_t), \mathbf{e}_j \rangle = 0,
$$

where the last equality uses Equation 12. This completes the inductive step for Equation 12, and consequently completes the induction. As a result, we have that $\nabla f(\boldsymbol{x}_t) = \epsilon$ for all $0 \leq t \leq T-1$.

The only remaining detail is whether the objective $f$ satisfies the condition $f(\boldsymbol{x}_0) - \inf_{\boldsymbol{x}} f(\boldsymbol{x}) \leq \Delta$. Actually, $f$ does not satisfy this condition because $f$ is not even lower bounded, due to the linear term $\epsilon \langle \boldsymbol{x}, \mathbf{e}_1 \rangle$. Similarly to Drori & Shamir (2020), we instead argue that there exists a lower bounded function $\hat{f}$ that has the same first-order information as $f$ at all of the points $\boldsymbol{x}_t$ for $0 \leq t \leq T-1$. If this happens, then the behavior of $A$ when optimizing $\hat{f}$ is the same as that of $A$ when optimizing $f$, so the conclusion $\|\nabla \hat{f}(\boldsymbol{x}_t)\| = \epsilon$ still holds. Specifically, we need $\hat{f}$ which is lower bounded and that satisfies:

$$
\nabla \hat{f}(\boldsymbol{x}_t) = \nabla f(\boldsymbol{x}_t), \quad \hat{f}(\boldsymbol{x}_t) = f(\boldsymbol{x}_t)
$$

for all $0 \leq t \leq T$. The existence of such an $\hat{f}$ follows immediately from Lemma 1 of Drori & Shamir (2020), and this $\hat{f}$ satisfies

$$
\inf_{\boldsymbol{x}} \hat{f}(\boldsymbol{x}) \geq \min_{0 \leq t \leq T-1} f(\boldsymbol{x}_t) - \frac{3\epsilon^2}{2L_0},
$$

so that

$$
\hat{f}(\boldsymbol{x}_0) - \inf_{\boldsymbol{x}} \hat{f}(\boldsymbol{x}) \leq \frac{3\epsilon^2}{2L_0} + \max_{0 \leq t \leq T-1} -f(\boldsymbol{x}_t). \tag{13}
$$

Recall that $f(\boldsymbol{x}_0) = 0$. For all $t \geq$, we can write each $-f(\boldsymbol{x}_t)$ as:

$$
\begin{aligned}
-f(\boldsymbol{x}_t) &= -\epsilon \langle \boldsymbol{x}_t, \mathbf{e}_1 \rangle - \sum_{i=2}^{T} h_i(\langle \boldsymbol{x}_t, \mathbf{e}_i \rangle) \\
&\stackrel{(i)}{=} \epsilon^2 \sum_{i=2}^{t+1} \alpha_i - \sum_{i=2}^{t+1} h_i(a_i) - \sum_{i=t+2}^{T} h_i(0) \\
&\stackrel{(ii)}{=} \epsilon^2 \sum_{i=2}^{t+1} \alpha_i - \frac{L_0}{4} \sum_{i=2}^{t+1} a_i^2 \\
&= \epsilon^2 \eta \underbrace{\sum_{i=0}^{t-1} \frac{1}{\sqrt{\gamma^2 + i(\epsilon^2 + \sigma^2)}}}_{S_1} - \frac{L_0 \sigma^2}{4} \eta^2 \underbrace{\sum_{i=0}^{t-1} \frac{1}{\gamma^2 + i(\epsilon^2 + \sigma^2)}}_{S_2}, \tag{14}
\end{aligned}
$$

where $(i)$ uses Equation 10, Equation 11, and Equation 12, and $(ii)$ uses the definition of $h_i$. We can bound $S_1$ as follows:

$$
\begin{aligned}
S_1 &= \sum_{i=0}^{t-1} \frac{1}{\sqrt{\gamma^2 + i(\epsilon^2 + \sigma^2)}} = \frac{1}{\gamma} + \sum_{i=1}^{t-1} \frac{1}{\sqrt{\gamma^2 + i(\epsilon^2 + \sigma^2)}} \\
&\leq \frac{1}{\gamma} + \int_0^{t-1} \frac{1}{\sqrt{\gamma^2 + x(\epsilon^2 + \sigma^2)}} \, dx \stackrel{(i)}{=} \frac{1}{\gamma} + \frac{1}{\epsilon^2 + \sigma^2} \int_{\gamma^2}^{\gamma^2 + (t-1)(\epsilon^2 + \sigma^2)} \frac{1}{\sqrt{u}} \, du \\
&= \frac{1}{\gamma} + \frac{2}{\epsilon^2 + \sigma^2} \left( \sqrt{\gamma^2 + (t-1)(\epsilon^2 + \sigma^2)} - \gamma \right) \\
&= \frac{1}{\gamma} + \frac{2}{\epsilon^2 + \sigma^2} \left( \sqrt{\gamma^2 + (t-1)(\epsilon^2 + \sigma^2)} - \gamma \right) \frac{\sqrt{\gamma^2 + (t-1)(\epsilon^2 + \sigma^2)} + \gamma}{\sqrt{\gamma^2 + (t-1)(\epsilon^2 + \sigma^2)} + \gamma} \\
&= \frac{1}{\gamma} + \frac{2}{\epsilon^2 + \sigma^2} \frac{(t-1)(\epsilon^2 + \sigma^2)}{\sqrt{\gamma^2 + (t-1)(\epsilon^2 + \sigma^2)} + \gamma} = \frac{1}{\gamma} + \frac{2(t-1)}{\sqrt{\gamma^2 + (t-1)(\epsilon^2 + \sigma^2)} + \gamma} \\
&\leq \frac{1}{\gamma} + \frac{2(t-1)}{\sqrt{(t-1)(\epsilon^2 + \sigma^2)}} = \frac{1}{\gamma} + \frac{2\sqrt{t-1}}{\sqrt{\epsilon^2 + \sigma^2}},
\end{aligned}
\tag{15}
$$

where $(i)$ uses the substitution $u = \gamma^2 + x(\epsilon^2 + \sigma^2)$. Similarly for $S_2$:

$$
\begin{aligned}
S_2 &= \sum_{i=0}^{t-1} \frac{1}{\gamma^2 + i(\epsilon^2 + \sigma^2)} \geq \int_0^t \frac{1}{\gamma^2 + x(\epsilon^2 + \sigma^2)} \, dx \\
&\stackrel{(i)}{=} \frac{1}{\epsilon^2 + \sigma^2} \int_{\gamma^2}^{\gamma^2 + t(\epsilon^2 + \sigma^2)} \frac{1}{u} \, du = \frac{1}{\epsilon^2 + \sigma^2} \log \left( \frac{\gamma^2 + t(\epsilon^2 + \sigma^2)}{\gamma^2} \right) \\
&= \frac{1}{\epsilon^2 + \sigma^2} \log \left( 1 + \frac{t(\epsilon^2 + \sigma^2)}{\gamma^2} \right) \geq \frac{1}{\epsilon^2 + \sigma^2} \log \left( 1 + \frac{t\sigma^2}{\gamma^2} \right),
\end{aligned}
\tag{16}
$$

where $(i)$ uses the substitution $u = \gamma^2 + x(\epsilon^2 + \sigma^2)$. Plugging Equation 15 and Equation 16 into Equation 14:

$$
\begin{aligned}
-f(\boldsymbol{x}_t) &\leq \epsilon^2 \left( \frac{1}{\gamma} + \frac{2\sqrt{t-1}}{\sqrt{\epsilon^2 + \sigma^2}} \right) \eta - \frac{L_0 \sigma^2 \log \left( 1 + \frac{t\sigma^2}{\gamma^2} \right)}{4(\epsilon^2 + \sigma^2)} \eta^2 \\
&\leq \epsilon^2 \left( \frac{1}{\gamma} + \frac{2\sqrt{T-1}}{\sqrt{\epsilon^2 + \sigma^2}} \right) \eta - \frac{L_0 \sigma^2 \log \left( 1 + \frac{\sigma^2}{\gamma^2} \right)}{4(\epsilon^2 + \sigma^2)} \eta^2
\end{aligned}
$$

We can decompose $T = 1 + T_1 + T_2$, where

$$
T_1 = \frac{\Delta^2 L_1^2 \sigma^2}{144\epsilon^4} + \frac{\Delta^2 L_1^2}{144\epsilon^2}, \quad T_2 = \frac{\Delta L_0 \sigma^2 \log(1 + \sigma^2/\gamma^2)}{24\epsilon^4}.
$$

Then $\sqrt{T-1} = \sqrt{T_1 + T_2} \leq \sqrt{T_1} + \sqrt{T_2}$, so

$$
-f(\boldsymbol{x}_t) \leq \underbrace{\epsilon^2 \left( \frac{1}{\gamma} + \frac{2\sqrt{T_1}}{\sqrt{\epsilon^2 + \sigma^2}} \right) \eta}_{D_1} + \underbrace{\frac{2\epsilon^2 \sqrt{T_2}}{\sqrt{\epsilon^2 + \sigma^2}} \eta - \frac{L_0 \sigma^2 \log \left( 1 + \frac{\sigma^2}{\gamma^2} \right)}{4(\epsilon^2 + \sigma^2)} \eta^2}_{D_2}.
\tag{17}
$$

We can bound $D_1$ and $D_2$ separately:

$$
\begin{aligned}
D_1 &= \left( \frac{\epsilon^2}{\gamma} + \frac{2\epsilon^2\sqrt{T_1}}{\sqrt{\epsilon^2+\sigma^2}} \right) \eta \\
&\stackrel{(i)}{\leq} \frac{\epsilon^2}{\gamma L_1} + \frac{2\epsilon^2\sqrt{T_1}}{L_1\sqrt{\epsilon^2+\sigma^2}} \\
&\stackrel{(ii)}{\leq} \frac{\Delta}{6} + \frac{2\epsilon^2\sqrt{T_1}}{L_1\sqrt{\epsilon^2+\sigma^2}} \\
&\stackrel{(iii)}{=} \frac{\Delta}{6} + \frac{2\epsilon^2}{L_1\sqrt{\epsilon^2+\sigma^2}} \sqrt{\frac{\Delta^2 L_1^2 \sigma^2}{144\epsilon^4} + \frac{\Delta^2 L_1^2}{144\epsilon^2}} \\
&\stackrel{(iv)}{\leq} \frac{\Delta}{6} + \frac{2\epsilon^2}{L_1\sqrt{\epsilon^2+\sigma^2}} \left( \frac{\Delta L_1 \sigma}{12\epsilon^2} + \frac{\Delta L_1}{12\epsilon} \right) \\
&= \frac{\Delta}{6} + \frac{\Delta\sigma}{6\sqrt{\epsilon^2+\sigma^2}} + \frac{\Delta\epsilon}{6\sqrt{\epsilon^2+\sigma^2}} \\
&\leq \frac{\Delta}{6} + \frac{\Delta}{6} + \frac{\Delta}{6} = \frac{\Delta}{3},
\end{aligned}
$$

where $(i)$ uses the condition $\eta \leq 1/L_1$, $(ii)$ uses the condition $\epsilon \leq \frac{1}{\sqrt{6}}\sqrt{\gamma\Delta L_1}$, $(iii)$ uses the definition of $T_1$, and $(iv)$ uses $\sqrt{a+b} \leq \sqrt{a} + \sqrt{b}$. Notice that $D_2$ is a quadratic function of $\eta$ with negative leading coefficient, so $D_2$ is upper bounded by the vertex of the corresponding parabola, i.e. $ax^2 + bx \leq -b^2/2a$ when $a < 0$. Therefore

$$
\begin{aligned}
D_2 &\leq \left( \frac{2\epsilon^2\sqrt{T_2}}{\sqrt{\epsilon^2+\sigma^2}} \right)^2 \frac{2(\epsilon^2+\sigma^2)}{L_0\sigma^2 \log\left(1+\frac{\sigma^2}{\gamma^2}\right)} \\
&= \frac{8\epsilon^4}{L_0\sigma^2 \log\left(1+\frac{\sigma^2}{\gamma^2}\right)} T_2 \\
&\stackrel{(i)}{=} \frac{8\epsilon^4}{L_0\sigma^2 \log\left(1+\frac{\sigma^2}{\gamma^2}\right)} \frac{\Delta L_0 \sigma^2 \log(1+\sigma^2/\gamma^2)}{24\epsilon^4} = \frac{\Delta}{3},
\end{aligned}
$$

where $(i)$ uses the definition of $T_2$.

Finally, plugging back to Equation 17 yields $-f(\boldsymbol{x}_t) \leq \frac{2\Delta}{3}$, and plugging this back into Equation 13:

$$
\begin{aligned}
\hat{f}(\boldsymbol{x}_0) - \inf_{\boldsymbol{x}} \hat{f}(\boldsymbol{x}) &\leq \frac{3\epsilon^2}{2L_0} + \frac{2\Delta}{3} \\
&\stackrel{(i)}{\leq} \frac{\Delta}{3} + \frac{2\Delta}{3} = \Delta,
\end{aligned}
$$

where $(i)$ uses the condition $\epsilon \leq \frac{\sqrt{2}}{3}\sqrt{\Delta L_0}$. Therefore $\hat{f}$ satisfies all conditions of $\mathcal{F}_{\text{as}}(\Delta, L_0, L_1, \sigma)$.
$\square$

**Theorem 5.** *[Restatement of Theorem 1] Let $\Delta, L_0, L_1, \sigma > 0$, and let $\mathcal{F} = \mathcal{F}_{\text{as}}(\Delta, L_0, L_1, \sigma)$. Let algorithm $A_{DAN}$ denote Decorrelated AdaGrad-Norm with parameters $\eta > 0$ and*

$$
0 < \gamma \leq \frac{\Delta L_1}{8 \log\left(1 + 48\frac{\Delta L_1^2}{L_0}\right)}.
$$

*Let $0 < \epsilon \leq \min\left\{ \frac{\sqrt{2}}{3}\sqrt{\Delta L_0}, \frac{1}{\sqrt{3}}\sqrt{\Delta L_1 \gamma}, \Delta L_1 \right\}$. If $\Delta L_1^2 \geq L_0$, then*

$$
\mathcal{T}(A_{DAN}, \mathcal{F}, \epsilon) \geq 1 + \frac{\Delta^2 L_1^2 \sigma^2}{144\epsilon^4} + \frac{\Delta L_0 \sigma^2 \log(1+\sigma^2/\gamma^2)}{24\epsilon^4} + \frac{\Delta^2 L_1^2}{144\epsilon^2}.
$$

*Proof.* We only need to combine Lemmas 1 and 2. If $\eta \leq \frac{1}{L_1}$, then

$$\frac{\gamma}{\eta} \leq \frac{\Delta L_1}{8\eta \log \left(1 + 48\frac{\Delta L_1^2}{L_0}\right)} \leq \frac{\Delta L_1^2}{8 \log \left(1 + 48\frac{\Delta L_1^2}{L_0}\right)},$$

so the conditions of $\gamma$ and $\eta$ in Lemma 1 are satisfied. Therefore, by Lemma 1 there exists a problem instance $(f, g, \mathcal{D}) \in \mathcal{F}$ for which $\|\nabla f(\boldsymbol{x}_t)\| \geq \Delta L_0 > \epsilon$ for all $t \geq 0$. If $\eta \leq \frac{1}{L_1}$, then by Lemma 1 there exists a problem instance $(f, g, \mathcal{D}) \in \mathcal{F}$ for which $\|\nabla f(\boldsymbol{x}_t)\| \geq \epsilon$ for all $t \leq T := 1 + \frac{\Delta^2 L_1^2 \sigma^2}{144\epsilon^4} + \frac{\Delta L_0 \sigma^2 \log(1+\sigma^2/\gamma^2)}{24\epsilon^4} + \frac{\Delta^2 L_1^2}{144\epsilon^2}$. In both cases, $A_{\text{DAN}}$ requires at least $T$ gradient queries to find an $\epsilon$-approximate stationary point. $\qquad \square$

## B  PROOFS OF THEOREM 2 AND 3

**Lemma 7** (Restatement of Lemma 3). *Let $0 < \epsilon < \Delta L_1$. If the parameters of Decorrelated AdaGrad satisfy $\eta \geq \frac{\sqrt{2}\gamma}{L_1\sigma} \log \left(1 + \frac{L_1\epsilon}{L_0}\right)$, then for any $T \geq 1$, there exists some $f \in \mathcal{F}_{\text{as}}(\Delta, L_0, L_1, \sigma)$ such that $\|\nabla f(\boldsymbol{x}_t)\| \geq \epsilon$ for all $0 \leq t \leq T - 1$.*

*Similarly, if the parameters of AdaGrad satisfy $\eta \geq \frac{\sqrt{2}}{L_1} \log \left(1 + \frac{L_1\epsilon}{L_0}\right)$ and $\gamma \leq \sigma$, then for any $T \geq 1$ there exists some $f \in \mathcal{F}_{\text{as}}(\Delta, L_0, L_1, \sigma)$ such that $\|\nabla f(\boldsymbol{x}_t)\| \geq \epsilon$ for all $0 \leq t \leq T - 1$.*

*Proof.* First, recall the definition of $\psi$:

$$\tilde{\psi}(x) = \frac{L_0}{L_1^2} \left(\exp\left(L_1|x|\right) - L_1|x| - 1\right).$$

Then define

$$f(\boldsymbol{x}) = \sum_{i=1}^{T} \psi(\langle \boldsymbol{x}, \mathbf{e}_i \rangle).$$

To see that $f$ is $(L_0, L_1)$-smooth, let $\boldsymbol{x}, \boldsymbol{y} \in \mathbb{R}^d$. Denoting $\boldsymbol{x} = (x_1, \ldots, x_T)$ and $\boldsymbol{y} = (y_1, \ldots, y_T)$,

$$\begin{aligned}
\|\nabla f(\boldsymbol{x}) - \nabla f(\boldsymbol{y})\|^2 &= \sum_{i=1}^{T} \left(\nabla_i f(\boldsymbol{x}) - \nabla_i f(\boldsymbol{x})\right)^2 \\
&= \sum_{i=1}^{T} \left(\psi'(x_i) - \psi'(y_i)\right)^2 \\
&\overset{(i)}{\leq} \sum_{i=1}^{T} \left(L_0 + L_1|\psi'(x_i)|\right)^2 (x_i - y_i)^2 \\
&\overset{(ii)}{\leq} \sum_{i=1}^{T} \left(L_0 + L_1\|\nabla f(\boldsymbol{x})\|\right)^2 (x_i - y_i)^2 \\
&= \left(L_0 + L_1\|\nabla f(\boldsymbol{x})\|\right)^2 \sum_{i=1}^{T} (x_i - y_i)^2 \\
&= \left(L_0 + L_1\|\nabla f(\boldsymbol{x})\|\right)^2 \|\boldsymbol{x} - \boldsymbol{y}\|^2,
\end{aligned}$$

where $(i)$ uses the fact that $\psi$ is $(L_0, L_1)$-smooth and $(ii)$ uses $|\psi'(x_i)| \leq \|\nabla f(\boldsymbol{x})\|$. Therefore $f$ is $(L_0, L_1)$-smooth. Also, define $m = \frac{1}{L_1} \log \left(1 + \frac{L_1\epsilon}{L_0}\right)$, so that $\psi'(m) = \epsilon$. Consider the initial point

$\boldsymbol{x}_0 = m\mathbf{e}_1$. Then

$$f(\boldsymbol{x}_0) - \inf_{\boldsymbol{x}} f(\boldsymbol{x}) = \psi(m)$$

$$= \frac{L_0}{L_1^2}\left(\exp\left(L_1 m\right) - L_1 m - 1\right)$$

$$= \frac{L_0}{L_1^2}\left(1 + \frac{L_1\epsilon}{L_0} - \log\left(1 + \frac{L_1\epsilon}{L_0}\right) - 1\right)$$

$$= \frac{\epsilon}{L_1} - \frac{L_0}{L_1^2}\log\left(1 + \frac{L_1\epsilon}{L_0}\right)$$

$$= \frac{\epsilon}{L_1} \overset{(i)}{\leq} \Delta,$$

where $(i)$ uses the condition $\epsilon \leq \Delta L_1$.

We also define a stochastic gradient for $f$ as follows:

$$F(\boldsymbol{x}, \xi) = \nabla f(\boldsymbol{x}) + (2\xi - 1)\sigma\mathbf{e}_{j(x)},$$

where

$$j(x) = \begin{cases} 0 & \langle \boldsymbol{x}, \mathbf{e}_i \rangle \neq 0 \text{ for all } 1 \leq i \leq T \\ \min\left\{1 \leq i \leq T \mid \langle \boldsymbol{x}, \mathbf{e}_i \rangle = 0\right\} & \text{otherwise} \end{cases},$$

and the distribution $\mathcal{D}$ of $\Xi$ is defined as $P(\xi = 0) = P(\xi = 1) = 0.5$. Then $\mathbb{E}_\xi[F(\boldsymbol{x}, \xi)] = \nabla f(\boldsymbol{x})$ and $\|F(\boldsymbol{x}, \xi) - \nabla f(\boldsymbol{x})\| \leq \sigma$ almost surely. Therefore, $(f, g, \mathcal{D}) \in \mathcal{F}_{\text{as}}(\Delta, L_0, L_1, \sigma)$.

Now consider the trajectory of Decorrelated AdaGrad when optimizing $(f, g, \mathcal{D})$ from the initial point $\boldsymbol{x}_0 = m\mathbf{e}_1$. We claim that for all $0 \leq t \leq T - 1$:

$$|\langle \boldsymbol{x}_t, \mathbf{e}_{t+1} \rangle| \geq m \tag{18}$$

$$\langle \boldsymbol{x}_t, \mathbf{e}_j \rangle = 0 \text{ for all } j > t + 1, \tag{19}$$

which we will prove by induction on $t$. The base case $t = 0$ holds from the choice of the initial point $\boldsymbol{x}_0$. So suppose that Equation 18 and Equation 19 hold for all $0 \leq i \leq t$ for some $0 \leq t \leq T - 2$. Then $j(\boldsymbol{x}_t) = t + 2$, so

$$\langle F(\boldsymbol{x}_t, \xi_t), \mathbf{e}_{t+2} \rangle = \langle \nabla f(\boldsymbol{x}_t), \mathbf{e}_{t+2} \rangle + \langle (2\xi - 1)\sigma\mathbf{e}_{t+2}, \mathbf{e}_{t+2} \rangle$$

$$= \tilde{\psi}'(\langle \boldsymbol{x}_t, \mathbf{e}_{t+2} \rangle) + (2\xi - 1)\sigma$$

$$\overset{(i)}{=} \tilde{\psi}'(0) + (2\xi - 1)\sigma$$

$$= (2\xi - 1)\sigma,$$

where $(i)$ uses Equation 19 from the inductive hypothesis. Therefore, for Decorrelated AdaGrad:

$$\langle \boldsymbol{x}_{t+1}, \mathbf{e}_{t+2} \rangle = \langle \boldsymbol{x}_t, \mathbf{e}_{t+2} \rangle - \frac{\eta}{\sqrt{\gamma^2 + \sum_{i=0}^{t-1}\left(\langle F(\boldsymbol{x}_i, \xi_i), \mathbf{e}_{t+2} \rangle\right)^2}}(2\xi_t - 1)\sigma$$

$$\overset{(i)}{=} -\frac{\eta}{\sqrt{\gamma^2 + \sum_{i=0}^{t-1}\left(\langle F(\boldsymbol{x}_i, \xi_i), \mathbf{e}_{t+2} \rangle\right)^2}}(2\xi_t - 1)\sigma$$

$$\overset{(ii)}{=} -\frac{\eta}{\sqrt{\gamma^2 + \sum_{i=0}^{t-1}\left(\psi'(0)\right)^2}}(2\xi_t - 1)\sigma$$

$$= -\frac{\eta}{\gamma}(2\xi_t - 1)\sigma,$$

where both $(i)$ and $(ii)$ use Equation 19 from the inductive hypothesis. Therefore

$$|\langle \boldsymbol{x}_{t+1}, \mathbf{e}_{t+2} \rangle| = \frac{\eta}{\gamma}\sigma \geq m,$$

where the inequality uses the condition $\eta \geq \frac{\gamma m}{\sigma}$ for Decorrelated AdaGrad. Similarly for AdaGrad:

$$
\begin{aligned}
\langle \boldsymbol{x}_{t+1}, \mathbf{e}_{t+2} \rangle &= \langle \boldsymbol{x}_t, \mathbf{e}_{t+2} \rangle - \frac{\eta}{\sqrt{\gamma^2 + \sum_{i=0}^{t} \left( \langle F(\boldsymbol{x}_i, \xi_i), \mathbf{e}_{t+2} \rangle \right)^2}} (2\xi_t - 1)\sigma \\
&\stackrel{(i)}{=} -\frac{\eta}{\sqrt{\gamma^2 + \sum_{i=0}^{t} \left( \langle F(\boldsymbol{x}_i, \xi_i), \mathbf{e}_{t+2} \rangle \right)^2}} (2\xi_t - 1)\sigma \\
&\stackrel{(ii)}{=} -\frac{\eta}{\sqrt{\gamma^2 + \sum_{i=0}^{t-1} \left( \psi'(0) \right)^2 + (\psi'(0) + \sigma)^2}} (2\xi_t - 1)\sigma \\
&= -\frac{\eta}{\sqrt{\gamma^2 + \sigma^2}} (2\xi_t - 1)\sigma.
\end{aligned}
$$

where both $(i)$ and $(ii)$ use Equation 19 from the inductive hypothesis. Therefore

$$
|\langle \boldsymbol{x}_{t+1}, \mathbf{e}_{t+2} \rangle| = \frac{\eta}{\sqrt{\gamma^2 + \sigma^2}} \sigma \stackrel{(i)}{\geq} \frac{\eta}{\sqrt{2}} \stackrel{(ii)}{\geq} \frac{\eta}{\sqrt{2}} \geq m,
$$

where $(i)$ uses the assumed condition $\gamma \leq \sigma$ and $(ii)$ uses the condition $\eta \geq \sqrt{2}m$. This proves the inductive step for Equation 18, for both Decorrelated AdaGrad and AdaGrad. The inductive step for Equation 19 follows immediately from the inductive hypothesis (Equation 19) together with the stochastic gradient definition and $j(\boldsymbol{x}_t) = t + 2$. This completes the induction.

For all $0 \leq t \leq T - 1$, the conclusion of the lemma follows from Equation 18 by:

$$
\|\nabla f(\boldsymbol{x}_t)\| \geq \langle \nabla f(\boldsymbol{x}_t), \mathbf{e}_{t+1} \rangle = \psi'(\langle \boldsymbol{x}_t, \mathbf{e}_{t+1} \rangle) \stackrel{(i)}{\geq} \psi'(m) = \epsilon,
$$

where $(i)$ uses Equation 18 together with the fact that $\psi'(x)$ increases with $|x|$. $\qquad \square$

**Lemma 8** (Restatement of Lemma 4). *Let*

$$
0 < \epsilon < \min \left\{ \frac{\Delta L_1}{2}, \sqrt{\frac{\Delta L_1 \sigma}{4\sqrt{2} \log \left( 1 + \frac{\Delta L_1^2}{L_0} \right)}} \right\}.
$$

*If the parameters of Decorrelated AdaGrad satisfy $\eta \leq \frac{\sqrt{2}\gamma}{L_1 \sigma} \log \left( 1 + \frac{L_1 \epsilon}{L_0} \right)$, then there exists some $(f, g, \mathcal{D}) \in \mathcal{F}_{as}(\Delta, L_0, L_1, \sigma)$ such that $\|\nabla f(\boldsymbol{x}_t)\| \geq \epsilon$ for all*

$$
t \leq \frac{\Delta^2 L_0^2 \sigma^2}{256 \gamma^2 \epsilon^4} + \frac{\Delta^2 L_1^2 \sigma^2}{256 \gamma^2 \epsilon^2 \log^2 \left( 1 + \frac{\Delta L_1^2}{L_0} \right)}.
$$

*Similarly, if the parameters of AdaGrad satisfy $\eta \leq \frac{\sqrt{2}}{L_1} \log \left( 1 + \frac{L_1 \epsilon}{L_0} \right)$, then there exists some $(f, g, \mathcal{D}) \in \mathcal{F}_{as}(\Delta, L_0, L_1, \sigma)$ such that $\|\nabla f(\boldsymbol{x}_t)\| \geq \epsilon$ for all*

$$
t \leq \frac{\Delta^2 L_0^2}{128 \epsilon^4} + \frac{\Delta^2 L_1^2}{128 \epsilon^2 \log^2 \left( 1 + \frac{\Delta L_1^2}{L_0} \right)}.
$$

*Proof.* Define $m = \frac{1}{L_1} \log \left( 1 + \frac{L_1 \epsilon}{L_0} \right)$, and consider the objective

$$
f(x) = \begin{cases} -\epsilon(x + m) + \psi(m) & x < -m \\ \psi(x) & x \in [-m, m] \\ \epsilon(x - m) + \psi(m) & x > m \end{cases}.
$$

This function is differentiable everywhere since $\psi'(m) = \epsilon$ and $\psi'(-m) = -\epsilon$. Since $\psi$ is $(L_0, L_1)$-smooth, so is $f$. Also, $m$ satisfies

$$\psi(m) = \frac{L_0}{L_1^2}\left(\exp(L_1 m) - L_1 m - 1\right)$$

$$= \frac{L_0}{L_1^2}\left(1 + \frac{L_1\epsilon}{L_0} - \log\left(1 + \frac{L_1\epsilon}{L_0}\right) - 1\right)$$

$$= \frac{\epsilon}{L_1} - \frac{L_0}{L_1^2}\log\left(1 + \frac{L_1\epsilon}{L_0}\right)$$

$$\overset{(i)}{\leq} \frac{\epsilon}{L_1} \leq \frac{\Delta}{2},$$

where $(i)$ uses the condition $\epsilon \leq \frac{1}{2}\Delta L_1$. Therefore, with the initial point $x_0 = m + \frac{\Delta}{2\epsilon}$, the objective satisfies

$$f(x_0) - \inf_x f(x) = \epsilon(x_0 - m) + \psi(m)$$

$$= \epsilon\frac{\Delta}{2\epsilon} + \frac{\Delta}{2}$$

$$= \Delta.$$

We will define the stochastic gradient $g$ with noise distribution $\mathcal{D}$ as equal to the true gradient, i.e. $g(x, \xi) = f'(x)$ for every $x, \xi$. Therefore $(f, g, \mathcal{D}) \in \mathcal{F}_{\text{as}}(\Delta, L_0, L_1, \sigma)$.

Now consider the trajectory of Decorrelated AdaGrad when optimizing $(f, g, \mathcal{D})$ from the initial point $x_0 = m + \frac{\Delta}{2\epsilon}$. Let $t_0 = \max\{t \geq 0 \mid x_t \geq m\}$. Then $f'(x_t) = \epsilon$ for all $t \leq t_0$, so that

$$x_{t+1} = x_t - \frac{\eta\epsilon}{\sqrt{\gamma^2 + \sum_{i=0}^{t-1}\epsilon^2}} = x_t - \frac{\eta\epsilon}{\sqrt{\gamma^2 + t\epsilon^2}},$$

and unrolling yields

$$x_{t+1} = x_0 - \eta\epsilon\sum_{i=0}^{t}\frac{1}{\sqrt{\gamma^2 + i\epsilon^2}}$$

$$= x_0 - \frac{\eta\epsilon}{\gamma} - \eta\epsilon\sum_{i=1}^{t}\frac{1}{\sqrt{\gamma^2 + i\epsilon^2}}$$

$$\geq x_0 - \frac{\eta\epsilon}{\gamma} - \eta\epsilon\int_0^t\frac{1}{\sqrt{\gamma^2 + x\epsilon^2}}dx$$

$$= x_0 - \frac{\eta\epsilon}{\gamma} - \frac{2\eta}{\epsilon}\left[\sqrt{\gamma^2 + x\epsilon^2}\right]_0^t$$

$$= x_0 - \frac{\eta\epsilon}{\gamma} - \frac{2\eta}{\epsilon}\left(\sqrt{\gamma^2 + t\epsilon^2} - \gamma\right)$$

$$= x_0 - \frac{\eta\epsilon}{\gamma} - \frac{2\eta}{\epsilon}\frac{t\epsilon^2}{\sqrt{\gamma^2 + t\epsilon^2} + \gamma}$$

$$\geq x_0 - \frac{\eta\epsilon}{\gamma} - \frac{2\eta t\epsilon}{\sqrt{\gamma^2 + t\epsilon^2}}.$$

Plugging $t = t_0$ then yields

$$x_{t_0+1} \geq x_0 - \frac{\eta\epsilon}{\gamma} - \frac{2\eta t_0\epsilon}{\sqrt{\gamma^2 + t_0\epsilon^2}}. \tag{20}$$

On the other hand,

$$x_{t_0+1} < m. \tag{21}$$

Combining Equation 20 and Equation 21:

$$m \geq x_0 - \frac{\eta\epsilon}{\gamma} - \frac{2\eta t_0\epsilon}{\sqrt{\gamma^2 + t_0\epsilon^2}}$$

$$\frac{2\eta t_0\epsilon}{\sqrt{\gamma^2 + t_0\epsilon^2}} \geq x_0 - m - \frac{\eta\epsilon}{\gamma}$$

$$\frac{2\eta t_0\epsilon}{\sqrt{\gamma^2 + t_0\epsilon^2}} \overset{(i)}{\geq} \frac{\Delta}{2\epsilon} - \frac{\eta\epsilon}{\gamma}$$

$$\frac{t_0}{\sqrt{\gamma^2 + t_0\epsilon^2}} \geq \frac{1}{\epsilon}\left(\frac{\Delta}{4\eta\epsilon} - \frac{\epsilon}{2\gamma}\right), \tag{22}$$

where $(i)$ uses the definition of $x_0$. The last term in Equation 22 can be bounded as

$$\frac{\epsilon}{2\gamma} \leq \frac{\Delta}{8\eta\epsilon} \frac{4\eta\epsilon^2}{\Delta\gamma}$$

$$\overset{(i)}{\leq} \frac{\Delta}{8\eta\epsilon} \frac{4\epsilon^2}{\Delta\gamma} \frac{\sqrt{2}\gamma}{L_1\sigma} \log\left(1 + \frac{L_1\epsilon}{L_0}\right)$$

$$= \frac{\Delta}{8\eta\epsilon} \frac{4\sqrt{2}\epsilon^2}{\Delta L_1\sigma} \log\left(1 + \frac{L_1\epsilon}{L_0}\right)$$

$$\overset{(ii)}{\leq} \frac{\Delta}{8\eta\epsilon} \frac{4\sqrt{2}\epsilon^2}{\Delta L_1\sigma} \log\left(1 + \frac{\Delta L_1^2}{L_0}\right)$$

$$\overset{(iii)}{\leq} \frac{\Delta}{8\eta\epsilon},$$

where $(i)$ uses the condition $\eta \leq \frac{\sqrt{2}\gamma}{L_1\sigma} \log\left(1 + \frac{L_1\epsilon}{L_0}\right)$, $(ii)$ uses the condition $\epsilon \leq \Delta L_1$, and $(iii)$ uses the condition $\epsilon \leq \sqrt{\frac{\Delta L_1\sigma}{4\sqrt{2}\log\left(1 + \frac{\Delta L_1^2}{L_0}\right)}}$. Plugging back to Equation 22 yields

$$\frac{t_0}{\sqrt{\gamma^2 + t_0\epsilon^2}} \geq \frac{\Delta}{8\eta\epsilon^2}$$

$$\frac{t_0}{\sqrt{t_0\epsilon^2}} \geq \frac{\Delta}{8\eta\epsilon^2}$$

$$\sqrt{t_0} \geq \frac{\Delta}{8\eta\epsilon}$$

$$t_0 \geq \frac{\Delta^2}{64\eta^2\epsilon^2} \tag{23}$$

From the assumed upper bound on $\eta$,

$$\eta \leq \frac{\sqrt{2}\gamma}{L_1\sigma} \log\left(1 + \frac{L_1\epsilon}{L_0}\right) \overset{(i)}{\leq} \frac{\sqrt{2}\gamma}{L_1\sigma} \log\left(1 + \frac{\Delta L_1^2}{L_0}\right),$$

and

$$\eta \leq \frac{\sqrt{2}\gamma}{L_1\sigma} \log\left(1 + \frac{L_1\epsilon}{L_0}\right) \overset{(ii)}{\leq} \frac{\sqrt{2}\gamma\epsilon}{L_0\sigma},$$

where $(i)$ uses the condition $\epsilon \leq \Delta L_1$ and $(ii)$ uses $\log(1+x) \leq x$. Therefore

$$\frac{1}{\eta} \geq \max\left\{\frac{L_1\sigma}{\sqrt{2}\gamma\log\left(1 + \frac{\Delta L_1^2}{L_0}\right)}, \frac{L_0\sigma}{\sqrt{2}\gamma\epsilon}\right\}$$

$$\frac{1}{\eta^2} \geq \max\left\{\frac{L_1^2\sigma^2}{2\gamma^2\log^2\left(1 + \frac{\Delta L_1^2}{L_0}\right)}, \frac{L_0^2\sigma^2}{2\gamma^2\epsilon^2}\right\} \geq \frac{L_1^2\sigma^2}{4\gamma^2\log^2\left(1 + \frac{\Delta L_1^2}{L_0}\right)} + \frac{L_0^2\sigma^2}{4\gamma^2\epsilon^2}.$$

Plugging back to Equation 23 yields

$$t_0 \geq \frac{\Delta^2 L_0^2 \sigma^2}{256\gamma^2 \epsilon^4} + \frac{\Delta^2 L_1^2 \sigma^2}{256\gamma^2 \epsilon^2 \log^2 \left(1 + \frac{\Delta L_1^2}{L_0}\right)}.$$

Since $|f'(x_t)| = \epsilon$ for all $t \leq t_0$, this completes the proof for Decorrelated AdaGrad.

The corresponding proof for AdaGrad is nearly identical, so we list only the key steps here. For all $t \leq t_0$,

$$x_{t+1} = x_t - \frac{\eta \epsilon}{\sqrt{\gamma^2 + (t+1)\epsilon^2}} \geq x_t - \frac{\eta}{\sqrt{t+1}}.$$

After unrolling and applying the same bound for $\sum_i \frac{1}{\sqrt{i}}$ as in the decorrelated case, then choosing $t = t_0$, we have

$$x_{t_0+1} \geq x_0 - 2\eta\sqrt{t_0 + 1}.$$

From $x_{t_0+1} \leq m$, we have

$$m \geq x_0 - 2\eta\sqrt{t_0 + 1}$$

$$t_0 + 1 \geq \frac{\Delta^2}{16\eta^2 \epsilon^2}.$$

The assumed upper bound on $\eta$ yields

$$\frac{1}{\eta^2} \geq \frac{L_1^2}{4\log^2\left(1 + \frac{\Delta L_1^2}{L_0}\right)} + \frac{L_0^2}{4\epsilon^2},$$

so that

$$t_0 + 1 \geq \frac{\Delta^2 L_0^2}{64\epsilon^4} + \frac{\Delta^2 L_1^2}{64\epsilon^2 \log^2\left(1 + \frac{\Delta L_1^2}{L_0}\right)}.$$

Since $2t_0 \geq t_0 + 1$ for all $t_0 \geq 1$, this means

$$t_0 \geq \frac{\Delta^2 L_0^2}{128\epsilon^4} + \frac{\Delta^2 L_1^2}{128\epsilon^2 \log^2\left(1 + \frac{\Delta L_1^2}{L_0}\right)}.$$

$\square$

**Theorem 6.** *[Restatement of Theorem 2] Let $\Delta, L_0, L_1, \sigma > 0$ and let $\mathcal{F} = \mathcal{F}_{as}(\Delta, L_0, L_1, \sigma)$. Let $A_{DA}$ and $A_{ada}$ denote Decorrelated AdaGrad and AdaGrad (respectively) with parameters $\eta, \gamma > 0$. Suppose*

$$0 < \epsilon < \min\left\{\frac{\Delta L_1}{2}, \sqrt{\frac{\Delta L_1 \sigma}{4\sqrt{2}\log\left(1 + \frac{\Delta L_1^2}{L_0}\right)}}\right\}.$$

*Then*

$$\mathcal{T}(A_{DA}, \mathcal{F}, \epsilon, \delta) \geq \frac{\Delta^2 L_0^2 \sigma^2}{256\gamma^2 \epsilon^4} + \frac{\Delta^2 L_1^2 \sigma^2}{256\gamma^2 \epsilon^2 \log^2\left(1 + \frac{\Delta L_1^2}{L_0}\right)}.$$

*Also, if $\gamma \leq \sigma$, then*

$$\mathcal{T}(A_{ada}, \mathcal{F}, \epsilon, \delta) \geq \frac{\Delta^2 L_0^2}{128\epsilon^4} + \frac{\Delta^2 L_1^2}{128\epsilon^2 \log^2\left(1 + \frac{\Delta L_1^2}{L_0}\right)}.$$

*Proof.* We only have to combine Lemmas 3 and 4. We first consider Decorrelated AdaGrad. If $\eta \geq \frac{\sqrt{2}\gamma}{L_1\sigma}\log\left(1 + \frac{L_1\epsilon}{L_0}\right)$, then by Lemma 3 there exists a problem instance for which Decorrelated AdaGrad will never find an $\epsilon$-approximate stationary point. Otherwise, by Lemma 4 there exists a problem instance for which Decorrelated AdaGrad requires a number of steps at least as large as

$$\frac{\Delta^2 L_0^2 \sigma^2}{256\gamma^2 \epsilon^4} + \frac{\Delta^2 L_1^2 \sigma^2}{256\gamma^2 \epsilon^2 \log^2\left(1 + \frac{\Delta L_1^2}{L_0}\right)}.$$

Therefore in either case, we have

$$\mathcal{T}(A_{\text{DA}}, \mathcal{F}, \epsilon, \delta) \geq \frac{\Delta^2 L_0^2 \sigma^2}{256 \gamma^2 \epsilon^4} + \frac{\Delta^2 L_1^2 \sigma^2}{256 \gamma^2 \epsilon^2 \log^2 \left(1 + \frac{\Delta L_1^2}{L_0}\right)}.$$

$\square$

The corresponding proof for AdaGrad (Theorem 3) is nearly identical, so we omit it.

## C PROOF OF THEOREM 4

### C.1 PRELIMINARY DEFINITIONS

We first provide definitions of constants and objects that will be used throughout the proof.

For $p \in (0, 1)$ and $\lambda > 0$, consider the random walk parameterized by $(p, \lambda)$:

$$\begin{aligned} X_0 &= 1 \\ P(X_{t+1} = X_t + \lambda) &= p \\ P(X_{t+1} = X_t - 1) &= 1 - p. \end{aligned} \tag{24}$$

Then we can define

$$\begin{aligned} z_{p,\lambda} &= P(\exists t > 0 : X_t \leq 0) \\ \lambda_0(p, \delta) &= \inf \{\lambda \geq 0 : z_{p,\lambda} \leq 1 - \delta\} \\ \zeta(p, \delta) &= \lambda_0(p, \delta) - \lambda_0(p, 0). \end{aligned} \tag{25}$$

Informally, $z_{p,\lambda}$ is the probability that the random walk reaches a non-positive value, and $\lambda_0(p, \delta)$ is the smallest $\lambda$ required to ensure that the chance of never reaching a non-positive value is at least $\delta$.

For $\sigma_2 \geq 3$, define the following constants:

$$\gamma_1 = \frac{\log\left(1 + 2\zeta(2/3, \delta)\right)}{\log 2}, \quad \gamma_2 = 1 - \frac{\log 2}{\log\left(2 + \frac{6}{\sigma_2 - 2}\right)}, \quad \gamma_3 = \frac{\log\left(1 + 2\zeta(2/3, \delta)\right)}{\log\left(2 + \frac{6}{\sigma_2 - 2}\right)} \tag{26}$$

For $\sigma_2 \in (1, 3)$, define the following constants:

$$\begin{aligned} \gamma_4 &= \frac{\log\left(1 + 2\zeta(\frac{1}{12}(\sigma_2 + 5), \delta)\right)}{\log\left(\frac{12}{-\sigma_2 + 7} - 1\right)}, \quad \gamma_5 = 1 - \frac{\log\left(\frac{12}{-\sigma_2 + 7} - 1\right)}{\log\left(\frac{18}{\sigma_2 - 1} - 1\right)} \\ \gamma_6 &= \frac{\log\left(1 + 2\zeta(\frac{1}{12}(\sigma_2 + 5), \delta)\right)}{\log\left(\frac{18}{\sigma_2 - 1} - 1\right)}. \end{aligned}$$

Also, we will denote:

$$G = \frac{\Delta L_1}{1 + 4\log\left(1 + \frac{\Delta L_1^2}{L_0}\right)}, \tag{27}$$

which will be used in the following definition.

**Definition 1.** *For $p \in \left(\frac{1}{2}, \frac{\sigma_2}{\sigma_2+1}\right)$ and $\delta \in (0,1)$, we say that $\boldsymbol{g}_1, \boldsymbol{g}_2 \in \mathbb{R}^d$ forms a $(p,\delta)$-tricky pair with respect to the stepsize function $\alpha$ if all of the following conditions hold:*

$$\boldsymbol{g}_1 = c_1 \boldsymbol{g}, \quad \text{and} \quad \boldsymbol{g}_2 = c_2 \boldsymbol{g} \text{ for some } \boldsymbol{g} \in \mathbb{R}^d \text{ with } \|\boldsymbol{g}\| = 1 \tag{28}$$

$$sign(c_1) \neq sign(c_2) \tag{29}$$

$$|c_1| \geq \epsilon \quad and \quad |c_2| \geq \epsilon \tag{30}$$

$$|c_1| \leq \frac{1-p}{p}\sigma_1 + \left(\frac{1-p}{p}\sigma_2 - 1\right)G \tag{31}$$

$$|c_2| \geq \begin{cases} \frac{p|c_1|+\epsilon}{1-p} & |c_1| \leq \frac{1-p}{p}\sigma_1 + \left(\frac{1-p}{p}\sigma_2 - 1\right)\epsilon \\ \frac{(\sigma_2+1)p|c_1|-\sigma_1}{(\sigma_2+1)(1-p)-1} & |c_1| > \frac{1-p}{p}\sigma_1 + \left(\frac{1-p}{p}\sigma_2 - 1\right)\epsilon \end{cases} \tag{32}$$

$$|c_2| \leq \frac{p|c_1| + G}{1-p} \tag{33}$$

$$\frac{\alpha(\boldsymbol{g}_1)\|\boldsymbol{g}_1\|}{\alpha(\boldsymbol{g}_2)\|\boldsymbol{g}_2\|} \geq \lambda_0(p,\delta). \tag{34}$$

Notice that the lower bound of $|c_2|$ for the second case of Equation 32 is positive, since $p < \frac{\sigma_2}{\sigma_2+1}$. The significance of a tricky pair, as shown in Lemma 10, is that it can be used to construct an instance $(f, g, \mathcal{D}) \in \mathcal{F}_{\text{aff}}(\Delta, L_0, L_1, \sigma_1, \sigma_2)$ for which $A$ diverges with probability at least $\delta$.

Finally, let $\hat{P}_{\boldsymbol{y}}(\boldsymbol{x}) = \frac{\langle \boldsymbol{x}, \boldsymbol{y}\rangle}{\|\boldsymbol{y}\|}$, so that $\hat{P}_{\boldsymbol{y}}(\boldsymbol{x})$ denotes the component of $\boldsymbol{x}$ in the direction of $\frac{\boldsymbol{y}}{\|\boldsymbol{y}\|}$. Note the difference from the common notation $P_{\boldsymbol{y}}(\boldsymbol{x}) = \frac{\langle \boldsymbol{x}, \boldsymbol{y}\rangle}{\|\boldsymbol{y}\|^2}\boldsymbol{x}$.

### C.2 PROOFS

We now provide proofs of the lemmas mentioned in Section 6.1.

**Lemma 9.** *Suppose that there exists some $\boldsymbol{g} \in \mathbb{R}^d$ with $\|\boldsymbol{g}\| \in [\epsilon, \sigma_1 + (\sigma_2+1)\Delta L_1]$ and*

$$\alpha(\boldsymbol{g}) \leq 0, \quad or \quad \alpha(\boldsymbol{g}) \geq \frac{4}{L_1\|\boldsymbol{g}\|}\log\left(1 + \frac{L_1\min(\|\boldsymbol{g}\|, \Delta L_1)}{L_0}\right).$$

*Then there exists $(f_{\exp}, g_{\exp}, \mathcal{D}_{\exp}) \in \mathcal{F}_{\text{aff}}(\Delta, L_0, L_1, \sigma_1, \sigma_2)$ such that $\|\nabla f_{\exp}(\boldsymbol{x}_t)\| \geq \epsilon$ for all $t \geq 0$.*

*Proof.* We will construct $f : \mathbb{R}^d \to \mathbb{R}$ piecewise with linear and exponential pieces so that $\|\nabla f(x_t)\| = \min(\|\boldsymbol{g}\|, \Delta L_1) \geq \epsilon$ for all $t \geq 0$.

First, define $\tilde{\boldsymbol{g}} := \min(\|\boldsymbol{g}\|, \Delta L_1)\frac{\boldsymbol{g}}{\|\boldsymbol{g}\|}$, $m := \frac{1}{L_1}\log\left(1 + \frac{L_1\|\tilde{\boldsymbol{g}}\|}{L_0}\right)$. Recall the function $\psi : \mathbb{R} \to \mathbb{R}$ defined as

$$\psi(x) = \frac{L_0}{L_1^2}\left(\exp(L_1|x|) - L_1|x| - 1\right).$$

It is straightforward to verify that $\psi$ bounded from below by 0, continuously differentiable, $(L_0, L_1)$-smooth, and satisfies

$$\psi(-m) = \psi(m) = \frac{\|\tilde{\boldsymbol{g}}\|}{L_1} - \frac{L_0}{L_1^2}\log\left(1 + \frac{L_1\|\tilde{\boldsymbol{g}}\|}{L_0}\right) \leq \frac{\|\tilde{\boldsymbol{g}}\|}{L_1} \leq \frac{\Delta L_1}{L_1} \leq \Delta$$

$$|x| \leq m \implies |\psi'(x)| \leq \|\tilde{\boldsymbol{g}}\| \leq \Delta L_1$$

$$\psi'(-m) = -\|\tilde{\boldsymbol{g}}\|$$

$$\psi'(m) = \|\tilde{\boldsymbol{g}}\|.$$

The condition in the lemma statement gives two cases: $\alpha(\boldsymbol{g}) \leq 0$ or $\alpha(\boldsymbol{g}) \geq \frac{4m}{\|\boldsymbol{g}\|}$. We handle the two cases separately below.

**Case 1:** $\alpha(\boldsymbol{g}) \le 0$. This case is easy: the algorithm $A$ is essentially employing a negative learning rate! For a piecewise linear function that has a piece with gradient equal to $\boldsymbol{g}$, the trajectory $\{x_t\}$ moves away from the minimum indefinitely. To handle the case that $\|\boldsymbol{g}\| > \Delta L_1$, we instead use a gradient of $\tilde{\boldsymbol{g}} = \min(\|\boldsymbol{g}\|, \Delta L_1)\frac{\boldsymbol{g}}{\|\boldsymbol{g}\|} = \Delta L_1 \frac{\boldsymbol{g}}{\|\boldsymbol{g}\|}$, and construct a stochastic gradient that always returns either $\boldsymbol{g}$ or $\boldsymbol{0}$, so that each updated iterate $x_{t+1}$ either moves further from the minimum than $x_t$, or doesn't move at all.

Define the objective

$$f(\boldsymbol{x}) = \begin{cases} -\|\tilde{\boldsymbol{g}}\|(\hat{P}_{\boldsymbol{g}}(\boldsymbol{x}) + m) + \psi(m) & \hat{P}_{\boldsymbol{g}}(\boldsymbol{x}) < m \\ \psi(\hat{P}_{\boldsymbol{g}}(\boldsymbol{x})) & \hat{P}_{\boldsymbol{g}}(\boldsymbol{x}) \in [-m, m] \\ \|\tilde{\boldsymbol{g}}\|(\hat{P}_{\boldsymbol{g}}(\boldsymbol{x}) - m) + \psi(m) & \hat{P}_{\boldsymbol{g}}(\boldsymbol{x}) > m \end{cases}.$$

Notice that $f$ is bounded from below by $\psi(0) = 0$. Since $\psi'(m) = \|\tilde{\boldsymbol{g}}\|$ and $\psi'(-m) = -\|\tilde{\boldsymbol{g}}\|$, then $f$ is continuously differentiable. Since $\psi$ is $(L_0, L_1)$-smooth, so is $f$. Consider the initial point $\boldsymbol{x}_0 = m\boldsymbol{g}$. From the properties of $\psi$ from above, $f$ satisfies

$$f(\boldsymbol{x}_0) - f_* = \psi(m) \le \Delta.$$

and for all $\boldsymbol{x}$ with $f(\boldsymbol{x}) \le f(\boldsymbol{x}_0)$, it must be that $\hat{P}_{\boldsymbol{g}}(\boldsymbol{x}) \in [-m, m]$, so

$$\|\nabla f(\boldsymbol{x})\| = |\psi'(\hat{P}_{\boldsymbol{g}}(\boldsymbol{x}))| \le \Delta L_1. \tag{35}$$

Define a stochastic gradient $F$ for $f$ as follows:

$$F(x, \xi) = \begin{cases} (\|\boldsymbol{g}\|/\|\tilde{\boldsymbol{g}}\|)\nabla f(\boldsymbol{x}) & \xi = 0 \\ 0 & \xi = 1, \end{cases}$$

where $\xi \in \{0, 1\}$ has distribution $\mathcal{D}$, defined as $P(\xi = 0) = \|\tilde{\boldsymbol{g}}\|/\|\boldsymbol{g}\|$. Then $\mathbb{E}_\xi[F(\boldsymbol{x}, \xi)] = \nabla f(\boldsymbol{x})$. To see that this stochastic gradient satisfies the noise condition: If $\|\boldsymbol{g}\| \le \Delta L_1$, then $\tilde{\boldsymbol{g}} = \boldsymbol{g}$ and $P(\xi = 0) = 1$, so $F(\boldsymbol{x}; \xi) = \nabla f(\boldsymbol{x})$ almost surely. Otherwise,

$$\begin{aligned} \|F(\boldsymbol{x}; 0) - \nabla f(\boldsymbol{x})\| &= \left(\frac{\|\boldsymbol{g}\|}{\|\tilde{\boldsymbol{g}}\|} - 1\right)\|\nabla f(\boldsymbol{x})\| \\ &= \left(\frac{\|\boldsymbol{g}\|}{\Delta L_1} - 1\right)\|\nabla f(\boldsymbol{x})\| \\ &\overset{(i)}{\le} \left(\frac{\sigma_1 + (\sigma_2 + 1)\Delta L_1}{\Delta L_1} - 1\right)\|\nabla f(\boldsymbol{x})\| \\ &= \frac{\sigma_1 + \sigma_2 \Delta L_1}{\Delta L_1}\|\nabla f(\boldsymbol{x})\| \\ &= \frac{\|\nabla f(\boldsymbol{x})\|}{\Delta L_1}\sigma_1 + \sigma_2\|\nabla f(\boldsymbol{x})\| \\ &\overset{(ii)}{\le} \sigma_1 + \sigma_2\|\nabla f(\boldsymbol{x})\|, \end{aligned}$$

where $(i)$ uses the assumption $\|\boldsymbol{g}\| \le \sigma_1 + (\sigma_2 + 1)\Delta L_1$, and $(ii)$ uses Equation 35. Also

$$\|F(\boldsymbol{x}; 1) - \nabla f(\boldsymbol{x})\| = \|\nabla f(\boldsymbol{x})\| \le \sigma_1 + \sigma_2\|f(\boldsymbol{x})\|,$$

which uses the assumption $\sigma_2 > 1$. So the noise condition is satisfied, and therefore $(f, F, \mathcal{D}) \in \mathcal{F}_{\text{aff}}(\Delta, L_0, L_1, \sigma_1, \sigma_2)$.

Now consider the trajectory of $A$ from the initial point $\boldsymbol{x}_0 = m\boldsymbol{g}$. We claim that $\boldsymbol{x}_t = c_t\boldsymbol{g}$ for some $c_t \ge m$ for all $t \ge 0$. Clearly this holds for $t = 0$. Suppose it holds for some $t \ge 0$. Then $\|\nabla f(\boldsymbol{x}_t)\| = \tilde{\boldsymbol{g}}$. The stochastic gradient has two cases: if $\xi = 0$, then

$$F(\boldsymbol{x}_t, \xi) = (\|\boldsymbol{g}\|/\|\tilde{\boldsymbol{g}}\|)\nabla f(\boldsymbol{x}) = (\|\boldsymbol{g}\|/\|\tilde{\boldsymbol{g}}\|)\tilde{\boldsymbol{g}} = \boldsymbol{g},$$

so

$$\boldsymbol{x}_{t+1} = \boldsymbol{x}_t - \alpha(\boldsymbol{g})\boldsymbol{g} = (c_t - \alpha(\boldsymbol{g}))\boldsymbol{g} = c_{t+1}\boldsymbol{g},$$

and $c_{t+1} \ge c_t \ge m$ since $\alpha(\boldsymbol{g}) \le 0$. If $\xi = 1$, then $F(\boldsymbol{x}_t, \xi) = 0$, so $\boldsymbol{x}_{t+1} = \boldsymbol{x}_t = c_t\boldsymbol{g}$. Either way, $\boldsymbol{x}_{t+1} = c_{t+1}\boldsymbol{g}$ holds for some $c_{t+1} \ge m$, which completes the induction. Therefore $\|\nabla f(\boldsymbol{x}_t)\| = \|\tilde{\boldsymbol{g}}\| \ge \epsilon$ for all $t \ge 0$.

**Case 2:** $\alpha(\boldsymbol{g}) \geq \frac{4m}{\|\boldsymbol{g}\|}$. In this case, the learning rate $\alpha(\boldsymbol{g})$ is large enough to ensure that $f(\boldsymbol{x}_{t+1}) \geq f(\boldsymbol{x}_t)$ for an exponentially increasing $f$. By creating $f$ that only depends on $\langle \boldsymbol{x}, \boldsymbol{g} \rangle$ and which is piecewise linear and exponential in $\langle \boldsymbol{x}, \boldsymbol{g} \rangle$, this increase of the objective function continues indefinitely.

Define $m' = \alpha(\boldsymbol{g})\|\boldsymbol{g}\|$ and $\phi : [0, m']$ as

$$\phi(x) = \begin{cases} \psi(x - m) & x \in [0, 2m) \\ \|\tilde{\boldsymbol{g}}\|(x - 2m) + \psi(m) & x \in (2m, m' - 2m) \\ -\psi(x - (m' - m)) + \|\tilde{\boldsymbol{g}}\|(m' - 4m) + 2\psi(m) & x \in (m' - 2m, m'] \end{cases}$$

Note that the above definition makes sense since we assumed that $m' = \alpha(\boldsymbol{g})\|\boldsymbol{g}\| \geq 4m$, so $m' - 2m \geq 2m$. Again, $\phi$ is continuously differentiable, bounded from below, $(L_0, L_1)$-smooth, and satisfies

$$\begin{aligned} |\phi'(x)| &\leq \|\tilde{\boldsymbol{g}}\| \leq \Delta L_1 \text{ for all } x \in [0, m'] \\ \phi(x) &\geq 0 \\ \phi'(0) &= -\|\tilde{\boldsymbol{g}}\| \\ \phi'(m') &= -\|\tilde{\boldsymbol{g}}\|. \end{aligned}$$

Now, we can define the objective $f$ as follows:

$$f(\boldsymbol{x}) = \begin{cases} -\|\tilde{\boldsymbol{g}}\|\hat{P}_{\boldsymbol{g}}(\boldsymbol{x}) + \phi(0) & \hat{P}_{\boldsymbol{g}}(\boldsymbol{x}) \leq 0 \\ \phi(\hat{P}_{\boldsymbol{g}}(\boldsymbol{x}) - m'\lfloor \hat{P}_{\boldsymbol{g}}(\boldsymbol{x})/m' \rfloor) + \tilde{g}(m' - 4m)\lfloor \hat{P}_{\boldsymbol{w}}(\boldsymbol{x})/m' \rfloor & \hat{P}_{\boldsymbol{g}}(\boldsymbol{x}) > 0 \end{cases}$$

$f$ is continuous inside each "piece" (i.e. each region with $\hat{P}_{\boldsymbol{g}}(\boldsymbol{x}) \in (km', (k+1)m')$ for $k \in \mathbb{Z}_{\geq 0}$). Also, using $\phi(0) = 0$, $f$ is continuous at the boundary of each piece. Similarly, $f$ is continuously differentiable inside each piece, and using the fact that $\phi'(0) = -\|\tilde{\boldsymbol{g}}\| = \phi'(m')$, is continuously differentiable at the boundary of each piece. Also, $f$ is bounded below by $\min_{x \in [0, m']} \phi(x) = \min_x \psi(x) = 0$. So with the initial point $\boldsymbol{x}_0 = \boldsymbol{0}$, $f$ satisfies

$$f(\boldsymbol{x}_0) - f_* = \phi(0) - 0 = \psi(-m) \leq \Delta.$$

Since $\phi$ is $(L_0, L_1)$-smooth, so is $f$. Also, $\|\nabla f(\boldsymbol{x})\| \leq |\psi'(m)| = \|\tilde{\boldsymbol{g}}\| \leq \Delta L_1$ for every $x$.

Now we can define a stochastic gradient $F$ for $f$ as follows:

$$F(\boldsymbol{x}; \xi) = \begin{cases} (\|\boldsymbol{g}\|/\|\tilde{\boldsymbol{g}}\|) \nabla f(\boldsymbol{x}) & \xi = 0 \\ 0 & \xi = 1 \end{cases},$$

where $\xi \in \{0, 1\}$ has distribution $\mathcal{D}$, defined as $P(\xi = 0) = \|\tilde{\boldsymbol{g}}\|/\|\boldsymbol{g}\|$. This is the same stochastic gradient that we used in Case 1, and an identical argument shows that the noise conditions are satisfied. Therefore $(f, F, \mathcal{D}) \in \mathcal{F}_{\text{aff}}(\Delta, L_0, L_1, \sigma_1, \sigma_2)$.

Consider the execution of $A$ on $(f, F, \mathcal{D})$ from the initial point $x_0 = \boldsymbol{0}$. We claim that $\boldsymbol{x}_t$ is an integer multiple of $m'\frac{\boldsymbol{g}}{\|\boldsymbol{g}\|}$ for all $t \geq 0$, which we will show by induction. The base case $t = 0$ holds by construction. If $\boldsymbol{x}_t = -km'\frac{\boldsymbol{g}}{\|\boldsymbol{g}\|}$ for some $t \geq 0$, then there are two outcomes of the stochastic gradient. If $\xi_t = 1$, then $\boldsymbol{x}_{t+1} = \boldsymbol{x}_t = -km'\frac{\boldsymbol{g}}{\|\boldsymbol{g}\|}$. Otherwise $\xi_t = 0$, so

$$\begin{aligned} \boldsymbol{x}_{t+1} &= \boldsymbol{x}_t - \alpha(F(\boldsymbol{x}_t, 0))F(\boldsymbol{x}_t, 0) \\ &= -km'\frac{\boldsymbol{g}}{\|\boldsymbol{g}\|} - \alpha\left(\frac{\|\boldsymbol{g}\|}{\|\tilde{\boldsymbol{g}}\|}\nabla f(\boldsymbol{x}_t)\right)\frac{\|\boldsymbol{g}\|}{\|\tilde{\boldsymbol{g}}\|}\nabla f(\boldsymbol{x}_t) \\ &\overset{(i)}{=} -km'\frac{\boldsymbol{g}}{\|\boldsymbol{g}\|} - \alpha(\boldsymbol{g})\boldsymbol{g} \\ &= -km'\frac{\boldsymbol{g}}{\|\boldsymbol{g}\|} - \alpha(\boldsymbol{g})\|\boldsymbol{g}\|\frac{\boldsymbol{g}}{\|\boldsymbol{g}\|} \\ &= -km'\frac{\boldsymbol{g}}{\|\boldsymbol{g}\|} - m'\frac{\boldsymbol{g}}{\|\boldsymbol{g}\|} \\ &= -(k+1)m'\frac{\boldsymbol{g}}{\|\boldsymbol{g}\|}, \end{aligned}$$

where $(i)$ uses the fact that $\boldsymbol{x}_t = -km'\frac{\boldsymbol{g}}{\|\boldsymbol{g}\|} \implies \nabla f(\boldsymbol{x}_t) = \|\tilde{\boldsymbol{g}}\|\frac{\boldsymbol{g}}{\|\boldsymbol{g}\|}$. This completes the induction.

Therefore $\|\nabla f(\boldsymbol{x}_t)\| = \left\|\nabla f\left(-km'\frac{\boldsymbol{g}}{\|\boldsymbol{g}\|}\right)\right\| = \|\tilde{\boldsymbol{g}}\| \geq \epsilon$ for all $t$. $\qquad\square$

**Lemma 10.** *Suppose that*

$$0 < \alpha(\boldsymbol{g}) < \frac{4}{L_1\boldsymbol{g}}\log\left(1 + \frac{L_1\min(\|\boldsymbol{g}\|, \Delta L_1)}{L_0}\right), \tag{36}$$

*for all $\boldsymbol{g} \in \mathbb{R}^d$ with $\|\boldsymbol{g}\| \in [\epsilon, \sigma_1 + (\sigma_2+1)\Delta L_1]$, and suppose that there exist $\boldsymbol{g}_1, \boldsymbol{g}_2 \in \mathbb{R}$ which is a $(p, \delta)$-tricky pair with respect to $\alpha$. Then there exists $(f, g, \mathcal{D}) \in \mathcal{F}_{\mathrm{aff}}(\Delta, L_0, L_1, \sigma_1, \sigma_2)$ such that $\|\nabla f(\boldsymbol{x}_t)\| \geq \epsilon$ for all $t \geq 0$ with probability at least $\delta$.*

*Proof.* We will construct $(f, g, \mathcal{D})$ such that $f$ is a piecewise linear function, where one piece has stochastic gradient equal to $\boldsymbol{g}_1$ with probability $p$ and $\boldsymbol{g}_2$ with probability $1 - p$. Using the properties of a $(p, \delta)$-tricky pair, this instance is a member of $\mathcal{F}_{\mathrm{aff}}(\Delta, L_0, L_1, \sigma_1, \sigma_2)$, and $A$ will diverge with probability at least $\delta$ when optimizing this instance.

From the tricky pair definition, $\boldsymbol{g}_1 = c_1\boldsymbol{g}$ and $\boldsymbol{g}_2 = c_2\boldsymbol{g}$ for a unit vector $\boldsymbol{g}$. Without loss of generality, assume that $c_1 < 0$ and $c_2 > 0$. The following argument applies in the excluded case $c_1 > 0, c_2 < 0$ by replacing the objective $f(x)$ below with $f(-x)$. Denote $\ell = pc_1 + (1-p)c_2$ and $a = \frac{1}{L_1}\log\left(1 + \frac{L_1\ell}{L_0}\right)$, and define $f : \mathbb{R}^d \to \mathbb{R}$ as

$$f(x) = \begin{cases} -\ell\left(\hat{P}_{\boldsymbol{g}}(\boldsymbol{x}) + a\right) + \psi(a) & \hat{P}_{\boldsymbol{g}}(\boldsymbol{x}) \leq -a \\ \psi(\hat{P}_{\boldsymbol{g}}(\boldsymbol{x})) & \hat{P}_{\boldsymbol{g}}(\boldsymbol{x}) \in (-a, a) \\ \ell\left(\hat{P}_{\boldsymbol{g}}(\boldsymbol{x}) - a\right) + \psi(a) & \hat{P}_{\boldsymbol{g}}(\boldsymbol{x}) \geq a \end{cases}$$

where $\psi$ is as defined in Lemma 1. Notice that $f$ is continuously differentiable, bounded from below by $f_* = 0$, and $(L_0, L_1)$-smooth.

Next, set $\Xi = \{0, 1\}$ and define $F : \mathbb{R}^d \times \Xi \to \mathbb{R}^d$ as

$$F(\boldsymbol{x}, \xi) = \begin{cases} -\boldsymbol{g}_1 & \hat{P}_{\boldsymbol{g}}(\boldsymbol{x}) \leq -a \text{ and } \xi = 0 \\ -\boldsymbol{g}_2 & \hat{P}_{\boldsymbol{g}}(\boldsymbol{x}) \leq -a \text{ and } \xi = 1 \\ \psi'(x) & \hat{P}_{\boldsymbol{g}}(\boldsymbol{x}) \in (-a, a) \\ \boldsymbol{g}_1 & \hat{P}_{\boldsymbol{g}}(\boldsymbol{x}) \geq a \text{ and } \xi = 0 \\ \boldsymbol{g}_2 & \hat{P}_{\boldsymbol{g}}(\boldsymbol{x}) \geq a \text{ and } \xi = 1 \end{cases}$$

and define the distribution $\mathcal{D}$ over $\Xi$ as

$$\xi = \begin{cases} 0 & \text{with probability } p \\ 1 & \text{with probability } 1 - p \end{cases}$$

for $\xi \sim \mathcal{D}$. Notice that $F(x, \xi) = \nabla f(\boldsymbol{x})$ for $\boldsymbol{x}$ with $\hat{P}_{\boldsymbol{g}}(\boldsymbol{x}) \in (-a, a)$. Also, $\mathbb{E}_{\xi\sim\mathcal{D}}[F(\boldsymbol{x}, \xi)] = p\boldsymbol{g}_1 + (1-p)\boldsymbol{g}_2 = \ell\boldsymbol{g} = \nabla f(\boldsymbol{x})$ for $\boldsymbol{x}$ with $\hat{P}_{\boldsymbol{g}}(\boldsymbol{x}) \geq a$, and similarly for $\hat{P}_{\boldsymbol{g}}(\boldsymbol{x}) \leq -a$. Using the fact that $\boldsymbol{g}_1, \boldsymbol{g}_2$ is a $p$-tricky pair, we have for all $x$:

$$\|\nabla f(\boldsymbol{x})\| \leq \ell = pc_1 + (1-p)c_2 = -p|c_1| + (1-p)|c_2| \overset{(i)}{\leq} -p|c_1| + (1-p)\left(\frac{p|c_1| + G}{1-p}\right) = G. \tag{37}$$

where $(i)$ uses Equation 33. We can also use the tricky pair properties to show that $(f, g, \mathcal{D})$ satisfies $\ell \geq \epsilon$ and the noise condition, depending on the two cases in Equation 32. In the first case,

$$|c_1| \leq \frac{1-p}{p}\sigma_1 + \left(\frac{1-p}{p}\sigma_2 - 1\right)\epsilon \tag{38}$$

$$|c_2| \geq \frac{p|c_1| + \epsilon}{1-p}, \tag{39}$$

so

$$\ell = (1-p)|c_2| + p(-|c_1|) \geq \epsilon,$$

and

$$
\begin{aligned}
\|\boldsymbol{g}_2 - \ell \boldsymbol{g}\| &= |c_2 - \ell| \|\boldsymbol{g}\| \\
&= c_2 - \ell \\
&= \frac{p}{1-p}(\ell - c_1) \\
&\overset{(i)}{\le} \frac{p}{1-p}\ell + \frac{p}{1-p}\left(\frac{1-p}{p}\sigma_1 + \left(\frac{1-p}{p}\sigma_2 - 1\right)\epsilon\right) \\
&= \frac{p}{1-p}\ell + \sigma_1 + \left(\sigma_2 - \frac{p}{1-p}\right)\epsilon \\
&\overset{(ii)}{\le} \frac{p}{1-p}\ell + \sigma_1 + \left(\sigma_2 - \frac{p}{1-p}\right)\ell \\
&= \sigma_1 + \sigma_2\ell,
\end{aligned}
$$

where $(i)$ uses Equation 38 and $(ii)$ uses $\ell \ge \epsilon$. Also,

$$
\|\boldsymbol{g}_1 - \ell \boldsymbol{g}\| = |c_1 - \ell| \|\boldsymbol{g}\| = \ell - c_1 = \frac{1-p}{p}(c_2 - \ell) \le \frac{1-p}{p}(\sigma_1 + \sigma_2\ell) \le \sigma_1 + \sigma_2\ell,
$$

where the last inequality uses $p > \frac{1}{2}$. Therefore $(f, g, \mathcal{D})$ satisfies $\ell \ge \epsilon$ and the noise condition in the first case. In the second case,

$$
|c_1| > \frac{1-p}{p}\sigma_1 + \left(\frac{1-p}{p}\sigma_2 - 1\right)\epsilon \tag{40}
$$

$$
|c_2| \ge \frac{(\sigma_2 + 1)p|c_1| - \sigma_1}{(\sigma_2 + 1)(1-p) - 1}, \tag{41}
$$

so

$$
\begin{aligned}
c_2 &\ge \frac{(\sigma_2 + 1)p(-c_1) - \sigma_1}{(\sigma_2 + 1)(1-p) - 1} \\
((\sigma_2 + 1)(1-p) - 1)c_2 &\ge (\sigma_2 + 1)p(-c_1) - \sigma_1 \\
c_2 &\le \sigma_1 + (\sigma_2 + 1)pc_1 + (\sigma_2 + 1)(1-p)c_2 \\
c_2 &\le \sigma_1 + (\sigma_2 + 1)\ell \\
c_2 - \ell &\le \sigma_1 + \sigma_2\ell,
\end{aligned}
$$

and

$$
\begin{aligned}
c_2 &\ge \frac{(\sigma_2 + 1)p|c_1| - \sigma_1}{(\sigma_2 + 1)(1-p) - 1} \\
&= \frac{p}{1-p}|c_1| + \left(\frac{\sigma_2 + 1}{(\sigma_2 + 1)(1-p) - 1} - \frac{1}{1-p}\right)p|c_1| - \frac{\sigma_1}{(\sigma_2 + 1)(1-p) - 1} \\
&= \frac{p}{1-p}|c_1| + \frac{1}{((\sigma_2 + 1)(1-p) - 1)(1-p)}p|c_1| - \frac{\sigma_1}{(\sigma_2 + 1)(1-p) - 1} \\
&\overset{(i)}{\ge} \frac{p}{1-p}|c_1| + \frac{(1-p)\sigma_1 + ((1-p)\sigma_2 - p)\epsilon}{((\sigma_2 + 1)(1-p) - 1)(1-p)} - \frac{\sigma_1}{(\sigma_2 + 1)(1-p) - 1} \\
&\ge \frac{p}{1-p}|c_1| + \frac{\epsilon}{1-p},
\end{aligned}
$$

where $(i)$ uses Equation 40. Therefore $\ell = pc_1 + (1-p)c_2 \ge \epsilon$ as in the first case. Also as in the first case, $|c_1 - \ell| \le |c_2 - \ell|$. Therefore $(f, g, \mathcal{D})$ satisfies $\ell \ge \epsilon$ and the noise condition in the second case.

Consider the initial point $x_0 = (a + \alpha(\boldsymbol{g}_2)\|\boldsymbol{g}_2\|)\boldsymbol{g}$. Recall that

$$
\|\boldsymbol{g}_2\| = |c_2| \le |\ell| + |c_2 - \ell| \le \sigma_1 + (\sigma_2 + 1)\ell \overset{(i)}{\le} \sigma_1 + (\sigma_2 + 1)G \overset{(ii)}{\le} \sigma_1 + (\sigma_2 + 1)\Delta L_1,
$$

where $(i)$ uses Equation 37 and $(ii)$ uses the definition of $G$ (Equation 27). Also, by the tricky pair definition, $\|g_2\| = |c_2| \geq \epsilon$. Therefore $\|g_2\| \in [\epsilon, \sigma_1 + (\sigma_2 + 1)\Delta L_1]$, so we can use Equation 36 to conclude that

$$0 < \alpha(g_2) < \frac{4}{L_1\|g_2\|} \log\left(1 + \frac{\Delta L_1^2}{L_0}\right).$$

Therefore, $f$ satisfies

$$
\begin{aligned}
f(x_0) - f^* &= \psi(a) + \ell\alpha(g_2)\|g_2\| \\
&= \frac{\ell}{L_1} - \frac{L_0}{L_1^2} \log\left(1 + \frac{L_1\ell}{L_0}\right) + \ell\alpha(g_2)\|g_2\| \\
&\leq \frac{\ell}{L_1} + \ell\alpha(g_2)\|g_2\| \\
&\leq \frac{\ell}{L_1} + \frac{4\ell}{L_1} \log\left(1 + \frac{\Delta L_1^2}{L_0}\right) \\
&= \frac{\ell}{L_1}\left(1 + 4\log\left(1 + \frac{\Delta L_1^2}{L_0}\right)\right) \\
&\overset{(i)}{\leq} \frac{\Delta L_1}{1 + 4\log\left(1 + \frac{\Delta L_1^2}{L_0}\right)} \frac{1}{L_1}\left(1 + 4\log\left(1 + \frac{\Delta L_1^2}{L_0}\right)\right) \\
&= \Delta,
\end{aligned}
$$

where $(i)$ uses Equation 37. This shows that $(f, g, \mathcal{D}) \in \mathcal{F}_{\mathrm{aff}}(\Delta, L_0, L_1, \sigma_1, \sigma_2)$.

We now claim that $\|\nabla f(x_t)\| \geq \epsilon$ for all $t \geq 0$ with probability $\delta$ when $A$ is initialized with $x_0 = (a + \alpha(g_2)\|g_2\|)g$. To see this, consider the sequence

$$
y_t = \begin{cases} \frac{1}{\alpha(g_2)\|g_2\|}\left(\langle x_t, g\rangle - a\right) & \langle x_i, g\rangle \geq a \text{ for all } i \leq t \\ 0 & \text{otherwise} \end{cases}.
$$

As long as $\langle x_t, g\rangle > a$, the sequence $y_t$ follows the exact same distribution as the random walk in Equation 24 with $\lambda = \frac{\alpha(g_1)\|g_1\|}{\alpha(g_2)\|g_2\|} > 0$. Since $g_1, g_2$ is a $(p, \delta)$-tricky pair, $\lambda \geq \lambda_0(p, \delta)$, so that $z_{p,\lambda} \leq 1 - \delta$ by the tricky pair definition. Therefore

$$
\begin{aligned}
P\left(\|\nabla f(x_t)\| \geq \epsilon \text{ for all } t \geq 0\right) &\geq P\left(\langle x_t, g\rangle > a \text{ for all } t \geq 0\right) \\
&= P\left(y_t > 0 \text{ for all } t \geq 0\right) \\
&= 1 - z_{p,\lambda} \\
&\geq \delta.
\end{aligned}
$$

$\square$

**Lemma 11.** *Suppose that*

$$0 < \alpha(g) < \frac{4}{L_1\|g\|} \log\left(1 + \frac{L_1 \min(\|g\|, \Delta L_1)}{L_0}\right), \tag{42}$$

*for all $g \in \mathbb{R}^d$ with $\|g\| \in [\epsilon, \sigma_1 + (\sigma_2 + 1)\Delta L_1]$, and that there do not exist any $(p, \delta)$-tricky pairs with respect to $\alpha$. Suppose $g \in \mathbb{R}^d$ with $\|g\| = \epsilon$. If $\sigma_2 \geq 3$, then*

$$\alpha(g) \leq \tilde{\mathcal{O}}\left(\frac{1}{L_1(\Delta L_1)^{1-\gamma_2-\gamma_3}\epsilon^{\gamma_1}\sigma_1^{\gamma_2+\gamma_3-\gamma_1}}\right).$$

*On the other hand, if $\sigma_2 \in (1, 3)$, then*

$$\alpha(g) \leq \tilde{\mathcal{O}}\left(\frac{1}{(\sigma_2 - 1)^{2-\gamma_4-\gamma_5-\gamma_6}\epsilon^{\gamma_4}L_1(\Delta L_1)^{1-\gamma_5-\gamma_6}\sigma_1^{\gamma_5+\gamma_6-\gamma_4}}\right).$$

*Proof.* Different from the proof sketch in Section 6.1, in our actual construction below, we use two sequences $\{x_i\}$ and $\{y_i\}$ instead of one sequence $\{z_i\}$. Every $x$ in the sequence $\{x_i\}$ satisfies

$|\hat{P}_{\boldsymbol{g}}(\boldsymbol{x})| \in [\epsilon, \sigma_1 + (\sigma_2 - 1)\epsilon]$, and every $\boldsymbol{y}$ in $\{\boldsymbol{y}_i\}$ satisfies $|\hat{P}_{\boldsymbol{g}}(\boldsymbol{x})| \in [\sigma_1, +(\sigma_2 - 1)\epsilon, \sigma_1 + (\sigma_2 - 1)G]$ (see Equation 27 for the definition of $G$).

Denote $\beta(\boldsymbol{x}) = \alpha(\boldsymbol{x})\|\boldsymbol{x}\|$, fix any $p_0 \in \left(\frac{1}{2}, \frac{\sigma_2}{\sigma_2 + 1}\right)$ and define a sequence $\{\boldsymbol{x}_i\}_{i=0}^{\infty}$ as follows:

$$\boldsymbol{x}_0 = \boldsymbol{g}$$

$$\boldsymbol{x}_i = (-1)^i \frac{p_0\|\boldsymbol{x}_{i-1}\| + \epsilon}{1 - p_0} \frac{\boldsymbol{g}}{\|\boldsymbol{g}\|}.$$

Also, denote $k_0 = \max\left\{i \geq 0 : \|\boldsymbol{x}_i\| \leq \frac{1-p_0}{p_0}\sigma_1 + \left(\frac{1-p_0}{p_0}\sigma_2 - 1\right)\epsilon\right\}$. We claim that, for each $i$ with $0 \leq i \leq k_0$, the pair $(\boldsymbol{x}_i, \boldsymbol{x}_{i+1})$ satisfies all of the conditions of a $(p_0, \delta)$-tricky pair, other than possibly Equation 34. Notice that $\|\boldsymbol{x}_i\|$ is increasing and $\langle \boldsymbol{x}_i, \boldsymbol{g} \rangle$ has alternating sign, so Equation 29 and Equation 30 are satisfied. Recall that $\|\boldsymbol{x}_i\| \leq \frac{1-p_0}{p_0}\sigma_1 + \left(\frac{1-p_0}{p_0}\sigma_2 - 1\right)\epsilon$ by the definition of $k_0$. Since $\epsilon \leq G$ was assumed in Theorem 4, this implies $\|\boldsymbol{x}_i\| \leq \frac{1-p_0}{p_0}\sigma_1 + \left(\frac{1-p_0}{p_0}\sigma_2 - 1\right)G$. So Equation 31 is satisfied. Since $\|\boldsymbol{x}_i\| \leq \frac{1-p_0}{p_0}\sigma_1 + \left(\frac{1-p_0}{p_0}\sigma_2 - 1\right)\epsilon$, we must fulfill the first branch of the RHS of Equation 32. This only requires $\|\boldsymbol{x}_{i+1}\| \geq \frac{p_0\|vx_i\| + \epsilon}{1 - p_0}$, which holds by construction of the sequence $\{\boldsymbol{x}_i\}$. Finally, Equation 33 is satisfied again from $\epsilon \leq G$, since

$$\|\boldsymbol{x}_{i+1}\| = \frac{p_0\|\boldsymbol{x}_i\| + \epsilon}{1 - p_0} \leq \frac{p_0\|\boldsymbol{x}_i\| + G}{1 - p_0}.$$

This verifies the claim that the pair $(\boldsymbol{x}_i, \boldsymbol{x}_{i+1})$ satisfies Equation 29 through Equation 33. If $(\boldsymbol{x}_i, \boldsymbol{x}_{i+1})$ also satisfied Equation 34, then it would be a $(p_0, \delta)$-tricky pair. Since it was assumed that there do not exist any $(p, \delta)$-tricky pairs, it must be that Equation 34 is not satisfied by $(\boldsymbol{x}_i, \boldsymbol{x}_{i+1})$, so that

$$\beta(\boldsymbol{x}_i) \leq \lambda_0(p_0, \delta)\beta(\boldsymbol{x}_{i+1})$$

for all $0 \leq i \leq k_0$. Choosing $i = 0$ and unrolling to $i = k_0 - 2$:

$$\beta(\boldsymbol{x}_0) \leq (\lambda_0(p_0, \delta))^{k_0 - 1}\beta(\boldsymbol{x}_{k_0 - 1}). \tag{43}$$

Now choose $\boldsymbol{y}_0 = (-1)^{k_0}(\sigma_1 + (\sigma_2 - 1)\epsilon)\frac{\boldsymbol{g}}{\|\boldsymbol{g}\|}$. Then $\|\boldsymbol{y}_0\| \geq \|\boldsymbol{x}_{k_0}\|$ from the definition of $k_0$. We again want to show that $(\boldsymbol{x}_{k_0 - 1}, \boldsymbol{y}_0)$ satisfies Equation 29 through Equation 33. We can use an identical argument as above to demonstrate Equation 29 through Equation 32, so it only remains to show Equation 33. It was assumed in the statement of Theorem 4 that $\sigma_1 + (\sigma_2 - 1)\epsilon \leq G$. Therefore

$$\|\boldsymbol{y}_0\| = \sigma_1 + (\sigma_2 - 1)\epsilon \leq \Delta L_1 \leq \frac{p\|\boldsymbol{x}_{k_0 - 1}\| + G}{1 - p},$$

which demonstrates Equation 33. This verifies the claim for $(\boldsymbol{x}_{k_0 - 1}, \boldsymbol{y}_0)$. Again, Equation 34 would imply that $(\boldsymbol{x}_{k_0 - 1}, \boldsymbol{y}_0)$ is a $(p_0, \delta)$-tricky pair. But we assumed there are none, so Equation 34 cannot be satisfied. Therefore

$$\beta(\boldsymbol{x}_{k_0 - 1}) \leq \lambda_0(p_0, \delta)\beta(\boldsymbol{y}_0),$$

and combining with Equation 43 yields

$$\beta(\boldsymbol{x}_0) \leq (\lambda_0(p_0, \delta))^{k_0}\beta(\boldsymbol{y}_0). \tag{44}$$

Now fix some $p_1 \in \left(\frac{1}{2}, \frac{\sigma_2}{\sigma_2 + 1}\right)$, define the sequence $\{\boldsymbol{y}_i\}_{i=0}^{\infty}$ as:

$$\boldsymbol{y}_i = (-1)^{k_0 + i} \frac{(\sigma_2 + 1)p_1\|\boldsymbol{y}_{i-1}\| - \sigma_1}{(\sigma_2 + 1)(1 - p_1) - 1} \frac{\boldsymbol{g}}{\|\boldsymbol{g}\|}.$$

Denote $k_1 = \max\left\{i \geq 0 : \|\boldsymbol{y}_i\| \leq \frac{1-p_1}{p_1}\sigma_1 + \left(\frac{1-p_1}{p_1}\sigma_2 - 1\right)G\right\}$. Similarly as for the sequence $\{\boldsymbol{x}_i\}$, we claim that for each $i$ with $0 \leq i \leq k_1$, the pair $(\boldsymbol{y}_i, \boldsymbol{y}_{i+1})$ satisfies all of the conditions of a $(p_1, \delta)$-tricky pair, other than possibly Equation 34. Equation 29 and Equation 30 are satisfied, since $\|\boldsymbol{y}_i\|$ is increasing and $\langle \boldsymbol{y}_i, \boldsymbol{g} \rangle$ alternates in sign. The upper bound of $\|\boldsymbol{y}_i\|$ in the definition of $k_1$ ensures that Equation 31 is satisfied. Since

$$\|\boldsymbol{y}_i\| \geq \|\boldsymbol{y}_0| = \sigma_1 + (\sigma_2 - 1)\epsilon \geq \frac{1 - p_1}{p_1}\sigma_1 + \left(\frac{1 - p_1}{p_1}\sigma_2 - 1\right)\epsilon,$$

we must fulfill the second branch of the RHS of Equation 32. This only requires

$$\|\boldsymbol{y}_{i+1}\| \geq \frac{(\sigma_2 + 1)p_1\|\boldsymbol{y}_i\| - \sigma_1}{(\sigma_2 + 1)(1 - p_1) - 1},$$

which holds by construction of the sequence $\{\boldsymbol{y}_i\}$. Finally, to show Equation 33, we need

$$\|\boldsymbol{y}_{i+1}\| \leq \frac{p_1\|\boldsymbol{y}_i\| + G}{1 - p_1},$$

which is equivalent to

$$\frac{(\sigma_2 + 1)p_1\|\boldsymbol{y}_{i-1}\| - \sigma_1}{(\sigma_2 + 1)(1 - p_1) - 1} \leq \frac{p_1\|\boldsymbol{y}_i\| + G}{1 - p_1}$$

$$(1 - p_1)(\sigma_2 + 1)p_1\|\boldsymbol{y}_{i-1}\| - (1 - p_1)\sigma_1 \leq ((\sigma_2 + 1)(1 - p_1) - 1)p_1\|\boldsymbol{y}_i\| + ((\sigma_2 + 1)(1 - p_1) - 1)G$$

$$p_1\|\boldsymbol{y}_i\| \leq (1 - p_1)\sigma_1 + ((\sigma_2 + 1)(1 - p_1) - 1)G$$

$$\|\boldsymbol{y}_i\| \leq \frac{1 - p_1}{p_1}\sigma_1 + \frac{(\sigma_2 + 1)(1 - p_1) - 1}{p_1}G$$

$$\|\boldsymbol{y}_i\| \leq \frac{1 - p_1}{p_1}\sigma_1 + \left(\frac{1 - p_1}{p_1}\sigma_2 - 1\right)G.$$

All steps in this sequence are reversible, and the last inequality holds by the upper bound of $\|\boldsymbol{y}_i\|$ in the definition of $k_1$. Therefore, Equation 33 is satisfied. This verifies the claim that $(\boldsymbol{y}_i, \boldsymbol{y}_{i+1})$ satisfies all of the conditions of a $(p_1, \delta)$-tricky pair, other than possibly Equation 34. Again, Equation 34 cannot hold, since this would imply the existence of a $(p_1, \delta)$-tricky pair, and we have already assumed otherwise. Therefore

$$\beta(\boldsymbol{y}_i) \leq \lambda_0(p_1, \delta)\beta(\boldsymbol{y}_{i+1})$$

for all $0 \leq i \leq k_1$. Unrolling from $i = 0$ to $i = k_1 - 1$ yields

$$\beta(\boldsymbol{y}_0) \leq (\lambda_0(p_1, \delta))^{k_1} \beta(\boldsymbol{y}_{k_1}). \tag{45}$$

Combining Equation 44 and Equation 45 yields

$$\beta(\boldsymbol{x}_0) \leq (\lambda_0(p_0, \delta))^{k_0} (\lambda_0(p_1, \delta))^{k_1} \beta(\boldsymbol{y}_{k_1}). \tag{46}$$

We can use Lemma 16 for the sequences $\{\|\boldsymbol{x}_i\|\}_i$ and $\{\|\boldsymbol{y}_i\|\}_i$ to lower bound $k_0$ and $k_1$. For $k_0$, we apply Lemma 16 with

$$a_0 = \epsilon, \quad r = \frac{p_0}{1 - p_0}, \quad b = \frac{\epsilon}{1 - p_0}, \quad A = \frac{1 - p_0}{p_0}\sigma_1 + \left(\frac{1 - p_0}{p_0}\sigma_2 - 1\right)\epsilon.$$

Then

$$\frac{A(r - 1) + b}{a_0(r - 1) + b} = \frac{\left(\frac{1-p_0}{p_0}\sigma_1 + \left(\frac{1-p_0}{p_0}\sigma_2 - 1\right)\epsilon\right)\frac{2p_0-1}{1-p_0} + \frac{\epsilon}{1-p_0}}{\epsilon\frac{2p_0-1}{1-p_0} + \frac{\epsilon}{1-p_0}}$$

$$= \frac{\left(\frac{1-p_0}{p_0}\sigma_1 + \left(\frac{1-p_0}{p_0}\sigma_2 - 1\right)\epsilon\right)(2p_0 - 1) + \epsilon}{\epsilon(2p_0 - 1) + \epsilon}$$

$$= \frac{\frac{1-p_0}{p_0}\left(\sigma_1 + \left(\sigma_2 - \frac{p_0}{1-p_0}\right)\epsilon\right)(2p_0 - 1) + \epsilon}{2p_0\epsilon}$$

$$= \frac{\frac{(2p_0-1)(1-p_0)}{p_0}(\sigma_1 + \sigma_2\epsilon) + 2(1 - p_0)\epsilon}{2p_0\epsilon}$$

$$= \frac{(2p_0 - 1)(1 - p_0)}{2p_0^2\epsilon}(\sigma_1 + \sigma_2\epsilon) + \frac{1 - p_0}{p_0},$$

so Lemma 16 implies

$$k_0 = \left\lceil \frac{\log\left(\frac{(2p_0-1)(1-p_0)}{2p_0^2\epsilon}(\sigma_1+\sigma_2\epsilon)+\frac{1-p_0}{p_0}\right)}{\log\frac{p_0}{1-p_0}} \right\rceil$$

$$\geq \frac{\log\left(\frac{(2p_0-1)(1-p_0)}{2p_0^2\epsilon}(\sigma_1+\sigma_2\epsilon)+\frac{1-p_0}{p_0}\right)}{\log\frac{p_0}{1-p_0}} - 1$$

$$= \frac{\log b_0}{\log\frac{p_0}{1-p_0}} - 1, \tag{47}$$

where we denoted

$$b_0 = \frac{(2p_0-1)(1-p_0)}{2p_0^2\epsilon}(\sigma_1+\sigma_2\epsilon)+\frac{1-p_0}{p_0}.$$

Similarly, for $k_1$, we apply Lemma 16 with

$$a_0 = \sigma_1 + (\sigma_2-1)\epsilon, \quad r = \frac{(\sigma_2+1)p_1}{(\sigma_2+1)(1-p_1)-1}, \quad b = -\frac{\sigma_1}{(\sigma_2+1)(1-p_1)-1}$$

$$A = \frac{1-p_1}{p_1}\sigma_1 + \left(\frac{1-p_1}{p_1}\sigma_2-1\right)G = \frac{1-p_1}{p_1}\left(\sigma_1+\left(\sigma_2-\frac{p_1}{1-p_1}\right)G\right).$$

Then

$$r - 1 = \frac{(\sigma_2+1)p_1}{(\sigma_2+1)(1-p_1)-1} - 1 = \frac{(\sigma_2+1)(2p_1-1)+1}{(\sigma_2+1)(1-p_1)-1},$$

so

$$A(r-1) + b = \frac{\frac{1-p_1}{p_1}\left(\sigma_1+\left(\sigma_2-\frac{p_1}{1-p_1}\right)G\right)((\sigma_2+1)(2p_1-1)+1)-\sigma_1}{(\sigma_2+1)(1-p_1)-1},$$

and

$$a_0(r-1) + b = \frac{(\sigma_1+(\sigma_2-1)\epsilon)((\sigma_2+1)(2p_1-1)+1)-\sigma_1}{(\sigma_2+1)(1-p_1)-1}.$$

So

$$\frac{A(r-1)+b}{a_0(r-1)+b} = \frac{\frac{1-p_1}{p_1}\left(\sigma_1+\left(\sigma_2-\frac{p_1}{1-p_1}\right)G\right)((\sigma_2+1)(2p_1-1)+1)-\sigma_1}{(\sigma_1+(\sigma_2-1)\epsilon)((\sigma_2+1)(2p_1-1)+1)-\sigma_1}.$$

Denoting the RHS as $b_1$, this yields

$$k_1 = \left\lceil \frac{\log b_1}{\log\frac{(\sigma_2+1)p_1}{(\sigma_2+1)(1-p_1)-1}} \right\rceil \geq \frac{\log b_1}{\log\frac{(\sigma_2+1)p_1}{(\sigma_2+1)(1-p_1)-1}} - 1. \tag{48}$$

Plugging Equation 47 and Equation 48 into Equation 46:

$$\beta(\boldsymbol{x}_0) \leq (\lambda_0(p_0,\delta))^{-1}(\lambda_0(p_1,\delta))^{-1}(\lambda_0(p_0,\delta))^{\frac{\log b_0}{\log\frac{p_0}{1-p_0}}}(\lambda_0(p_1,\delta))^{\frac{\log b_1}{\log\frac{(\sigma_2+1)p_1}{(\sigma_2+1)(1-p_1)-1}}}\beta(\boldsymbol{y}_{k_1}).$$

Using the fact that for any $\rho$,

$$(\lambda_0(p_0,\delta))^{\log\rho} = (\lambda_0(p_0,\delta))^{\frac{\log\rho}{\log\lambda_0(p_0,\delta)}\log\lambda_0(p_0,\delta)} = \rho^{\log\lambda_0(p_0,\delta)},$$

we can choose $\rho = \log b_0$ and $\rho = \log b_1$,

$$\beta(\boldsymbol{x}_0) \leq (\lambda_0(p_0,\delta))^{-1}(\lambda_0(p_1,\delta))^{-1}\left(\frac{1}{b_0}\right)^{\phi_0}\left(\frac{1}{b_1}\right)^{\phi_1}\beta(\boldsymbol{y}_{k_1}), \tag{49}$$

where

$$\phi_0 = \log\frac{1}{\lambda_0(p_0,\delta)}\bigg/\log\frac{p_0}{1-p_0}$$

$$\phi_1 = \log\frac{1}{\lambda_0(p_1,\delta)}\bigg/\log\frac{(\sigma_2+1)p_1}{(\sigma_2+1)(1-p_1)-1}.$$

Note that $\phi_0 > \phi_1$, and denote $m = \frac{4}{L_1} \log \left(1 + \frac{\Delta L_1^2}{L_0}\right)$. We can also bound $\beta(\boldsymbol{y}_{k_1})$ using the assumed condition $\alpha(g) < \frac{4m}{|g|}$, since we previously showed that $(\boldsymbol{y}_{k_1}, \boldsymbol{y}_{k+1})$ satisfies Equation 29 through Equation 33. In particular, Equation 31 implies that

$$\|\boldsymbol{y}_{k_1}\| \leq \frac{1 - p_1}{p_1} \sigma_1 + \left(\frac{1 - p_1}{p_1} \sigma_2 - 1\right) G \leq \sigma_1 + (\sigma_2 + 1)G \leq \sigma_1 + (\sigma_2 + 1)\Delta L_1,$$

so that $\|\boldsymbol{y}_{k_1}\|$ falls within the range for which the bound on $\alpha(g)$ applies. Therefore $\alpha(\boldsymbol{y}_{k_1}) \leq \frac{4m}{\|y_{k_1}\|}$, or $\beta(\boldsymbol{y}_{k_1}) \leq 4m$. Plugging back to Equation 49 yields

$$\beta(\boldsymbol{x}_0) \leq 4m \left(\lambda_0(p_0, \delta)\right)^{-1} \left(\lambda_0(p_1, \delta)\right)^{-1} \left(\frac{1}{b_0}\right)^{\phi_0} \left(\frac{1}{b_1}\right)^{\phi_1}. \tag{50}$$

It only remains to choose $p_0$ and $p_1$ such that $b_0, b_1, \phi_0$, and $\phi_1$ can be bounded in terms of the problem parameters. First, using Lemma 15, we can rewrite $\lambda_0(p, \delta)$ as

$$\begin{aligned}
\lambda_0(p, \delta) &= \lambda_0(p, 0) + (\lambda_0(p, \delta) - \lambda_0(p, 0)) \\
&= \frac{1 - p}{p} + (\lambda_0(p, \delta) - \lambda_0(p, 0)) \\
&= \frac{1 - p}{p} + \zeta(p, \delta),
\end{aligned}$$

so that

$$\frac{1}{\lambda_0(p, \delta)} = \frac{p}{1 - p + p\zeta(p, \delta)} = \frac{p}{1 - p} \frac{1 - p}{1 - p + p\zeta(p, \delta)}.$$

We can then rewrite $\phi_0$ as

$$\begin{aligned}
\phi_0 &= \left(\log \frac{p_0}{1 - p_0} + \log \left(\frac{1 - p_0}{1 - p_0 + p_0 \zeta(p_0, \delta)}\right)\right) / \log \frac{p_0}{1 - p_0} \\
&= 1 - \frac{\log \left(\frac{1 - p_0 + p_0 \zeta(p_0, \delta)}{1 - p_0}\right)}{\log \frac{p_0}{1 - p_0}}
\end{aligned}$$

and $\phi_1$ as

$$\begin{aligned}
\phi_1 &= \left(\log \frac{p_1}{1 - p_1} + \log \left(\frac{1 - p_1}{1 - p_1 + p_1 \zeta(p_1, \delta)}\right)\right) / \log \frac{(\sigma_2 + 1)p_1}{(\sigma_2 + 1)(1 - p_1) - 1} \\
&= \frac{\log \left(\frac{p_1}{1 - p_1}\right)}{\log \left(\frac{(\sigma_2 + 1)p_1}{(\sigma_2 + 1)(1 - p_1) - 1}\right)} - \frac{\log \left(\frac{1 - p_1 + p_1 \zeta(p_1, \delta)}{1 - p_1}\right)}{\log \left(\frac{(\sigma_2 + 1)p_1}{(\sigma_2 + 1)(1 - p_1) - 1}\right)}
\end{aligned}$$

We choose $p_0$ and $p_1$ differently depending on the magnitude of $\sigma_2$. We consider two cases: $\sigma_2 \geq 3$ (bounded away from 1), and $\sigma_2 \in (1, 3)$ (close to 1).

**Case 1:** $\sigma_1 \geq 3$. Here we choose $p_0 = p_1 = \frac{2}{3}$, and this satisfies $p_0, p_1 \in \left(\frac{1}{2}, \frac{\sigma_2}{\sigma_2 + 1}\right)$. We now bound the remaining constants. For $b_0$:

$$\begin{aligned}
b_0 &= \frac{(2p_0 - 1)(1 - p_0)}{2p_0^2 \epsilon} (\sigma_1 + \sigma_2 \epsilon) + \frac{1 - p_0}{p_0} \\
&= \frac{1}{8\epsilon} (\sigma_1 + \sigma_2 \epsilon) + \frac{1}{2} \\
&\geq \frac{\sigma_1}{8\epsilon}.
\end{aligned}$$

For $b_1$:

$$
\begin{aligned}
b_1 &= \frac{\frac{1-p_1}{p_1}\left(\sigma_1 + \left(\sigma_2 - \frac{p_1}{1-p_1}\right)G\right)((\sigma_2+1)(2p_1-1)+1) - \sigma_1}{(\sigma_1 + (\sigma_2-1)\epsilon)((\sigma_2+1)(2p_1-1)+1) - \sigma_1} \\
&= \frac{\left(\frac{1-p_1}{p_1}((\sigma_2+1)(2p_1-1)+1) - 1\right)\sigma_1 + \left(\frac{1-p_1}{p_1}\sigma_2 - 1\right)((\sigma_2+1)(2p_1-1)+1)G}{(\sigma_1 + (\sigma_2-1)\epsilon)((\sigma_2+1)(2p_1-1)+1) - \sigma_1} \\
&\overset{(i)}{\geq} \frac{\left(\frac{1-p_1}{p_1}\sigma_2 - 1\right)((\sigma_2+1)(2p_1-1)+1)G}{(\sigma_1 + (\sigma_2-1)\epsilon)((\sigma_2+1)(2p_1-1)+1) - \sigma_1} \\
&\overset{(ii)}{\geq} \frac{\left(\frac{1-p_1}{p_1}\sigma_2 - 1\right)((\sigma_2+1)(2p_1-1)+1)G}{(\sigma_1 + (\sigma_2-1)\sigma_1)((\sigma_2+1)(2p_1-1)+1) - \sigma_1} \\
&= \frac{\left(\frac{1-p_1}{p_1}\sigma_2 - 1\right)((\sigma_2+1)(2p_1-1)+1)G}{(\sigma_2((\sigma_2+1)(2p_1-1)+1) - 1)\sigma_1} \\
&\overset{(iii)}{=} \frac{\left(\frac{1}{2}\sigma_2 - 1\right)\left(\frac{1}{3}\sigma_2 + \frac{4}{3}\right)G}{\left(\sigma_2\left(\frac{1}{3}\sigma_2 + \frac{4}{3}\right) - 1\right)\sigma_1} = \frac{(\sigma_2 - 2)(\sigma_2+4)G}{2(\sigma_2(\sigma_2+4) - 3)\sigma_1} \geq \frac{(\sigma_2 - 2)(\sigma_2+4)G}{2\sigma_2(\sigma_2+4)\sigma_1} \\
&= \frac{(\sigma_2 - 2)G}{2\sigma_2\sigma_1} \overset{(iv)}{\geq} \frac{G}{6\sigma_1},
\end{aligned}
$$

where $(i)$ uses the fact that

$$
\frac{1-p_1}{p_1}((\sigma_2+1)(2p_1-1)+1) - 1 = \frac{1}{2}\left(\frac{1}{3}\sigma_2 + \frac{4}{3}\right) - 1 = \frac{1}{6}\sigma_2 - \frac{1}{3} > 0,
$$

$(ii)$ uses $\epsilon \leq \sigma_1$ as assumed in the statement of Theorem 4, $(iii)$ plugs in $p_1 = 2/3$, and $(iv)$ uses $\sigma_2 \geq 3$. For $\phi_0$ :

$$
\phi_0 = 1 - \frac{\log\left(\frac{1-p_0+p_0\zeta(p_0,\delta)}{1-p_0}\right)}{\log\frac{p_0}{1-p_0}} = 1 - \frac{\log(1 + 2\zeta(2/3,\delta))}{\log 2} = 1 - \gamma_1,
$$

where we denoted

$$
\gamma_1 = \frac{\log(1 + 2\zeta(2/3,\delta))}{\log 2}.
$$

For $\phi_1$:

$$
\begin{aligned}
\phi_1 &= \frac{\log\left(\frac{p_1}{1-p_1}\right)}{\log\left(\frac{(\sigma_2+1)p_1}{(\sigma_2+1)(1-p_1)-1}\right)} - \frac{\log\left(\frac{1-p_1+p_1\zeta(p_1,\delta)}{1-p_1}\right)}{\log\left(\frac{(\sigma_2+1)p_1}{(\sigma_2+1)(1-p_1)-1}\right)} \\
&= \frac{\log 2}{\log\left(2 + \frac{6}{\sigma_2-2}\right)} - \frac{\log(1 + 2\zeta(2/3,\delta))}{\log\left(2 + \frac{6}{\sigma_2-2}\right)} \\
&= 1 - \left(1 - \frac{\log 2}{\log\left(2 + \frac{6}{\sigma_2-2}\right)}\right) - \frac{\log(1 + 2\zeta(2/3,\delta))}{\log\left(2 + \frac{6}{\sigma_2-2}\right)} \\
&= 1 - \gamma_2 - \gamma_3,
\end{aligned}
$$

where we denoted

$$
\gamma_2 = 1 - \frac{\log 2}{\log\left(2 + \frac{6}{\sigma_2-2}\right)}
$$

$$
\gamma_3 = \frac{\log(1 + 2\zeta(2/3,\delta))}{\log\left(2 + \frac{6}{\sigma_2-2}\right)}.
$$

Finally, we can plug our bounds of $b_0, b_1, \phi_0, \phi_1$ into Equation 50:

$$
\begin{aligned}
\beta(\boldsymbol{x}_0) &\le 4m \left(\lambda_0(2/3, \delta)\right)^{-1} \left(\lambda_0(2/3, \delta)\right)^{-1} \left(\frac{8\epsilon}{\sigma_1}\right)^{1-\gamma_1} \left(\frac{6\sigma_1}{G}\right)^{1-\gamma_2-\gamma_3} \\
&\overset{(i)}{\le} 192m \left(\lambda_0(2/3, 0)\right)^{-1} \left(\lambda_0(2/3, 0)\right)^{-1} \left(\frac{\epsilon}{\sigma_1}\right)^{1-\gamma_1} \left(\frac{\sigma_1}{G}\right)^{1-\gamma_2-\gamma_3} \\
&\overset{(ii)}{\le} 768m \left(\frac{\epsilon}{\sigma_1}\right)^{1-\gamma_1} \left(\frac{\sigma_1}{G}\right)^{1-\gamma_2-\gamma_3} \\
&= 768m \frac{\epsilon^{1-\gamma_1}}{G^{1-\gamma_2-\gamma_3}\sigma_1^{\gamma_2+\gamma_3-\gamma_1}} \\
&= 3072 \frac{\epsilon^{1-\gamma_1}}{L_1 G^{1-\gamma_2-\gamma_3}\sigma_1^{\gamma_2+\gamma_3-\gamma_1}} \log\left(1 + \frac{\Delta L_1^2}{L_0}\right),
\end{aligned}
$$

where $(i)$ uses the fact that $\lambda_0(p, \delta)$ is decreasing in terms of $\delta$, and $(ii)$ uses $\lambda_0(p, 0) = \frac{1-p}{p}$ (from Lemma 14). As noted after Equation 49, $\phi_0 > \phi_1$, so that $\gamma_1 < \gamma_2 + \gamma_3$. Replacing $\beta(\boldsymbol{x}_0) = \beta(\boldsymbol{g}) = \alpha(\boldsymbol{g})\epsilon$ yields

$$
\begin{aligned}
\alpha(\boldsymbol{g}) &\le \frac{3072}{L_1 G^{1-\gamma_2-\gamma_3}\epsilon^{\gamma_1}\sigma_1^{\gamma_2+\gamma_3-\gamma_1}} \log\left(1 + \frac{\Delta L_1^2}{L_0}\right) \\
&\le \frac{3072}{L_1 (\Delta L_1)^{1-\gamma_2-\gamma_3}\epsilon^{\gamma_1}\sigma_1^{\gamma_2+\gamma_3-\gamma_1}} \log\left(1 + \frac{\Delta L_1^2}{L_0}\right)\left(1 + 4\log\left(1 + \frac{\Delta L_1^2}{L_0}\right)\right),
\end{aligned}
$$

which is the desired result.

**Case 2:** $\sigma_2 \in (1, 3)$. Here we choose $p_0 = p_1 = \frac{\sigma_2+5}{12}$, which satisfies $p_0, p_1 \in \left(\frac{1}{2}, \frac{\sigma_2}{\sigma_2+1}\right)$. With this choice,

$$
1 - p_0 = \frac{-\sigma_2 + 7}{12}, \quad 2p_0 - 1 = \frac{\sigma_2 - 1}{6},
$$

and similarly for $p_1$. We can now bound the remaining constants. For $b_0$:

$$
\begin{aligned}
b_0 &= \frac{(2p_0 - 1)(1 - p_0)}{2p_0^2\epsilon}(\sigma_1 + \sigma_2\epsilon) + \frac{1-p_0}{p_0} \\
&= \frac{(\sigma_2 - 1)(-\sigma_2 + 7)}{(\sigma_2 + 5)^2}\frac{\sigma_1 + \sigma_2\epsilon}{\epsilon} + \frac{-\sigma_2 + 7}{\sigma_2 + 5} \\
&\overset{(i)}{\ge} \frac{6(\sigma_2 - 1)}{64}\frac{\sigma_1 + \sigma_2\epsilon}{\epsilon} + \frac{1}{2} \\
&\ge \frac{3(\sigma_2 - 1)\sigma_1}{32\epsilon}
\end{aligned}
$$

where $(i)$ uses $\sigma_2 \in (1, 3)$. For $b_1$:

$$
\begin{aligned}
b_1 &= \frac{\frac{1-p_1}{p_1}\left(\sigma_1 + \left(\sigma_2 - \frac{p_1}{1-p_1}\right)G\right)((\sigma_2+1)(2p_1-1)+1) - \sigma_1}{(\sigma_1 + (\sigma_2-1)\epsilon)((\sigma_2+1)(2p_1-1)+1) - \sigma_1} \\
&= \frac{\left(\frac{1-p_1}{p_1}((\sigma_2+1)(2p_1-1)+1)-1\right)\sigma_1 + \left(\frac{1-p_1}{p_1}\sigma_2 - 1\right)((\sigma_2+1)(2p_1-1)+1)G}{(\sigma_1 + (\sigma_2-1)\epsilon)((\sigma_2+1)(2p_1-1)+1) - \sigma_1} \\
&\overset{(i)}{\geq} \frac{\left(\frac{1-p_1}{p_1}\sigma_2 - 1\right)((\sigma_2+1)(2p_1-1)+1)G}{(\sigma_1 + (\sigma_2-1)\epsilon)((\sigma_2+1)(2p_1-1)+1) - \sigma_1} \\
&\overset{(ii)}{\geq} \frac{\left(\frac{1-p_1}{p_1}\sigma_2 - 1\right)((\sigma_2+1)(2p_1-1)+1)G}{(\sigma_1 + (\sigma_2-1)\sigma_1)((\sigma_2+1)(2p_1-1)+1) - \sigma_1} \\
&= \frac{\left(\frac{1-p_1}{p_1}\sigma_2 - 1\right)((\sigma_2+1)(2p_1-1)+1)G}{(\sigma_2((\sigma_2+1)(2p_1-1)+1)-1)\sigma_1} \\
&\geq \frac{\left(\frac{1-p_1}{p_1}\sigma_2 - 1\right)G}{\sigma_2 \sigma_1} = \left(\frac{1-p_1}{p_1} - \frac{1}{\sigma_2}\right)\frac{G}{\sigma_1} = \left(\frac{-\sigma_2+7}{\sigma_2+5} - \frac{1}{\sigma_2}\right)\frac{G}{\sigma_1} \\
&= \frac{(\sigma_2-1)(-\sigma_2+5)}{\sigma_2(\sigma_2+5)}\frac{G}{\sigma_1} \overset{(iii)}{\geq} \frac{(\sigma_2-1)G}{12\sigma_1},
\end{aligned}
$$

where $(i)$ uses

$$
\begin{aligned}
\frac{1-p_1}{p_1}((\sigma_2+1)(2p_1-1)+1) - 1 &= \frac{-\sigma_2+7}{\sigma_2+5}\left((\sigma_2+1)\frac{\sigma_2-1}{6}+1\right) - 1 \\
&= \frac{(-\sigma_2+7)(\sigma_2^2+5)}{6(\sigma_2+5)} - 1 \\
&= \frac{-\sigma_2+7}{6} - 1 > 0,
\end{aligned}
$$

$(ii)$ uses $\epsilon \leq \sigma_1$, as was assumed in the statement of Theorem 4, and $(iii)$ uses $\sigma_2 \in (1, 3)$. For $\phi_0$:

$$
\begin{aligned}
\phi_0 &= 1 - \frac{\log\left(\frac{1-p_0+p_0\zeta(p_0,\delta)}{1-p_0}\right)}{\log\frac{p_0}{1-p_0}} = 1 - \frac{\log\left(1 + \frac{\sigma_2+5}{-\sigma_2+7}\zeta(p_0,\delta)\right)}{\log\frac{\sigma_2+5}{-\sigma_2+7}} \\
&= 1 - \frac{\log\left(1 + \frac{\sigma_2+5}{-\sigma_2+7}\zeta(p_0,\delta)\right)}{\log\left(\frac{12}{-\sigma_2+7}-1\right)} = 1 - \frac{\log\left(1 + 2\zeta(p_0,\delta)\right)}{\log\left(\frac{12}{-\sigma_2+7}-1\right)} \\
&= 1 - \gamma_4,
\end{aligned}
$$

where we denoted

$$
\gamma_4 = \frac{\log\left(1 + 2\zeta(\frac{1}{12}(\sigma_2+5),\delta)\right)}{\log\left(\frac{12}{-\sigma_2+7}-1\right)}.
$$

Lastly, for $\phi_1$, notice that

$$
\begin{aligned}
\frac{(\sigma_2+1)p_1}{(\sigma_2+1)(1-p_1)-1} &= \frac{(\sigma_2+1)(\sigma_2+5)}{12}\left(\frac{(\sigma_2+1)(-\sigma_2+7)}{12}-1\right)^{-1} \\
&= \frac{(\sigma_2+1)(\sigma_2+5)}{(\sigma_2+1)(-\sigma_2+7)-12} = \frac{(\sigma_2+1)(\sigma_2+5)}{(\sigma_2-1)(-\sigma_2+5)} \\
&= \frac{3}{\sigma_2-1} + \frac{15}{-\sigma_2+5} - 1 \leq \frac{18}{\sigma_2-1} - 1.
\end{aligned}
$$

Therefore

$$\phi_1 = \frac{\log\left(\frac{p_1}{1-p_1}\right)}{\log\left(\frac{(\sigma_2+1)p_1}{(\sigma_2+1)(1-p_1)-1}\right)} - \frac{\log\left(\frac{1-p_1+p_1\zeta(p_1,\delta)}{1-p_1}\right)}{\log\left(\frac{(\sigma_2+1)p_1}{(\sigma_2+1)(1-p_1)-1}\right)}$$

$$= \frac{\log\left(\frac{\sigma_2+5}{-\sigma_2+7}\right)}{\log\left(\frac{18}{\sigma_2-1}-1\right)} - \frac{\log\left(1+\frac{\sigma_2+5}{-\sigma_2+7}\zeta(p_1,\delta)\right)}{\log\left(\frac{18}{\sigma_2-1}-1\right)}$$

$$\geq \frac{\log\left(\frac{12}{-\sigma_2+7}-1\right)}{\log\left(\frac{18}{\sigma_2-1}-1\right)} - \frac{\log\left(1+2\zeta(p_1,\delta)\right)}{\log\left(\frac{18}{\sigma_2-1}-1\right)}$$

$$= 1 - \gamma_5 - \gamma_6$$

where we denoted

$$\gamma_5 = 1 - \frac{\log\left(\frac{12}{-\sigma_2+7}-1\right)}{\log\left(\frac{18}{\sigma_2-1}-1\right)}$$

$$\gamma_6 = \frac{\log\left(1+2\zeta(\frac{1}{12}(\sigma_2+5),\delta)\right)}{\log\left(\frac{18}{\sigma_2-1}-1\right)}.$$

Finally, we can plug our bounds of $b_0, b_1, \phi_0, \phi_1$ into Equation 50:

$$\beta(\boldsymbol{x}_0) \leq 4m\left(\lambda_0(p_0,\delta)\right)^{-1}\left(\lambda_0(p_1,\delta)\right)^{-1}\left(\frac{32\epsilon}{3(\sigma_2-1)\sigma_1}\right)^{1-\gamma_4}\left(\frac{12\sigma_1}{(\sigma_2-1)G}\right)^{1-\gamma_5-\gamma_6}$$

$$\overset{(i)}{\leq} 512m\left(\lambda_0(2/3,0)\right)^{-1}\left(\lambda_0(2/3,0)\right)^{-1}\left(\frac{\epsilon}{(\sigma_2-1)\sigma_1}\right)^{1-\gamma_4}\left(\frac{\sigma_1}{(\sigma_2-1)G}\right)^{1-\gamma_5-\gamma_6}$$

$$\overset{(ii)}{\leq} 2048m\left(\frac{\epsilon}{(\sigma_2-1)\sigma_1}\right)^{1-\gamma_4}\left(\frac{\sigma_1}{(\sigma_2-1)G}\right)^{1-\gamma_5-\gamma_6}$$

$$\leq 2048m\frac{\epsilon^{1-\gamma_4}}{(\sigma_2-1)^{2-\gamma_4-\gamma_5-\gamma_6}G^{1-\gamma_5-\gamma_6}\sigma_1^{\gamma_5+\gamma_6-\gamma_4}}$$

$$\leq \frac{8192\epsilon^{1-\gamma_4}}{(\sigma_2-1)^{2-\gamma_4-\gamma_5-\gamma_6}L_1 G^{1-\gamma_5-\gamma_6}\sigma_1^{\gamma_5+\gamma_6-\gamma_4}}\log\left(1+\frac{\Delta L_1^2}{L_0}\right)$$

$$= \frac{8192\epsilon^{1-\gamma_4}}{(\sigma_2-1)^{2-\gamma_4-\gamma_5-\gamma_6}L_1(\Delta L_1)^{1-\gamma_5-\gamma_6}\sigma_1^{\gamma_5+\gamma_6-\gamma_4}}\log\left(1+\frac{\Delta L_1^2}{L_0}\right)\left(1+4\log\left(1+\frac{\Delta L_1^2}{L_0}\right)\right),$$

where $(i)$ uses the fact that $\lambda_0(p,\delta)$ is increasing in terms of $p$ and decreasing in terms of $\delta$, and $(ii)$ uses $\lambda_0(p,0) = \frac{1-p}{p}$ (from Lemma 14). As noted after Equation 49, $\phi_0 > \phi_1$, so that $\gamma_4 < \gamma_5 + \gamma_6$. Replacing $\beta(\boldsymbol{x}_0) = \beta(\boldsymbol{g}) = \alpha(\boldsymbol{g})\epsilon$ yields

$$\alpha(\boldsymbol{g}) \leq \frac{8192}{(\sigma_2-1)^{2-\gamma_4-\gamma_5-\gamma_6}\epsilon^{\gamma_4}L_1(\Delta L_1)^{1-\gamma_5-\gamma_6}\sigma_1^{\gamma_5+\gamma_6-\gamma_4}}\log\left(1+\frac{\Delta L_1^2}{L_0}\right)\left(1+4\log\left(1+\frac{\Delta L_1^2}{L_0}\right)\right),$$

which is the desired result. $\qquad\square$

**Lemma 12.** *Define*

$$\alpha_0 = \max_{\|\boldsymbol{g}\|=\epsilon}\alpha(\boldsymbol{g}).$$

*There exists $f \in \mathcal{F}_{\mathrm{aff}}(\Delta, L_0, L_1, 0, 0)$ such that $\|\nabla f(\boldsymbol{x}_t)\| \geq \epsilon$ for all $t$ with*

$$t \leq \frac{\Delta}{2\alpha_0\epsilon^2}.$$

*Proof.* Denote $a = \frac{1}{L_1} \log\left(1 + \frac{L_1\epsilon}{L_0}\right)$, and let $\boldsymbol{g} \in \mathbb{R}^d$ such that $\|\boldsymbol{g}\| = \epsilon$ and $\alpha(\boldsymbol{g}) = \alpha_0$. Define the objective $f : \mathbb{R}^d \to \mathbb{R}$ as follows:

$$f(\boldsymbol{x}) = \begin{cases} -\epsilon\left(\hat{P}_{\boldsymbol{g}}(\boldsymbol{x}) + a\right) + \psi(a) & \hat{P}_{\boldsymbol{g}}(\boldsymbol{x}) \leq -a \\ \psi(\hat{P}_{\boldsymbol{g}}(\boldsymbol{x})) & \hat{P}_{\boldsymbol{g}}(\boldsymbol{x}) \in (-a, a) \\ \epsilon\left(\hat{P}_{\boldsymbol{g}}(\boldsymbol{x}) - a\right) + \psi(a) & \hat{P}_{\boldsymbol{g}}(\boldsymbol{x}) \geq a \end{cases},$$

where $\psi$ is as defined in Lemma 1. It is straightforward to show that $f$ is continuously differentiable, bounded from below ($f_* = 0$), and $(L_0, L_1)$-smooth. Also, with the initial point $\boldsymbol{x}_0 = \left(a + \frac{\Delta - \psi(a)}{\epsilon}\right) \frac{\boldsymbol{g}}{\|\boldsymbol{g}\|}$, $f$ satisfies $f(x_0) - f^* = \Delta$. So $f \in \mathcal{F}_{\text{aff}}(\Delta, L_0, L_1, 0, 0)$.

Consider the execution of $A$ on $f$ from $\boldsymbol{x}_0 = \left(a + \frac{\Delta - \psi(a)}{\epsilon}\right) \frac{\boldsymbol{g}}{\|\boldsymbol{g}\|}$, and let $t_0 = \max\{t \geq 0 \mid P_{\boldsymbol{g}}(\boldsymbol{x}_s) \geq a \text{ for all } 0 \leq s \leq t\}$. Then $\nabla f(x_t) = \boldsymbol{g}$ for any $t \leq t_0$, so that

$$\boldsymbol{x}_{t+1} = \boldsymbol{x}_t - \alpha(\nabla f(\boldsymbol{x}_t))\nabla f(\boldsymbol{x}_t) = \boldsymbol{x}_t - \alpha(\boldsymbol{g})\boldsymbol{g} = \boldsymbol{x}_t - \alpha_0 \boldsymbol{g},$$

so

$$\hat{P}_{\boldsymbol{g}}(\boldsymbol{x}_{t+1}) = \left\langle \boldsymbol{x}_{t+1}, \frac{\boldsymbol{g}}{\|\boldsymbol{g}\|} \right\rangle = \left\langle \boldsymbol{x}_t - \alpha_0 \boldsymbol{g}, \frac{\boldsymbol{g}}{\|\boldsymbol{g}\|} \right\rangle = \hat{P}_{\boldsymbol{g}}(\boldsymbol{x}_t) - \alpha_0\|\boldsymbol{g}\| = \hat{P}_{\boldsymbol{g}}(\boldsymbol{x}_t) - \alpha_0\epsilon.$$

Unrolling over $t$ yields

$$\hat{P}_{\boldsymbol{g}}(\boldsymbol{x}_{t+1}) = \hat{P}_{\boldsymbol{g}}(\boldsymbol{x}_0) - (t+1)\alpha_0\epsilon.$$

In particular, choosing $t = t_0$ yields $\hat{P}_{\boldsymbol{g}}(\boldsymbol{x}_{t_0+1}) \geq \hat{P}_{\boldsymbol{g}}(\boldsymbol{x}_0) - (t_0+1)\alpha(\epsilon)\epsilon$. By the definition of $t_0$, we also know $\hat{P}_{\boldsymbol{g}}(\boldsymbol{x}_{t_0+1}) < a$. Therefore $\hat{P}_{\boldsymbol{g}}(\boldsymbol{x}_0) - (t_0+1)\alpha(\epsilon)\epsilon < a$, and rearranging yields

$$t_0 + 1 > \frac{\hat{P}_{\boldsymbol{g}}(\boldsymbol{x}_0) - a}{\alpha(\epsilon)\epsilon} = \frac{\Delta - \psi(a)}{\alpha(\epsilon)\epsilon^2}.$$

Also,

$$\psi(a) = \frac{\epsilon}{L_1} - \log\left(1 + \frac{L_1\epsilon}{L_0}\right) \leq \frac{\epsilon}{L_1} \leq \frac{\Delta L_1}{2L_1} \leq \frac{\Delta}{2},$$

where the last inequality uses the condition $\epsilon \leq \frac{\Delta L_1}{2}$ from Theorem 4. So

$$t_0 + 1 > \frac{\Delta}{2\alpha(\epsilon)\epsilon^2}.$$

Therefore, $t \leq \frac{\Delta}{2\alpha(\epsilon)\epsilon^2}$ implies that $t < t_0 + 1$, so that $\hat{P}_{\boldsymbol{g}}(\boldsymbol{x}_t) \geq a$ by the definition of $t_0$, and finally $\|\nabla f(\boldsymbol{x}_t)\| = \epsilon$. □

The following lemma is nearly identical to parts of the proof of Theorem 2 in Drori & Shamir (2020), with some small modifications to fit our requirements. We include it here for the sake of completeness.

**Lemma 13.** *For any sufficiently large $d \in \mathbb{N}$ and any $\alpha : \mathbb{R}^d \to \mathbb{R}^d$, there exists some $(f, g, \Xi) \in \mathcal{F}_{\text{aff}}(\Delta, L_0, L_1, \sigma_1, \sigma_2)$ such that $\|\nabla f(\boldsymbol{x}_t)\| = \epsilon$ for all $0 \leq t \leq T$, where*

$$T = \frac{\Delta L_0 \sigma_1^2}{2\epsilon^4}.$$

*Proof.* Suppose $d \geq T$. Let $\alpha : \mathbb{R}^d \to \mathbb{R}^d$ and define $f : \mathbb{R}^d \to \mathbb{R}$ as

$$f(\boldsymbol{x}) = \epsilon\langle \boldsymbol{x}, \mathbf{e}_1 \rangle + \sum_{i=2}^T h_i(\langle \boldsymbol{x}, \mathbf{e}_i \rangle),$$

where

$$h_i(x) = \begin{cases} \frac{L_0}{4} a_i^2 & |x| < -a_i \\ -\frac{L_0}{2}(x + a_i)^2 + \frac{L_0}{4} a_i^2 & |x| \in \left[-a_i, -\frac{a_i}{2}\right] \\ \frac{L_0}{2} x^2 & |x| \in \left(-\frac{a_i}{2}, \frac{b_i}{2}\right) \\ -\frac{L_0}{2}(x - b_i)^2 + \frac{L_0}{4} b_i^2 & |x| \in \left[\frac{b_i}{2}, b_i\right] \\ \frac{L_0}{4} b_i^2 & |x| > b_i \end{cases}$$

$$a_i = \sigma_1 \alpha(\epsilon \mathbf{e}_1 + \sigma_1 \mathbf{e}_i)$$
$$b_i = \sigma_1 \alpha(\epsilon \mathbf{e}_1 - \sigma_1 \mathbf{e}_i).$$

For any $\boldsymbol{x}, \boldsymbol{y} \in \mathbb{R}^d$,

$$\|\nabla f(\boldsymbol{x}) - \nabla f(\boldsymbol{y})\|^2 = \sum_{i=1}^d (\nabla_i f(\boldsymbol{x}) - \nabla_i f(\boldsymbol{y}))^2$$

$$= \sum_{i=2}^d (h'(x_i) - h'(y_i))^2$$

$$\overset{(i)}{\leq} L_0^2 \sum_{i=2}^d (x_i - y_i)^2$$

$$\leq L_0^2 \|\boldsymbol{x} - \boldsymbol{y}\|^2,$$

where $(i)$ uses the fact that $h$ is $L_0$-smooth. Therefore $f$ is $L_0$-smooth, and consequently is $(L_0, L_1)$-smooth. Also, define the following stochastic gradient for $f$:

$$F(\boldsymbol{x}, \xi) = \nabla f(\boldsymbol{x}) + (2\xi - 1)\sigma_1 \mathbf{e}_{j(\boldsymbol{x})},$$

where

$$j(\boldsymbol{x}) = \min\{1 \leq i \leq d \mid \langle \boldsymbol{x}, \mathbf{e}_i \rangle = 0\}.$$

This oracle is defined so that the stochastic gradient noise at step $t$ only affects coordinate $t + 1$ (this will be shown later). Let $\mathcal{D}$ be the distribution of $\xi$, defined as $P(\xi = 0) = P(\xi = 1) = \frac{1}{2}$. With this definition, the stochastic gradient $F$ satisfies

$$\mathbb{E}[F(\boldsymbol{x}, \xi)] = \nabla f(\boldsymbol{x})$$
$$\|F(\boldsymbol{x}, \xi) - \nabla f(\boldsymbol{x})\| \leq \sigma_1 \quad \text{(almost surely)}.$$

Therefore, all of the conditions for $(f, F, \mathcal{D}) \in \mathcal{F}_{\text{aff}}(\Delta, L_0, L_1, \sigma_1, \sigma_2)$ are satisfied other than the condition that $f$ is bounded from below and $f(\boldsymbol{x}_0) - \inf_{\boldsymbol{x}} f(\boldsymbol{x}) \leq \Delta$. This condition will be addressed at the end of this lemma's proof.

Now consider the trajectory of $A$ on $f$ from the initial point $\boldsymbol{x}_0 = \mathbf{0}$. We claim that for all $0 \leq t \leq T$:

$$\langle \boldsymbol{x}_t, \mathbf{e}_1 \rangle = -\epsilon \sum_{i=0}^{t-1} \alpha(F(\boldsymbol{x}_i, \xi_i)) \tag{51}$$

$$\langle \boldsymbol{x}_t, \mathbf{e}_j \rangle = \begin{cases} -a_j & \xi_j = 1 \\ b_j & \xi_j = 0 \end{cases} \quad \text{for all } 2 \leq j \leq t + 1 \tag{52}$$

$$\langle \boldsymbol{x}_t, \mathbf{e}_j \rangle = 0 \quad \text{for all } j > t + 1, \tag{53}$$

which we will prove by induction on $t$. By construction, all three of the above hold for the base case $t = 0$. Now, suppose that they hold for some $0 \leq t \leq T - 1$. Then for $j \geq 2$,

$$\nabla_j f(\boldsymbol{x}_t) = h'_j(\langle \boldsymbol{x}, \mathbf{e}_j \rangle) \overset{(i)}{=} \begin{cases} h'_j(-a_j) & j \leq t + 1 \text{ and } \xi_j = 1 \\ h'_j(b_j) & j \leq t + 1 \text{ and } \xi_j = 0 \\ h'_j(0) & j > t + 1 \end{cases} \overset{(ii)}{=} 0,$$

where $(i)$ uses Equation 52 and Equation 53 from the induction hypothesis, and $(ii)$ comes from the definition of $h$. Therefore $\nabla f(\boldsymbol{x}_t) = \epsilon \mathbf{e}_1$. Also, Equation 52 and Equation 53 imply that $j(\boldsymbol{x}_t) = t + 2$, so

$$F(\boldsymbol{x}_t, \xi_t) = \nabla f(\boldsymbol{x}_t) + (2\xi_t - 1)\sigma_1 \mathbf{e}_{t+2}.$$

Therefore, the next iterate $\boldsymbol{x}_{t+1}$ is:

$$
\begin{aligned}
\boldsymbol{x}_{t+1} &= \boldsymbol{x}_t - \alpha(F(\boldsymbol{x}_t, \xi_t))F(\boldsymbol{x}_t, \xi_t) \\
&= \boldsymbol{x}_t - \alpha(F(\boldsymbol{x}_t, \xi_t))\left(\epsilon \mathbf{e}_1 + (2\xi_t - 1)\sigma_1 \mathbf{e}_{t+2}\right) \\
&\overset{(i)}{=} -\epsilon\left(\sum_{i=0}^{t-1} \alpha(F(\boldsymbol{x}_i, \xi_i))\right)\mathbf{e}_1 + \sum_{i=2}^{t+1} \langle \boldsymbol{x}_t, \mathbf{e}_i\rangle\mathbf{e}_i - \alpha(F(\boldsymbol{x}_t, \xi_t))\left(\epsilon \mathbf{e}_1 + (2\xi_t - 1)\sigma_1 \mathbf{e}_{t+2}\right) \\
&= -\epsilon\left(\sum_{i=0}^{t} \alpha(F(\boldsymbol{x}_i, \xi_i))\right)\mathbf{e}_1 + \sum_{i=2}^{t+1} \langle \boldsymbol{x}_t, \mathbf{e}_i\rangle\mathbf{e}_i - (2\xi_t - 1)\alpha(F(\boldsymbol{x}_t, \xi_t))\sigma_1 \mathbf{e}_{t+2},
\end{aligned}
$$

where $(i)$ uses Equation 51 from the inductive hypothesis. Notice that the last term (i.e. the coefficient of $\mathbf{e}_{t+2}$) equals $-a_i$ when $\xi_t = 1$ and it equals $b_i$ when $\xi_t = 0$. This proves Equation 51, Equation 52, and Equation 53 for step $t + 1$. This completes the induction. Together, these three equations imply that $\|\nabla f(\boldsymbol{x}_t)\| = \epsilon$ for all $t \leq T$, which is the desired conclusion.

The only remaining detail is the satisfaction of the condition $f(\boldsymbol{x}_0) - \inf_{\boldsymbol{x}} f(\boldsymbol{x}) \leq \Delta$. As currently stated, the objective $f$ does not satisfy this condition because it is not even bounded from below due to the linear term $\epsilon\langle \boldsymbol{x}, \mathbf{e}_1\rangle$. Similarly to Drori & Shamir (2020), we instead argue that there exists a lower bounded function $\hat{f}$ that has the same first-order information as $f$ at all of the points $\boldsymbol{x}_t$ for $0 \leq t \leq T$. If this happens, then the behavior of $A$ when optimizing $\hat{f}$ is the same as that of $A$ when optimizing $f$, so the conclusion $\|\nabla \hat{f}(\boldsymbol{x}_t)\| = \epsilon$ still holds. Specifically, we need $\hat{f}$ which is lower bounded and that satisfies:

$$
\nabla \hat{f}(\boldsymbol{x}_t) = \nabla f(\boldsymbol{x}_t), \quad \hat{f}(\boldsymbol{x}_t) = f(\boldsymbol{x}_t)
$$

for all $0 \leq t \leq T$. The existence of such an $\hat{f}$ follows immediately from Lemma 1 of Drori & Shamir (2020), and this $\hat{f}$ satisfies

$$
\inf_{\boldsymbol{x}} \hat{f}(\boldsymbol{x}) \geq \min_{0 \leq t \leq T} f(\boldsymbol{x}_t) - \frac{3\epsilon^2}{2L_0}.
$$

Therefore

$$
\begin{aligned}
\hat{f}(\boldsymbol{x}_0) - \inf_{\boldsymbol{x}} \hat{f}(\boldsymbol{x}) &\leq \frac{3\epsilon^2}{2L_0} - \min_{0 \leq t \leq T} f(\boldsymbol{x}_t) \\
&\leq \frac{3\epsilon^2}{2L_0} + \max_{0 \leq t \leq T}\left(-\epsilon\langle \boldsymbol{x}_t, \mathbf{e}_1\rangle - \sum_{i=2}^{T} h_i(\langle \boldsymbol{x}_t, \mathbf{e}_i\rangle)\right) \\
&= \frac{3\epsilon^2}{2L_0} + \max_{0 \leq t \leq T}\left(\epsilon^2 \sum_{i=0}^{t-1} \alpha(F(\boldsymbol{x}_i, \xi_i)) - \sum_{i=2}^{t+1} h_i(\langle \boldsymbol{x}_t, \mathbf{e}_i\rangle)\right). \quad (54)
\end{aligned}
$$

Denote $\alpha_t = \alpha(F(\boldsymbol{x}_t, \xi_t))$. Then for $2 \leq i \leq t + 1$,

$$
h_i(\langle \boldsymbol{x}_t, \mathbf{e}_i\rangle) = \frac{1}{4}L_0 \sigma_1^2 \alpha_{i-2}^2.
$$

Plugging into Equation 54:

$$
\begin{aligned}
\hat{f}(\boldsymbol{x}_0) - \inf_{\boldsymbol{x}} \hat{f}(\boldsymbol{x}) &\leq \frac{3\epsilon^2}{2L_0} + \max_{0 \leq t \leq T}\left(\epsilon^2 \sum_{i=0}^{t-1} \alpha_i - \frac{1}{4}L_0 \sigma_1^2 \sum_{i=2}^{t+1} \alpha_{i-2}^2\right) \\
&\leq \frac{3\epsilon^2}{2L_0} + \max_{0 \leq t \leq T} \sum_{i=0}^{t-1}\left(\underbrace{\epsilon^2 \alpha_i - \frac{1}{4}L_0 \sigma_1^2 \alpha_i^2}_{Q_i}\right)
\end{aligned}
$$

$Q_i$ can be upper bounded by the maximum value of $\epsilon^2 x - \frac{1}{4}L_0\sigma_1^2 x^2$ as a function of $x$, which is $\frac{\epsilon^4}{L_0\sigma_1^2}$. Therefore

$$
\begin{aligned}
\hat{f}(\boldsymbol{x}_0) - \inf_{\boldsymbol{x}} &\le \frac{3\epsilon^2}{2L_0} + \max_{0 \le t \le T} \sum_{i=0}^{t-1} \frac{\epsilon^4}{L_0\sigma_1^2} \\
&= \frac{3\epsilon^2}{2L_0} + \frac{T\epsilon^4}{L_0\sigma_1^2} \\
&\overset{(i)}{\le} \frac{3\epsilon^2}{2L_0} + \frac{\Delta}{2} \\
&\overset{(ii)}{\le} \Delta,
\end{aligned}
$$

where $(i)$ uses $T \le \frac{\Delta L_0 \sigma_1^2}{2\epsilon^4}$ and $(ii)$ uses $\epsilon \le \sqrt{\Delta L_0/3}$. $\qquad\square$

**Theorem 7.** *[Restatement of Theorem 4] Let $\Delta, L_0, L_1, \sigma_1 > 0$ and $\sigma_2 > 1$. Denote*

$$
G = \frac{\Delta L_1}{1 + 4\log\left(1 + \frac{\Delta L_1^2}{L_0}\right)},
$$

*and suppose $G \ge \sigma_1$. Let $0 < \epsilon \le \min\left\{\sigma_1, \frac{G}{2}, \frac{G-\sigma_1}{\sigma_2-1}\right\}$. Let algorithm $A_{ada}$ denote single-step adaptive SGD with any step size function $\alpha : \mathbb{R}^d \to \mathbb{R}$ for sufficiently large $d$, and let $\mathcal{F} = \mathcal{F}_{\text{aff}}(\Delta, L_0, L_1, \sigma_1, \sigma_2)$. If $\sigma_2 \ge 3$, then*

$$
\mathcal{T}(A_{ada}, \mathcal{F}, \epsilon, \delta) \ge \tilde{\Omega}\left(\frac{(\Delta L_1)^{2-\gamma_2-\gamma_3}\sigma_1^{\gamma_2+\gamma_3-\gamma_1}}{\epsilon^{2-\gamma_1}}\right).
$$

*Otherwise, if $\sigma_2 \in (1, 3)$, then*

$$
\mathcal{T}(A_{ada}, \mathcal{F}, \epsilon, \delta) \ge \tilde{\Omega}\left(\frac{(\Delta L_1)^{2-\gamma_5-\gamma_6}\sigma_1^{\gamma_5+\gamma_6-\gamma_4}}{\epsilon^{2-\gamma_4}}(\sigma_2-1)^{2+\gamma_4-\gamma_5-\gamma_6}\right).
$$

*Proof of Theorem 4.* We only need to combine Lemmas 9, 10, 11, 12 and 13. If there exists any $\boldsymbol{g} \in \mathbb{R}^d$ such that $\|\boldsymbol{g}\| \in [\epsilon, \sigma_1 + (\sigma_2+1)M]$ and

$$
\alpha(\boldsymbol{g}) \le 0 \quad \text{or} \quad \alpha(\boldsymbol{g}) \ge \frac{4}{L_1\|\boldsymbol{g}\|}\log\left(1 + \frac{L_1\min(\|\boldsymbol{g}\|, M)}{L_0}\right),
$$

then there exists some problem instance $(f, F, \mathcal{D})$ such that $\|\nabla f(\boldsymbol{x}_t)\| \ge \epsilon$ for all $t \ge 0$ (Lemma 9). If no such $\boldsymbol{g}$ exists, and there exist any tricky pairs with respect to the stepsize function $\alpha$, then there exists some problem instance $(f, F, \mathcal{D})$ such that $\|\nabla f(\boldsymbol{x}_t)\| \ge \epsilon$ for all $t \ge 0$ with probability at least $\delta$ (Lemma 10). Suppose neither of these cases hold. Lemma 12 implies that there exists a problem instance $(f, F, \mathcal{D})$ such that $\|\nabla f(\boldsymbol{x}_t)\| \ge \epsilon$ for all $t$ with

$$
T \le \frac{\Delta}{4\alpha_0\epsilon^2}.
$$

Since neither of the above cases hold, the conditions of Lemma 11 hold, so we can bound $\alpha_0$ with two cases. If $\sigma_2 \ge 3$, then

$$
\alpha_0 \le \frac{3072}{L_1(\Delta L_1)^{1-\gamma_2-\gamma_3}\epsilon^{\gamma_1}\sigma_1^{\gamma_2+\gamma_3-\gamma_1}}\log\left(1 + \frac{\Delta L_1^2}{L_0}\right)\left(1 + 4\log\left(1 + \frac{\Delta L_1^2}{L_0}\right)\right),
$$

so $\|\nabla f(\boldsymbol{x}_t)\| \ge \epsilon$ for all $t$ with

$$
t \le \tilde{O}\left(\frac{\Delta L_0 \sigma_1^2}{\epsilon^4} + \frac{(\Delta L_1)^{2-\gamma_2-\gamma_3}\sigma_1^{\gamma_2+\gamma_3-\gamma_1}}{\epsilon^{2-\gamma_1}}\right). \tag{55}
$$

If $\sigma_2 \in (1, 3)$, then

$$\alpha_0 \leq \frac{8192}{(\sigma_2 - 1)^{2-\gamma_4-\gamma_5-\gamma_6} \epsilon^{\gamma_4} L_1 (\Delta L_1)^{1-\gamma_5-\gamma_6} \sigma_1^{\gamma_5+\gamma_6-\gamma_4}} \log\left(1 + \frac{\Delta L_1^2}{L_0}\right)\left(1 + 4\log\left(1 + \frac{\Delta L_1^2}{L_0}\right)\right),$$

so $\|\nabla f(\boldsymbol{x}_t)\| \geq \epsilon$ for all $t$ with

$$t \leq \tilde{O}\left(\frac{\Delta L_0 \sigma_1^2}{\epsilon^4} + \frac{(\Delta L_1)^{2-\gamma_5-\gamma_6}}{\epsilon^{2-\gamma_4}}(\sigma_2 - 1)^2 \left(\frac{\sigma_1}{\sigma_2 - 1}\right)^{\gamma_5+\gamma_6-\gamma_4}\right). \tag{56}$$

Lemma 13 implies that

$$\mathcal{T}(A_{\text{ada}}, \mathcal{F}, \epsilon, \delta) \geq \frac{\Delta L_0 \sigma_1^2}{2\epsilon^4},$$

and this can be combined with Equation 55 and Equation 56 to obtain the two conclusions of Theorem 4. □

## D  AUXILIARY LEMMAS

Lemmas 14 and 15 deal with the asymmetric random walk described in the proof of Theorem 4. We restate the associated definitions below.

For $p \in (0, 1)$ and $\lambda > 0$, consider the random walk parameterized by $(p, \lambda)$:

$$\begin{aligned} X_0 &= 1 \\ P(X_{t+1} = X_t + \lambda) &= p \\ P(X_{t+1} = X_t - 1) &= 1 - p. \end{aligned} \tag{57}$$

Define

$$\begin{aligned} z_{p,\lambda} &= P(\exists t > 0 : X_t \leq 0) \\ \lambda_0(p, \delta) &= \inf\{\lambda \geq 0 : z_{p,\lambda} \leq 1 - \delta\} \\ \zeta(p, \delta) &= \lambda_0(p, \delta) - \lambda_0(p, 0). \end{aligned}$$

Informally, $z_{p,\lambda}$ is the probability that the random walk reaches a non-positive value, and $\lambda_0(p, \delta)$ is the smallest $\lambda$ required to ensure that the chance of never reaching a non-positive value is at least $\delta$.

**Lemma 14.** *Let $X_t$ be as defined in Equation 57. Then $\lambda_0(p, 0) = \frac{1-p}{p}$.*

*Proof.* Denote $a \wedge b = \min(a, b)$. Define $\tau = \inf_t\{t > 0 : X_t < 0\}$. Note that $X_t = X_0 + \sum_{i=1}^t \xi_i$, where $\{\xi_i\}_{i=1}^t$ are i.i.d. and follow the same distribution: $\Pr(\xi_i = \lambda) = p$ and $\Pr(\xi_i = -1) = 1 - p$.

Now we first prove that $\{X_t - t((\lambda + 1)p - 1)\}_{t=1}^\infty$ is a martingale with respect to itself. To see this, note that for any $t > 0$, we have

$$\mathbb{E}\left[X_t - t((\lambda+1)p - 1) \mid X_{t-1}\right] = \mathbb{E}\left[X_{t-1} + \xi_t - t((\lambda+1)p - 1) \mid X_{t-1}\right] = X_{t-1} - (t-1)((\lambda+1)p - 1),$$

where the last inequality holds because $\mathbb{E}\left[\xi_t \mid X_{t-1}\right] = \lambda p - (1 - p) = (\lambda + 1)p - 1$.

Let $T > 0$ be a fixed constant. Note that $\tau \wedge T$ is a stopping time which is almost surely bounded. Then by the optional sampling theorem,

$$\mathbb{E}\left[X_{\tau \wedge T} - (\tau \wedge T)((\lambda + 1)p - 1)\right] = \mathbb{E}[X_0] = 1.$$

Therefore, we have

$$\mathbb{E}\left[\tau \wedge T\right] = \frac{\mathbb{E}\left[X_{\tau \wedge T}\right] - 1}{(\lambda + 1)p - 1}.$$

Let $T \to \infty$. By the monotone convergence theorem,

$$\mathbb{E}[\tau] = \frac{\lim_{T \to \infty} \mathbb{E}\left[X_{\tau \wedge T}\right] - 1}{(\lambda + 1)p - 1}$$

We consider the following cases.

- If $\lambda < \frac{1-p}{p}$, then $(\lambda + 1)p - 1 < 0$. Combined with $\mathbb{E}[\tau] > 0$, this implies

$$\lim_{T \to \infty} \mathbb{E}[X_{\tau \wedge T}] - 1 < 0.$$

  Now we show that $\mathbb{E}[\tau] < \infty$ by contradiction. If $\mathbb{E}[\tau] = \infty$, then $\lim_{T \to \infty} \mathbb{E}[X_{\tau \wedge T}] = -\infty$, which is impossible because $X_{\tau \wedge T} \geq -1$ for any $T$. Therefore $\Pr(\tau = \infty) = 0$ and $z_{p,\lambda} = 1$.

- If $\lambda \geq \frac{1-p}{p}$, then $(\lambda + 1)p - 1 \geq 0$. Combined with $\mathbb{E}[\tau] > 0$, this implies

$$\lim_{T \to \infty} \mathbb{E}[X_{\tau \wedge T}] - 1 \geq 0.$$

  Now we show that $\Pr(\tau = \infty) > 0$ by contradiction. If $\Pr(\tau = \infty) = 0$, then by the bounded convergence theorem, we have $\lim_{T \to \infty} \mathbb{E}[X_{\tau \wedge T}] = \mathbb{E}[X_\tau] < 0$, which contradicts $\lim_{T \to \infty} \mathbb{E}[X_{\tau \wedge T}] - 1 \geq 0$. Therefore $\Pr(\tau = \infty) > 0$ and $z_{p,\lambda} < 1$.

Therefore $\lambda_0(p) = \frac{1-p}{p}$. $\qquad \square$

**Lemma 15.** *Let $X_t$ be as defined in Equation 57. Then $\lim_{\delta \to 0+} \lambda_0(p, \delta) = \lambda_0(p, 0)$ for all $p \in (0, 1)$.*

*Proof.* The idea of the proof is, given some $\lambda$, to find some $\alpha \in (0, 1)$ such that $Y_t = \alpha^{X_t}$ is a martingale. We can then apply the optional sampling theorem to $\alpha^{X_t}$ in order to get a bound of $z_{p,\lambda}$ in terms of $\lambda$, which we can use to upper bound $\lambda_0(p, \delta)$. This upper bound goes to $\lambda_0(p, 0)$ as $\delta \to 0^+$. Combining with the fact that $\lambda_0(p, \delta)$ is increasing in terms of $\delta$ yields $\lim_{\delta \to 0+} \lambda_0(p, \delta) = \lambda_0(p, 0)$.

Let $p \in (0, 1)$ and $\delta \in (0, p)$. We want to find $\tilde{\lambda}$ such that $z_{p,\tilde{\lambda}} \leq 1 - \delta$ (so that $\lambda_0(p, \delta) \leq \tilde{\lambda}$) and $\tilde{\lambda} \to \lambda_0(p, 0)$ as $\delta \to 0$. First, we need $\alpha \in (0, 1)$ such that $\alpha^{X_t}$ is a martingale. This requires

$$\mathbb{E}[\alpha^{X_{t+1}} \mid X_t] = \alpha^{X_t}$$
$$p\alpha^{X_t + \lambda} + (1-p)\alpha^{X_t - 1} = \alpha^{X_t}$$
$$p\alpha^{X_t + \lambda} - \alpha^{X_t} + (1-p)\alpha^{X_t - 1} = 0$$
$$p\alpha^{\lambda + 1} - \alpha + (1-p) = 0, \tag{58}$$

so we are looking for a root of $h_\lambda(x) = px^{\lambda + 1} - x + (1 - p)$ in the interval $x \in (0, 1)$. We claim that for all $\lambda > \frac{1-p}{p}$, there is exactly one root of $h_\lambda$ in $(0, 1)$. To see that such a root exists, notice that $h_\lambda(0) = 1 - p > 0$ and $h_\lambda(1) = 0$. Also, $h'_\lambda(1) = p(\lambda + 1) - 1 > 0$ (since $\lambda > \frac{1-p}{p}$). Therefore $h_\lambda(1 - z) < 0$ for sufficiently small $z > 0$. Then we can apply the intermediate value theorem to $h_\lambda$ at $h_\lambda(0) > 0$ and $h_\lambda(1 - z) < 0$ to conclude that $h$ must have a root in $(0, 1)$.

To see that this root is unique, note that $h_\lambda$ is strictly convex in $(0, \infty)$, since $h''_\lambda(x) = p\lambda(\lambda + 1)x^{\lambda - 1} > 0$ for $x > 0$. Suppose $h_\lambda$ had two roots $x_1, x_2 \in (0, 1)$, with $x_1 < x_2$. Letting $\alpha = (x_2 - x_1)/(1 - x_1)$, we have by strict convexity $h_\lambda(x_2) < (1 - \alpha)h_\lambda(x_1) + \alpha h_\lambda(1) = 0$, which contradicts $h_\lambda(x_2) = 0$. Therefore, $h_\lambda$ has a unique root in $(0, 1)$ for every $\lambda > \frac{p}{1-p}$. Denote this root as $r(\lambda)$.

Now define $\tilde{\lambda} = \inf \left\{ \lambda > \frac{1-p}{p} \mid r(\lambda) \leq 1 - \delta \right\}$ (the threshold $1 - \delta$ will be used later to show $z_{p,\tilde{\lambda}} \leq 1 - \delta$). In order to show that $\tilde{\lambda}$ exists and that $\tilde{\lambda} \to \lambda_0(p, 0)$ as $\delta \to 0$, we need a few facts about $r(\lambda)$. Specifically, we need

$$r(\lambda) \text{ is decreasing} \tag{59}$$
$$\lim_{\lambda \to \frac{1-p}{p}+} r(\lambda) = 1 \tag{60}$$
$$\lim_{\lambda \to \infty} r(\lambda) = 1 - p. \tag{61}$$

To see Equation 59, let $\lambda_2 > \lambda_1 > \frac{1-p}{p}$. For any $x \in [r(\lambda_1), 1)$:

$$h_{\lambda_2(x)} < h_{\lambda_1(x)} \leq \left( 1 - \frac{x - r(\lambda_1)}{1 - r(\lambda_1)} \right) h_{\lambda_1}(r(\lambda_1)) + \frac{x - r(\lambda_1)}{1 - r(\lambda_1)} h_{\lambda_1}(1) = 0,$$

where the first inequality uses the fact that $h_\lambda(x)$ is decreasing in terms of $\lambda$ for any fixed $x$, and the second inequality uses convexity. Then $r(\lambda_2)$ cannot lie in the interval $[r(\lambda_1), 1)$, so $r(\lambda_2) < r(\lambda_1)$. This shows that $r(\lambda)$ is decreasing.

To prove Equation 60, notice that $r(\lambda) \in (0, 1)$ already implies $\lim_{\lambda \to \frac{1-p}{p}+} \le 1$. So it suffices to show for any $\epsilon \in (0, 1)$ that $r(\lambda) > 1 - \epsilon$ for sufficiently small $\lambda$. Denoting $\ell = \frac{1-p}{p}$,

$$\lim_{\lambda \to \frac{1-p}{p}+} h_\lambda(1 - \epsilon) = p(1-\epsilon)^{1/p} - x + (1 - p) = h_\ell(1-\epsilon) > h_\ell(1) - \epsilon h'_\ell(1) = 0, \qquad (62)$$

where the inequality uses strict convexity of $h_\ell$ and the last equality uses $h_\ell(1) = h'_\ell(1) = 0$. Also

$$\lim_{\lambda \to \frac{1-p}{p}+} h'_\lambda(1 - \epsilon) = \lim_{\lambda \to \frac{1-p}{p}+} p(\lambda + 1)x^\lambda - 1 = (1 - \epsilon)^{\frac{1-p}{p}} - 1 < 0. \qquad (63)$$

Together, Equation 62 and Equation 63 tell us that for sufficiently small $\lambda$: $h_\lambda(1 - \epsilon) > 0$ and $h'_\lambda(1 - \epsilon) < 0$. Then for any $x \le 1 - \epsilon$,

$$h_\lambda(x) \ge h_\lambda(1 - \epsilon) + (x - (1 - \epsilon))h'_\lambda(1 - \epsilon) > 0.$$

In other words, for sufficiently large $\lambda$, the root of $h_\lambda$ cannot be smaller than $1 - \epsilon$, or $r(\lambda) > 1 - \epsilon$. This proves Equation 60.

For Equation 61, let $x \in (0, 1)$ and $\lambda > \frac{1-p}{p}$. Then by strict convexity of $h_\lambda$:

$$h_\lambda(x) > h_\lambda(0) + xh'_\lambda(0) = (1 - p) - x,$$

so $h_\lambda(x) > 0$ for any $x \le 1 - p$. Therefore $r(\lambda) > 1 - p$ for any $\lambda$, so that $\lim_{\lambda \to \infty} r(\lambda) \ge 1 - p$. We can also show that $\lim_{\lambda \to \infty} r(\lambda) \le 1 - p$ by showing for any $\epsilon > 0$ that $r(\lambda) \le 1 - p + \epsilon$ for sufficiently large $\lambda$. By convexity of $h_\lambda$:

$$\lim_{\lambda \to \infty} h_\lambda(1 - p + \epsilon) = \lim_{\lambda \to \infty} p(1 - p + \epsilon)^{\lambda+1} - (1 - p + \epsilon) + (1 - p) = -\epsilon.$$

So $h_\lambda(1 - p + \epsilon) < -\epsilon/2$ sufficiently large $\lambda$. Then for any $x \ge 1 - p + \epsilon$,

$$h_\lambda(x) \le (1 - \alpha)h_\lambda(1 - p + \epsilon) + \alpha h_\lambda(1) = -(1 - \alpha)\epsilon < 0.$$

So the root of $h_\lambda$ must be smaller than $1 - p + \epsilon$, or $r(\lambda) \le 1 - p + \epsilon$. This proves that $\lim_{\lambda \to \infty} r(\lambda) \le 1 - p$, and completes the proof of Equation 61.

Recall the definition $\tilde{\lambda} = \inf\left\{\lambda > \frac{1-p}{p} \mid r(\lambda) \le 1 - \delta\right\}$. Equation 61 and Equation 60 together imply that $\tilde{\lambda}$ exists, since $\delta \in (0, p) \implies 1 - \delta \in (1 - p, 1)$. Also, Equation 59 and Equation 60 imply that $\tilde{\lambda} \to \frac{1-p}{p} = \lambda_0(p, \delta)$ as $\delta \to 0$.

We can now consider the random walk $X_t$ defined in Equation 57 with $\lambda = \tilde{\lambda}$. Our goal is to show that $z_{p,\tilde{\lambda}} \le 1 - \delta$, which implies that $\lambda_0(p, \delta) \le \tilde{\lambda}$. Let $\alpha = r(\tilde{\lambda}) \le 1 - \delta$. We have constructed $\alpha$ to be a root of $h_{\tilde{\lambda}}$, so that $\alpha^{X_t}$ is a martingale, as shown in Equation 58. Let $T_0 = \inf\{t \ge 0 \mid X_t \le 0\}$, and $T_b = \inf\{t \ge 0 \mid X_t \ge b\}$, where $b > 0$. Define $T = \min(T_0, T_b)$. We have $\alpha^{X_{\min(T,n)}}$ is bounded for any $n$ and it is nonnegative, therefore by optional sampling theorem and martingale convergence theorem (e.g., Theorem 4.8.2 in Durrett (2019)), we have

$$\alpha = \alpha^{X_0}$$
$$= \mathbb{E}\left[\alpha^{X_T}\right]$$
$$= \Pr(T_0 < T_b)\alpha^{X_{T_0}} + (1 - \Pr(T_0 < T_b))\alpha^{X_{T_b}}$$
$$\ge \Pr(T_0 < T_b) + (1 - \Pr(T_0 < T_b))\alpha^{b+\lambda},$$

where the inequality holds due to $\alpha \in (0, 1)$, $X_{T_0} \le 0$, and $X_{T_b} \le b + \lambda$. Let $b \to \infty$ on both sides, and note that $\alpha^b \to 0$, we have

$$\alpha \ge \Pr(T_0 < \infty) = z_{p,\tilde{\lambda}}. \qquad (64)$$

Therefore

$$z_{p,\tilde{\lambda}} \le \alpha \le 1 - \delta,$$

so that $\lambda_0(p, \delta) \leq \tilde{\lambda}$. Finally,

$$\lambda_0(p, 0) \leq \lim_{\delta \to 0^+} \lambda_0(p, \delta) \leq \lim_{\delta \to 0^+} \tilde{\lambda} = \lambda_0(p, 0),$$

so that $\lim_{\delta \to 0^+} \lambda_0(p, \delta) = \lambda_0(p, 0)$.

$\square$

**Lemma 16.** *Let $\{a_i\}_{i=0}^{\infty}$ be a positive sequence of reals satisfying $a_{i+1} = ra_i + b$ for $r > 1$, and let $A \geq a_0$. Define $k = \max\{i \geq 0 : a_i \leq A\}$. Then*

$$k = \left\lfloor \frac{\log\left(\frac{A(r-1)+b}{a_0(r-1)+b}\right)}{\log r} \right\rfloor$$

*Proof.* It is straightforward to show by induction that for any $i \geq 0$:

$$a_i = a_0 r^i + b \sum_{j=0}^{i-1} r^j = a_0 r^i + b \frac{r^i - 1}{r - 1} = r^i \left(a_0 + \frac{b}{r-1}\right) - \frac{b}{r-1}.$$

Then $a_i \leq A$ if and only if

$$r^i \left(a_0 + \frac{b}{r-1}\right) - \frac{b}{r-1} \leq A$$

$$r^i \left(a_0 + \frac{b}{r-1}\right) \leq A + \frac{b}{r-1}$$

$$r^i \leq \frac{A + \frac{b}{r-1}}{a_0 + \frac{b}{r-1}} = \frac{A(r-1)+b}{a_0(r-1)+b}$$

$$i \leq \frac{\log\left(\frac{A(r-1)+b}{a_0(r-1)+b}\right)}{\log r}.$$

So $k$ is the largest integer smaller than or equal to the RHS of the above.

$\square$

## E   DISCUSSION ON STABILIZATION CONSTANT $\gamma$

In Theorem 1, we showed that Decorrelated AdaGrad-Norm exhibits a quadratic dependence on $\Delta, L_1$ in the dominating term of its convergence rate, so that the number of iterations required to find an $\epsilon$-stationary point is $\Omega(\Delta^2 L_1^2 \sigma^2 \epsilon^{-4})$. This result depends on the condition $\gamma \leq \tilde{\mathcal{O}}(\Delta L_1)$, which covers the standard protocol in practice of choosing $\gamma$ to be a small constant, e.g. $\gamma = 1e - 8$. However, it is natural to ask whether our result can extend to any choice of $\gamma$.

In this section, we answer this question in the deterministic setting, that is, with $\sigma = 0$, we show that the lower bound of Theorem 1 can be recovered even if the condition $\gamma \leq \tilde{\mathcal{O}}(\Delta L_1)$ is removed. This shows that (deterministic) Decorrelated AdaGrad-Norm cannot recover the optimal complexity from the smooth (deterministic) setting, no matter the choice of $\gamma$. This result is stated below.

**Theorem 8.** *Denote $\mathcal{F}_{det} = \mathcal{F}_{as}(\Delta, L_0, L_1, 0)$, and let algorithm $A_{DAN}$ denote Decorrelated AdaGrad-Norm (Equation 1) with parameters $\eta, \gamma > 0$. Let $0 < \epsilon \leq \min\left\{\frac{\Delta L_1}{2}, \sqrt{\frac{\Delta \gamma}{4\eta}}\right\}$. If $\Delta L_1^2 \geq L_0$, then*

$$\mathcal{T}(A_{DAN}, \mathcal{F}_{det}, \epsilon) \geq \tilde{\Omega}\left(\frac{\Delta^2 L_1^2}{\epsilon^2}\right).$$

The proof structure is similar as Theorems 1, 2, and 3, by splitting into cases depending on the choice of $\eta$ and $\gamma$. However, for this proof we split into cases slightly differently than in these three theorems; here, the cases are determined by the magnitude of $\eta$ and $\gamma/\eta$. The proof relies on Lemma 1 for one case and reuses the hard instance of Lemma 4 for the other.

*Proof.* We consider two cases: In the first case, both of the following hold:

$$\eta \geq 1/L_1 \quad \text{and} \quad \gamma/\eta \leq \frac{\Delta L_1^2}{8 \log \left(1 + \frac{48 \Delta L_1^2}{L_0}\right)}. \tag{65}$$

In the second case, one or both of these two conditions fail:

$$\eta \leq 1/L_1 \quad \text{or} \quad \gamma/\eta \geq \frac{\Delta L_1^2}{8 \log \left(1 + \frac{48 \Delta L_1^2}{L_0}\right)}. \tag{66}$$

We will show that in the first case, there exists an objective for which Decorrelated AdaGrad-Norm will never converge, and in the second case, there exists an objective for which convergence requires $\Omega(\Delta^2 L_1^2 \epsilon^{-2})$ iterations.

**Case 1** This case is the simpler of the two, since we can directly apply Lemma 5. The conditions of this lemma are immediately satisfied by the conditions of this case. Therefore, in Case 1, there exists an objective $(f, g, \mathcal{D}) \in \mathcal{F}_{\text{det}}$ such that $\|\nabla f(\boldsymbol{x}_t)\| \geq \Delta L_1$ for all $t \geq 0$.

**Case 2** For this case, we will reuse the hard instance from Lemma 8. Denoting $m = \frac{1}{L_1} \log \left(1 + \frac{L_1 \epsilon}{L_0}\right)$, the objective is defined as

$$f(x) = \begin{cases} -\epsilon(x + m) + \psi(m) & x < -m \\ \psi(x) & x \in [-m, m] \\ \epsilon(x - m) + \psi(m) & x > m \end{cases},$$

where

$$\psi(x) = \frac{L_0}{L_1^2} \left(\exp(L_1 |x|) - L_1 |x| - 1\right).$$

With $g, \mathcal{D}$ defined so that $g(x, \xi) = f'(x)$ almost surely when $\xi \sim \mathcal{D}$, it was already shown in the proof of Lemma 8 that $(f, g, \mathcal{D}) \in \mathcal{F}_{\text{det}}$, when we use the initial point $x_0 = m + \frac{\Delta}{2\epsilon}$.

Letting $x_t$ be the sequence of iterates generated by Decorrelated AdaGrad-Norm, we define $t_0 = \max\{t \geq 0 \mid x_t \geq m\}$. Notice that $f'(x) = \epsilon$ for all $x \geq m$, so the definition of $t_0$ implies that $|f'(x_t)| = \epsilon$ for all $t \leq t_0$. Accordingly, we want to show that

$$t_0 \geq \tilde{\Omega}\left(\frac{\Delta^2 L_1^2}{\epsilon^2}\right).$$

Actually, the trajectory of Decorrelated AdaGrad-Norm for this objective is identical to that of Decorrelated AdaGrad, since the objective's domain is one-dimensional. Therefore, to analyze the trajectory $x_t$, we can reuse the analysis from the proof of Lemma 8. Starting from Equation 22,

$$\frac{t_0}{\sqrt{\gamma^2 + t_0 \epsilon^2}} \geq \frac{1}{\epsilon}\left(\frac{\Delta}{4\eta\epsilon} - \frac{\epsilon}{2\gamma}\right).$$

Using the assumed upper bound on $\epsilon$,

$$\epsilon \leq \sqrt{\frac{\Delta\gamma}{4\eta}}$$

$$\epsilon^2 \leq \frac{\Delta\gamma}{4\eta}$$

$$\frac{\epsilon}{2\gamma} \leq \frac{\Delta}{8\eta\epsilon},$$

so

$$\frac{t_0}{\sqrt{\gamma^2 + t_0 \epsilon^2}} \geq \frac{\Delta}{8\eta\epsilon^2}$$

$$8\eta\epsilon^2 t_0 \geq \Delta\sqrt{\gamma^2 + \epsilon^2 t_0}$$

$$64\eta^2 \epsilon^4 t_0^2 \geq \Delta^2 \gamma^2 + \Delta^2 \epsilon^2 t_0$$

$$t_0^2 \geq \frac{\Delta^2 \gamma^2}{64\eta^2 \epsilon^4} + \frac{\Delta^2}{64\eta^2 \epsilon^2} t_0.$$

Denoting $b = \frac{\Delta^2}{64\eta^2\epsilon^2}$ and $c = \frac{\Delta^2\gamma^2}{64\eta^2\epsilon^4}$, this gives the quadratic inequality

$$t_0^2 - bt_0 - c \geq 0.$$

Since $t_0 > 0$, this implies

$$t_0 \geq \frac{b + \sqrt{b^2 + 4c}}{2} \geq \frac{b}{2} + \sqrt{c} = \frac{\Delta^2}{128\eta^2\epsilon^2} + \frac{\Delta\gamma}{8\eta\epsilon^2}. \tag{67}$$

Finally, we can apply the conditions on $\eta$ and $\gamma/\eta$ from the case analysis. We know that either

$$\eta \leq 1/L_1 \quad \text{or} \quad \gamma/\eta \geq \frac{\Delta L_1^2}{8\log\left(1 + \frac{48\Delta L_1^2}{L_0}\right)}. \tag{68}$$

If $\eta \leq 1/L_1$, then Equation 67 implies

$$t_0 \geq \frac{\Delta^2}{128\eta^2\epsilon^2} \geq \frac{\Delta^2 L_1^2}{\epsilon^2}.$$

On the other hand, if

$$\gamma/\eta \geq \frac{\Delta L_1^2}{8\log\left(1 + \frac{48\Delta L_1^2}{L_0}\right)},$$

then Equation 67 implies

$$t_0 \geq \frac{\Delta\gamma}{8\eta\epsilon^2} \geq \frac{\Delta^2 L_1^2}{64\epsilon^2\log\left(1 + \frac{48\Delta L_1^2}{L_0}\right)}.$$

Either way, we have

$$t_0 \geq \tilde{\Omega}\left(\frac{\Delta^2 L_1^2}{\epsilon^2}\right),$$

which finishes the analysis for Case 2.

**Putting The Cases Together** The case analysis above shows that, no matter the choice of $\gamma, \eta$, there always exists some objective $(f, g, \mathcal{D}) \in \mathcal{F}_{\text{det}}$ such that the number of iterations to find an $\epsilon$-stationary point is at least

$$\tilde{\Omega}\left(\frac{\Delta^2 L_1^2}{\epsilon^2}\right).$$

$\square$

