# OpenReview forum: "Complexity Lower Bounds of Adaptive Gradient Algorithms for Non-convex Stochastic Optimization under Relaxed Smoothness"
_ICLR.cc/2025/Conference — ICLR 2025 Poster_

### Official Review · Reviewer_AVzC · 2024-10-27

**Soundness:** 4
**Presentation:** 4
**Contribution:** 3
**Rating:** 8
**Confidence:** 4

**Summary:**

This paper investigates the complexity of adaptive optimization algorithms in non-convex stochastic settings under $(L_0,L_1)$-smoothness assumption. The study addresses an open question regarding whether the complexity of these adaptive algorithms, which typically have higher-order dependencies on problem parameters, can be tightened.

The authors establish complexity lower bounds for different variants of AdaGrad, focusing on the dependency on parameters $Δ$, $L_0$, and $L_1$. They provide:
- A complexity lower bound for the Decorrelated AdaGrad-Norm variant.
- A lower bound for Decorrelated AdaGrad under specific hyperparameter conditions.
- A lower bound for SGD with a broad class of adaptive stepsizes.

The findings of the paper show that, for certain adaptive algorithms, the $(L_0, L_1)$-smooth setting is fundamentally more difficult than the standard smooth setting.

**Strengths:**

- The paper is clearly structured, with each theorem building on the previous results to form a coherent narrative.

- The theoretical contributions are solid, with clear mathematical rigor and well-constructed complexity bounds. By investigating multiple AdaGrad variations, the paper thoroughly explores different adaptive step-size strategies under non-standard conditions.

- The results significantly contribute to the understanding of optimization in non-convex and non-smooth settings, highlighting limitations of popular adaptive methods when applied to challenging optimization landscapes.

**Weaknesses:**

N/A

**Questions:**

N/A

---

> ### Author Response · Authors · 2024-11-21
>
> Thank you for the positive review. Please let us know if you think of any questions we can answer.

---

### Official Review · Reviewer_Bg2B · 2024-11-03

**Soundness:** 4
**Presentation:** 3
**Contribution:** 3
**Rating:** 6
**Confidence:** 3

**Summary:**

Recent advances in theory and optimization of transformers and particularly adaptive algorithms have shed light on a new type of assumption called $(L_0, L_1)$-smoothness. This paper proposes novel non-convex lower bound instances satisfying the $(L_0, L_1)$-smoothness condition for various types of adaptive algorithms including (decorrelated) AdaGrad and AdaGradNorm. The results show an iteration complexity lower bound of $\Omega(\Delta^2 L_1^2 \epsilon^{-4})$ (naive, omitting $\sigma, \gamma$) to achieve stationarity in expectation, which shows a gap with previous results on $L$-smooth functions. The paper also provides a new lower bound for a class of single-step adaptive SGD algorithms in terms of an iteration complexity lower bound of $\Omega(\Delta L_0 \epsilon^{-4})$ (naive) to achieve stationarity with high probability.

**Strengths:**

- The paper suggests novel lower bounds through diligent combinations of previous lower bound techniques.
- The overall writing is clear and easy to follow (except for only a few parts and bits of typos), and the proofs in the appendices are also organized well.
- The paper relevantly compares results with previous literature, provides rich discussion points, and is transparent about limitations.

**Weaknesses:**

- The lower bounds in Theorem 1 require high dimensions like $d = \Omega(\Delta^2 L_1^2 \sigma^2 \epsilon^{-4})$ which is quite large and implies that, for a fixed $d$, the proposed lower bound can only cover up to a finite range of $T$.
- As pointed out in the paper, the results (especially Theorems 1-3) are for specific algorithms rather than a class of algorithms. Although it seems that it will be quite difficult, it could be interesting to see if the lower bound results can extend to a *class* of Adam-type algorithms (Chen et al., 2019).
- This is a minor thing, but I think Section 6 and Theorem 4 could be organized better. It is hard to get what the new constants exactly are without reading Appendix C.1, especially the constants involving $\zeta$. IMHO, it would have been a bit easier to understand if the details about the constants (including the random walk parts) were all in the former part of Section 6. (I reckon that it was written this way just for now because of the page limit.)

**Questions:**

- In Theorem 1, the proof uses a 1-dimensional instance for the large $\eta$ case and a large dimensional instance for the small $\eta$ case. The $T = \Theta(\epsilon^{-4})$ condition for the latter case results in a dimension of $d = \Omega(\epsilon^{-4})$ as stated in W1. On the other hand, in Theorems 2 and 3, the proof is quite the opposite: it uses a $d \ge T$ dimensional instance for the large $\eta$ case which does NOT have requirements like $T = \Theta(\epsilon^{-4})$, and a 1-dimensional instance for the small $\eta$ case. Why can’t we use a similar approach to DAN to avoid high dimension requirements, or what are the technical difficulties in doing so?
- In Theorem 3, the lower bound holds when $\gamma \le \sigma$, but the rates themselves do not contain $\gamma, \sigma$ anywhere, which is probably because they cancel out somewhere if we follow the same procedure as the proof of Theorem 2. Then what can we say about the case of $\gamma > \sigma$ for now? By definition of AdaGrad (without decorrelation), I reckon that we must choose a not-too-small $\gamma$ so that the first step of the algorithm doesn’t ‘explode’, and I’m just curious about what behavior we could see or expect from this lower bound instance if we send the noise to zero for a fixed $\gamma > 0$. (The ‘better noise-dependency’ argument of AdaGrad over DAG is also true only when $\gamma \le \sigma$.)
- Why exactly does Theorem 4 derive iteration complexity lower bounds in terms of high probability (instead of expectation)? Are they just completely different results, or is this sort of a stronger result than getting lower bounds in terms of expectation (from the same instance)? (I think the two types of bounds are fundamentally different in general, but I’m asking this because the approach towards getting the lower bound instance with $\| \nabla f(\boldsymbol{x}_t) \| \ge \epsilon$ seems to share quite a similar spirit with those of Theorems 1-3.)
- Light Question: Is it possible to obtain other lower bound results analogous to Theorem 4 if we focus on the (stronger) bounded-noise assumption?
- Light Question: Can you compare Theorem 4 with the upper bounds in Section 7 of Gorbunov et al. (2023) which also has a few results assuming the finite-sum oracle setting? For SGD-PS (Algorithm 6, Gorbunov et al. (2023)), is this algorithm incomparable with the lower bound because it uses the values $f_i(\boldsymbol{x}^{\star})$ in the step sizes?
- **Small Typos.** In p.17, I think we should remove the subscript $t$, as in $f(\boldsymbol{x}) = \epsilon \langle \boldsymbol{x}, \textbf{e}_1 \rangle + \sum\_{i=2}^T h_i ( \langle \boldsymbol{x}, \textbf{e}_i \rangle ) $.

    In p.18, $j(\boldsymbol{x}) = i+2$ $\Rightarrow$ $j(\boldsymbol{x}_i) = i+2$.

    In p.21, $D_1 \le \dots \le \Delta/6 + \Delta/6 + \Delta/6 = \Delta/2$, and some constants after this part should be slightly modified according to this.

---

> ### Author Response · Authors · 2024-11-21
> **Response 1/2**
>
> Thank you for the positive review and helpful comments. We have responded to your questions and concerns below.
>
> **W1: High dimensional objectives** You are correct that the hard instances in our lower bounds are high-dimensional, i.e. $d \geq T$. However, this is a common situation for lower bounds of first-order algorithms, such as [(Arjevani et al, 2023)](https://arxiv.org/abs/1912.02365) and many classical lower bounds from [(Nesterov, 2013)](https://link.springer.com/book/10.1007/978-3-319-91578-4). To the best of our knowledge, there are no lower bounds using fixed dimension which can match the same bounds as these high-dimensional results.
>
> **W2: Generalizing for Adam** It is possible that our results could generalize to Adam-type algorithms, but there are some technical difficulties. In short, some parts of our analysis can be used to analyze Adam, though it will require further work to establish a complete analysis of Adam.
>
> To extend Theorems 2/3 for Decorrelated/Original Adam, we need to establish analogous results to Lemmas 3 and 4. Interestingly, the hard instance from Lemma 3 can be reused for Adam, and we can show that Adam will diverge on the hard instance when $\eta \geq \gamma/(L_1 \sigma) \log(1 + L_1 \epsilon/L_0)$ (decorrelated Adam) or $\eta \geq 1/L_1 \log(1 + L_1 \epsilon/L_0)$ (original Adam). This result is enabled by the fact that the trajectory of Adam is nearly identical to that of AdaGrad for our specific hard instance, which follows from an important property of our construction: each coordinate of the input has zero stochastic gradient for every timestep except for one. Therefore, most of the gradient history is zero, so the moving averages in the numerator and denominator of Adam's will behave very similarly to AdaGrad's update.
>
> However, Lemma 4 is not as easy to extend. Since Adam has a moving average in the denominator instead of a sum (as in AdaGrad), it's behavior on the hard instance of Lemma 4 will differ significantly from AdaGrad: the same hard instance does not yield the same complexity. So extending the analysis to Adam would require a new hard instance to replace the one in our Lemma 4.
>
> Overall, it seems promising to reuse both the overall proof structure and Lemma 3 to analyze Adam, but finishing the proof will require a new construction to fill the hole left by Lemma 4. We leave this question to future work.
>
> **W3: Explanation of Theorem 4 constants** Thank you for the feedback, and you are correct that the length put some limits on the amount of exposition that we can put in the main body. These constants are error terms arising from the probability of divergence of the biased random walk from Section C.2, and their order depends on $\sigma_2$. To provide a sense for these constants without specifying all technical details, we describe their order for different regimes of $\sigma_2$ and $\delta$ on page 9.
>
> **Q1: Difficulties of unified analysis** This is a good observation. There are many technical details that create difficulty in simultaneously analyzing AdaGrad-Norm (shared learning rate) and AdaGrad (coordinate-wise learning rate), so let us touch on one. The source of the difficulty is that with coordinate-wise learning rates, each coordinate is affected only by the history of gradients for that particular coordinate, but not for other coordinates.
>
> In Lemma 3, the coordinates of the objective correspond to time steps in the trajectory, and each coordinate sees a nonzero gradient for exactly one timestep. With coordinate-wise learning rates, each coordinate is unaffected by previous history, so Decorrelated AdaGrad and AdaGrad can be analyzed together. Back to the original question, Decorrelated AdaGrad-Norm uses a shared learning rate for each coordinate, so we cannot separate the behavior of each coordinate into separate timesteps, and the history becomes an important factor in the analysis. This is one reason why the objective from Lemma 1 is so different from that of Lemma 3, and it is not clear whether all three of these algorithms could have a unified analysis.

---

> ### Author Response · Authors · 2024-11-21
> **Response 2/2**
>
> **Q2: $\gamma$ requirement of Theorem 3** In the case that $\gamma > \sigma$, our current constructions cannot ``force divergence" of AdaGrad in the case that $\eta \geq 1/L_1$, which is a key component of our analysis. The reason is that our construction relies on noise in the stochastic gradient to force the algorithm along a trajectory where $\lVert \nabla F(x_t) \rVert$ never decreases. If $\gamma > \sigma$, then the noise in the stochastic gradient is dominated by the $\gamma$ in the adaptive learning rate denominator, and the step size is too small to follow this trajectory. This is not to say that the choice $\gamma > \sigma$ is impossible to handle, but doing so will likely require some new construction.
>
> Also, with AdaGrad there is no fear of the algorithm "exploding", since even when $\gamma = 0$ the denominator of the adaptive learning rate is always larger than the magnitude of the stochastic gradient. This means that the update size of AdaGrad is bounded by $\eta$, no matter the choice of $\gamma$.
>
> Lastly, can you elaborate what you meant by "The 'better noise-dependency' argument of AdaGrad over DAG is also true only when $\gamma \leq \sigma$"? We believe that this better noise-dependence of AdaGrad over DAG still holds when $\gamma > \sigma$, since the update size of AdaGrad is bounded by $\eta$, but the update size of DAG can grow with $\sigma$.
>
> **Q3: High probability analysis** Theorem 4 uses a high-probability analysis because it relies on the probability of divergence of a biased random walk (as opposed to the expectation of a random walk variable after a given time). You are correct that the structure of the arguments of Theorems 1-3 is similar in spirit to that of Theorem 4, but these two settings use very different technical tools, and this leads to the two different types of guarantees.
>
> **Q4: Affine noise in Theorem 4** Actually, the complexity of Single-Step Adaptive SGD is already completely characterized by existing work, since the upper bound of gradient clipping [(Zhang et al, 2020a)](https://arxiv.org/abs/2010.02519) matches the lower bound of adaptive SGD [(Drori and Shamir, 2020)](https://arxiv.org/abs/1910.01845), so that the best complexity of Single-Step Adaptive SGD is $\mathcal{O}(\Delta L_0 \sigma^2 \epsilon^{-4})$, which recovers the optimal rate from the smooth setting. Note that the lower bound of [(Drori and Shamir, 2020)](https://arxiv.org/abs/1910.01845) was introduced for the smooth setting, so their hard instance is consequently relaxed smooth. It also has almost surely bounded gradient noise. Since the complexity in this setting is already completely characterized, we focus on the slightly harder setting of affine noise.
>
> **Q5: Comparison with (Gorbunov et al, 2023)** Can you please provide the specific reference of (Gorbunov et al, 2023)? We are not sure to which paper you refer.

---

> ### Comment · Reviewer_Bg2B · 2024-11-24
>
> Thanks for the detailed response, sorry for the late reply, and sorry again for forgetting to paste the reference and the clumsy typo in the year. The paper I tried to refer to in **Q5** is the one below, and I was just curious if we can compare Algorithms 5 and 6 of (Gorbunov et al., 2024) and the lower bound in this work regarding single-step adaptive SGD (Theorem 4).
>
> Eduard Gorbunov, Nazarii Tupitsa, Sayantan Choudhury, Alen Aliev, Peter Richtárik, Samuel Horváth, Martin Takáč. Methods for Convex $(L_0, L_1)$-Smooth Optimization: Clipping, Acceleration, and Adaptivity. arXiv 2409.14989
>
> Regarding **Q2**, my initial naive thought was based on the former terms in the lower bounds in Theorem 2 and 3 are of order $\frac{\Delta^2 L_0^2 \sigma^2}{\gamma^2 \epsilon^4}$ and $\frac{\Delta^2 L_0^2}{\epsilon^4}$, respectively. If $\sigma > \gamma$, then losing a factor of $\frac{\sigma^2}{\gamma^2}$ does not directly imply *better noise dependence*. I agree with the high-level idea that AdaGrad is less sensitive to noise when $\sigma > \gamma$.

---

> ### Author Response · Authors · 2024-11-24
>
> **Q5**: Thank you for the clarification. Algorithms 5 and 6 from (Gorbunov et al, 2024) may at first appear to have the "single-step adaptive" structure in the update rules, but actually the stepsizes from these algorithms do not only depend on the current gradient norm: they also depend on the current objective value and the optimal objective value for the currently sampled data. This means that our Theorem 4 does not apply to these algorithms, since our definition of "single-step adaptive" stepsizes only consider stepsizes which are a function of the current gradient norm. Another important difference between our work and that of (Gorbunov et al, 2024) is the theoretical setting: these algorithms are considered in the convex, finite-sum setting, whereas we consider the non-convex, stochastic setting. We will cite this paper in the revised version and add detailed discussions.
>
> **Q2**: Ah, your comment makes sense now. You are correct that losing the factor of $\sigma^2/\gamma^2$ does not imply a better dependence on $\sigma$ for the iteration complexity. And yes, the high-level idea about AdaGrad's sensitivity to noise is a key difficulty in getting an improved lower bound for AdaGrad.

---

### Official Review · Reviewer_BkD6 · 2024-11-04

**Soundness:** 2
**Presentation:** 3
**Contribution:** 2
**Rating:** 6
**Confidence:** 3

**Summary:**

This paper studies the complexity **lower bounds** for certain adaptive gradient (descent) algorithms for non-convex optimization problems under relaxed smoothness assumptions, i.e., the objective function $f$ is continuously differentiable and $(L_0, L_1)$-smooth as stated in assumption 1. More specifically, the results are in Table 1 (line 77-84). From 4 lower bounds derived for AdagGad variants, the paper concluded that the Adagrad-type algorithms cannot converge under relaxed smoothness **without** a higher-order polynomial complexity that depends on the problem parameters $\nabla, L_0, \sigma, L_1, \epsilon$.

**Strengths:**

- The presentation of this paper is well-structured. For example, it effectively presents the related complexity upper bounds in Table 1 and provides clear proof intuitions along with key takeaways for each theorem.
- The results of this paper indicate that for decorrelated AdaGrad-Norm, the algorithm cannot achieve the optimal complexity attainable under a stronger smoothness assumption. This finding suggests a meaningful performance gap between $L$-smooth and $(L_0, L_1)$ smoothness conditions.

**Weaknesses:**

- As noted in the limitations, decorrelated AdaGrad is not commonly used in practice; therefore, proving a lower bound for decorrelated AdaGrad may lack significant technical novelty.

**Questions:**

- Assumption 2 appears to be rather strong. Are there existing references that support the validity of these assumptions?

---

> ### Author Response · Authors · 2024-11-21
>
> Thank you for your positive review and comments. Below we have answered your question.
>
> **W1: Novelty of decorrelated results** As you pointed out, we discussed this point in our limitations section. However, we believe that proving these lower bounds for the decorrelated algorithms still requires significant technical novelties. Our Lemmas 1 and 3 contain constructions of novel hard instances for these algorithms (Lemma 3 applies to both decorrelated and original AdaGrad).
>
> **Q1: Gradient noise assumptions** Assumption 2 states many variations on the stochastic gradient noise assumption, and the main one considered in our paper is almost surely bounded noise. This assumption is common in the literature on relaxed smoothness [(Zhang et al, 2020b)](https://arxiv.org/abs/1905.11881), [(Zhang et al, 2020a)](https://arxiv.org/abs/2010.02519), [(Crawshaw et al, 2022)](https://arxiv.org/abs/2208.11195). Our last theorem also considers the assumption of affine noise, which has also been used for both smooth and relaxed smooth optimization [(Bottou et al, 2016)](https://arxiv.org/abs/1606.04838), [(Faw et al, 2023)](https://arxiv.org/abs/2302.06570), [(Attia and Koren, 2023)](https://arxiv.org/abs/2302.08783).

---

### Official Review · Reviewer_Smgh · 2024-11-04

**Soundness:** 3
**Presentation:** 3
**Contribution:** 2
**Rating:** 6
**Confidence:** 2

**Summary:**

This paper investigates the theoretical lower bounds for finding $\epsilon$-stationary points using adaptive gradient algorithms under $(L_0,L_1)$-smoothness. With the assumption of bounded noise, the authors prove that the complexities of AdaGrad and its variants have a greater dependency on the problem parameters $\Delta$, $L_0$, $L_1$ compared to their $L$-smooth counterparts.

**Strengths:**

In my opinion, these results reveal the potential difficulty of adaptive methods under relaxed smoothness and add to the current understanding within the community.

**Weaknesses:**

The primary weakness of this paper is that the analysis is heavily relies on update policies similar to AdaGrad. Although the authors do investigate the behavior of adaptive SGD, they assume the noise has a less common affine property, and their results do not extend to popular variants of Adam.

**Questions:**

**Typos and minor suggestions:**

Line 81: the theorem for AdaGrad should be "Theorem 3".

Line 287: Table 1 provides the result of AdaGrad-Norm under affine noise, which is not equivalent to Equation 5.

Line 746: the logic of Equation 7 is not easy to comprehend at first glance, as it appears to tighten the condition for $\ell_{t}\geq 4 m_{t+1}$. I suggest that the authors should explicitly point out this relationship.

Line 774: "$\log x\leq 1+x$" -> "$\log(1+x)\leq x$".

Line 797: missing explanation for step (i).

Line 908: "$\mathbf{x}_t$" -> "$\mathbf{x}$".

Line 927-929: "$L$" -> "$L_0$".

Line 958-959: missing \\rangle for $\langle\mathbf{x}_i,\mathbf{e}_j\rangle$.

Line 1012: "$t\geq 1$".

Line 1134: "Lemma 1" -> "Lemma 2".



**Questions:**

* We note that Assumption 1 is generally less rigid than the typical assumption proposed by Zhang et al. (2019). Specifically, Assumption 1 aligns with the $\mathcal{L}^*_{\mathrm{asym}}$ function class introduced by Chen et al. (2023). Nonetheless, the lower bounds established in this paper should still hold under the typical assumption.
* Since the constructive proof of Lemma 2 is rather complicated, I encourage the authors to provide necessary remarks following the presentation of the results. Specifically, we are interested in how the higher dependency on $\Delta$ and $L_1$ emerges in the dominant term $\mathcal{O}(\Delta^2L_1^2\sigma^2\epsilon^{-4})$. The same applies to Lemma 4.
* In Line 484, the authors state that "the lower bound goes to $0$ when $\sigma_2\rightarrow 1$". However, the dominant term $\mathcal{O}(\Delta L_0\sigma^2\epsilon^{-4})$ does not approach zero, as $\sigma_1>0$.
* I encourage the authors to discuss the results of Li et al. (2023). They provide analyses for Adam under relaxed assumptions, which covers the case of $(L_0,L_1)$-smoothness.



References:

Jingzhao Zhang, Tianxing He, Suvrit Sra, Ali Jadbabaie. Why gradient clipping accelerates training: A theoretical justification for adaptivity. In International Conference on Learning Representations, 2019.

Ziyi Chen, Yi Zhou, Yingbin Liang, Zhaosong Lu. Generalized-Smooth Nonconvex Optimization is As Efficient As Smooth Nonconvex Optimization. In Proceedings of the 40th International Conference on Machine Learning, 2023.

Haochuan Li, Alexander Rakhlin, Ali Jadbabaie. Convergence of Adam Under Relaxed Assumptions. In In Advances in Neural Information Processing Systems 35, 2023.

---

> ### Author Response · Authors · 2024-11-21
>
> Thank you for your review and comments. Below we have responded to your questions and concerns.
>
> **W: Affine noise** You mentioned that we only consider the affine noise setting for adaptive SGD. We want to clarify that adaptive SGD in the bounded setting is already known to achieve the optimal rate $\mathcal{O}(\Delta L_0 \sigma^2 \epsilon^{-4})$ [(Zhang et al, 2020a)](https://arxiv.org/abs/2010.02519), so the setting of bounded noise is already resolved. This is our motivation for studying affine noise, which is a slightly harder setting for optimization.
>
> **T2: Comparison with (Wang et al, 2023)** You said that "Table 1 provides the result of AdaGrad-Norm under affine noise, which is not equivalent to Equation 5." We should clarify that Equation 5 states the upper bound of [(Wang et al, 2023)](https://arxiv.org/abs/2305.18471) in the case of bounded noise, which is a special case of their affine variance result stated in Table 1. We consider this special case in Equation 5 in order to compare against our lower bounds, which consider bounded noise.
>
> **Q1: Relaxed smoothness definition** Our definition of relaxed smoothness is a slightly weaker version which does not require the function to be twice-differentiable, and this version was shown to imply the original version [(Zhang et al, 2020a)](https://arxiv.org/abs/2010.02519). Also, since all of our hard instances are twice differentiable, our results still apply for the original version of relaxed smoothness. Please let us know if we have answered your question.
>
> **Q2: Parameter dependence explanation** Thank you for the suggestion. $L_1$ appears in our lower bounds because AdaGrad-type algorithms cannot operate in two stages, unlike gradient clipping. For example, Decorrelated AdaGrad-Norm must set $\eta \leq 1/L_1$ to avoid divergence on some exponential functions, and this choice of $\eta$ will affect every single update, even when the algorithm has nearly converged. On the contrary, gradient clipping can avoid divergence with a proper choice of the clipping threshold, and the learning rate can be chosen independently of $L_1$, so that when the algorithm is close to converging, the update size is unaffected by $L_1$. AdaGrad lacks this ability to branch into two options depending on the gradient norm, and this causes the additional dependence on $L_1$.
>
> **Q3: Incorrect description of Theorem 4** You are correct, it should say that the term with quadratic $\Delta, L_1$ goes to 0 instead. We have updated the paper to fix this.
>
> **Q4: Reference to (Li et al, 2023)** We have referenced [(Li et al, 2023)](https://arxiv.org/abs/2304.13972) in our related works, and we have added a comment to distinguish this work on Adam from other works on AdaGrad-Norm. Note that this work also exhibits a higher order polynomial dependence on $L_1$.

---

> > ### Comment · Reviewer_Smgh · 2024-11-25
> >
> > Thank you for your response. My concerns have been addressed, and I will maintain my positive score.

---

### Official Review · Reviewer_sfBG · 2024-11-05

**Soundness:** 3
**Presentation:** 4
**Contribution:** 2
**Rating:** 5
**Confidence:** 3

**Summary:**

This paper analyzes complexity lower bounds for AdaGrad variants under $(L_0, L_1)$-smoothness conditions. The investigation includes Decorrelated AdaGrad (in both norm and coordinate-wise versions), as well as AdaGrad and SGD with current step-based adaptive stepsizes. A key finding is that optimization under $(L_0, L_1)$-smoothness is fundamentally more challenging than standard $L$-smoothness, as evidenced by lower bounds showing quadratic dependence on constants rather than the linear dependence on $L$ and $\Delta$ typically seen in $L$-smooth settings.

**Strengths:**

- The paper is well-written, easy to follow, and well-organized.
- The problem it investigates is interesting and important to the study of adaptive algorithms in non-convex settings, given that the discussion of constant dependence in non-convex and $(L_0, L_1)$-smoothness settings is still not rich.
- The results are novel and provide some theoretical insights into adaptive gradient algorithms.

**Weaknesses:**

The reviewer has two major concerns about this paper:

- The definition of $(L_0, L_1)$-smoothness (Assumption 1) differs from most existing work in this topic ($(L_0, L_1)$-smoothness). Assumption 1 says the inequality holds for any $x$ and $y$. Essentially, for any $y$, we can set $x$ to be a stationary point and, without loss of generality, assume this point is the origin. This implies that $\\|\nabla f(y)\\| \leq L_0 \\|y\\|$, which means the growth of $\\nabla f(y)$ is linear. I guess this is $L$-smoothness.
- Another major concern is the restriction of hyper-parameter selection in the lower bounds. In Theorems 1 and 3, parameter $\gamma$ is required to be small. In Theorem 2, $\\gamma$ appears in the lower bound. While it’s understandable that a small $\gamma$ may be practical, shouldn’t lower bounds typically be related to the best algorithm within a family of algorithms? Parameter tuning should be allowed if one want to explore the limits of an algorithm family. In this case, I’m not sure how insightful these results are, particularly Theorem 2.

Other concerns are relatively minor:


- As mentioned in the limitation section, the lower bound for AdaGrad feels weak because it doesn’t depend on $\\sigma$, especially the $1/\\epsilon^4$ term. Lower bounds for the original AdaGrad should be more interesting than the decorrelated ones, which are not commonly used in practice but are primarily for analytical convenience. However, the difference in lower bounds would also be interesting if there are fundamental differences between decorrelated and the original versions, as mentioned in the paper, “so that the update size of AdaGrad is not as sensitive to noise as the decorrelated counterpart.” However, from the existing upper bounds, it doesn’t seem so.
- The lower bound for Single-step Adaptive SGD is also weak due to the affine-noise assumption. Therefore, we cannot compare it to existing rich upper bounds for algorithms like gradient clipping, which usually assume bounded noise or bounded variance.

**Questions:**

Could the author clarify the points in Weakness? If there are misunderstandings in these comments, I would like to increase my score.

Update: After reading the author’s response, my first major concern was addressed. However, the second concern persists. The lower bound in Theorem 2 may be vacant when $\\gamma$ is allowed to be tuned. I have adjusted my score accordingly.

---

> ### Author Response · Authors · 2024-11-21
>
> Thank you for your helpful comments on our paper. We have responded to your individual points below.
>
> **W1: Relaxed smoothness definition** Your concern actually comes from a typo: all of our proofs use the same definition of $(L_0, L_1)$-smoothness as previous work. We have updated the paper to fix this typo by including the condition $\lVert x - y \rVert \leq 1/L_1$. Since this was a major concern for you, we hope that you will reconsider your score.
>
> **W2: Condition for $\gamma$** Thank you for pointing this out. There are two points we would like to make about this. First, we investigated whether the condition can be removed from Theorem 1, and we succeeded in removing the condition for the deterministic setting while preserving the lower bound of $\Omega(\Delta^2 L_1^2 \epsilon^{-2})$. This additional result is included in Appendix E of our revised submission; please see Appendix E for a complete discussion of this new result. Removing this condition for the stochastic setting remains open. This leads us to our second point: for the stochastic setting, we cover the practical regime where $\gamma$ is chosen as a small constant (in Pytorch, the default value of the stabilization constant is $10^{-8}$). We agree that requiring $\gamma \leq \tilde{O}(\Delta L_1)$ is a theoretical limitation, but we believe that our results still capture the behavior of these algorithms with practical choices of hyperparameters.
>
> **W3: Original AdaGrad** As you mentioned, we did discuss this point in our limitations section. Although the lower bound for the original AdaGrad can likely be improved, we believe that our results for the decorrelated variants are an important first step towards understanding the original algorithm. This perspective was also taken by [(Li and Orabona, 2019)](https://arxiv.org/abs/1805.08114).
>
> **W4: Affine Noise** The reason that we focus on affine noise for Single-Step Adaptive SGD is that under the bounded noise assumption, there are single-step adaptive algorithms that are known to achieve the optimal rate. The prime example is gradient clipping, which was shown to achieve $\mathcal{O}(\Delta L_0 \sigma^2 \epsilon^{-4})$ by [(Zhang et al, 2020a)](https://arxiv.org/abs/2010.02519), and this matches the lower bound for SGD (with any adaptive learning rate) from [(Drori and Shamir, 2020)](https://arxiv.org/abs/1910.01845). Note that this lower bound uses a hard instance that is smooth (and therefore relaxed smooth) and with almost surely bounded noise. Therefore the analysis of Single-Step Adaptive SGD for the bounded noise case is already tight, and relaxed smoothness does not add any difficulty compared to smoothness. Because of this, we consider the affine noise assumption for Single-Step Adaptive SGD.

---

> > ### Author Response · Authors · 2024-11-25
> >
> > Thank you again for your reviewing efforts. We are following up with a gentle reminder that the discussion period ends tomorrow, so please let us know if we have addressed your concerns or if you have any more questions. In particular, your first major concern is due to our typo, and we have added a new result to address your second concern for the deterministic setting. With these additions, we kindly ask that you reconsider the score. Thank you.

---

### Author Response · Authors · 2024-11-21
**General Response**

Thank you to all of the reviewers for your efforts in reviewing our paper. We
have responded individually to each of your reviews, and we summarize the
major changes to our paper here. Changes to the submission are shown in red text, for
ease of understanding the modifications.

1. We have fixed a typo in the definition of $(L_0, L_1)$-smoothness (pointed out by reviewer sfBG). Our definition of $(L_0, L_1)$-smoothness now matches that of [(Zhang et al, 2020a)](https://arxiv.org/abs/2010.02519), which is standard in the literature.

2. In response to a comment by reviewer sfBG, we have added an additional result in Appendix E which removes the condition on $\gamma$ in Theorem 1 for the setting of deterministic gradients, while recovering the same lower bound as in Theorem 1. Please see Appendix E for a complete description of this new result.

---

### Meta-Review · Area_Chair_J4Ak · 2024-12-18

**Metareview:**

This paper analyzes complexity lower bounds for AdaGrad variants under $(L_0, L_1)$-smoothness conditions for finding approximate stationary points. A key result is that optimization under $(L_0, L_1)$-smoothness is inherently more challenging than under standard $L$-smoothness. This is demonstrated by lower bounds highlighting a gap compared to previous results for $L$-smooth functions.

A minor limitation of the work is that the results are derived for decorrelated variants of AdaGrad, which are less commonly used in practice.

In view of the fact that the study of $(L_0, L_1)$-smoothness is a timely topic, relevant to the ICLR audience, the present results complement the literature, which mostly focuses on deriving convergence upper bounds. Moreover, the findings presented in this paper can serve as a foundation for establishing lower bounds for adaptive algorithms under relaxed smoothness conditions, potentially extending to broader classes of algorithms.

**Additional Comments On Reviewer Discussion:**

The discussion between the authors and the reviewers clarified some technical questions and clarified which algorithms fall within the scope of the settings considered in this paper.

---

### Decision · Program_Chairs · 2025-01-22

Accept (Poster)